# Micropeptide hSPAR regulates glutamine levels and suppresses mammary tumor growth via a TRIM21-P27KIP1-mTOR axis

Yan Huang [1,2], Hua Lu[1,2,11], Yao Liu[3,11], Jiabei Wang[3,11], Qingan Xia[4,11], Xiangmin Shi[1,2,11], Yan Jin[1,2], Xiaolin Liang [1,2], Wei Wang[1,2], Xiaopeng Ma[5], Yangyi Wang [5], Meng Gong[1,2], Canjun Li[5], Chunlei Cang [5], Qinghua Cui[6,7], Ceshi Chen[8,9], Tao Shen [10✉], Lianxin Liu [3✉] & Xiangting Wang [1,2✉]

## Abstract

mTOR plays a pivotal role in cancer growth control upon amino acid response. Recently, CDK inhibitor P27KIP1 has been reported as a noncanonical inhibitor of mTOR signaling in MEFs, via unclear mechanisms. Here, we find that P27KIP1 degradation via E3 ligase TRIM21 is inhibited by human micropeptide hSPAR through its C-terminus (hSPAR-C), causing P27KIP1's cytoplasmic accumulation in breast cancer cells. Furthermore, hSPAR/hSPAR-C also serves as an inhibitor of glutamine transporter SLC38A2 expression and thereby decreases the cellular glutamine levels specifically in cancer cells. The resultant glutamine deprivation sequentially triggers translocation of cytoplasmic P27KIP1 to lysosomes, where P27KIP1 disrupts the Ragulator complex and suppresses mTORC1 assembly. Administration of hSPAR or hSPAR-C significantly impedes breast cancer cell proliferation and tumor growth in xenograft models. These findings define hSPAR as an intrinsic control factor for cellular glutamine levels and as a novel tumor suppressor inhibiting mTORC1 assembly.

**Keywords** Micropeptide/microprotein; mTOR; P27KIP1; Breast Cancer; TRIM21
**Subject Categories** Cancer; Metabolism; Signal Transduction

## Introduction

The mammalian target of rapamycin (mTOR), a serine/threonine protein kinase, is a part of the mTOR complex 1 (mTORC1) and acts as a central molecule essential for cell growth by connecting nutrient signals and the metabolic processes. In human cancer cells, mTOR is frequently activated and alters the activities of downstream targets, such as S6 kinases (S6K) and 4E-binding proteins (Alessi et al, 1997; Saxton and Sabatini, 2017; Zhang et al, 2017). For decades, amino acids, particularly—glutamine, arginine and leucine, have been shown to be essential nutrients for activating mTOR signaling (Bar-Peled and Sabatini, 2014; Mossmann et al, 2018; Sancak et al, 2010; Takahara et al, 2020). The cyclin-dependent kinase inhibitor 1B (CDKN1B or P27KIP1) was initially identified as a cyclin-dependent kinase inhibitor (CDKi) and is frequently found to be mis-regulated in cancer cells (Alkarain and Slingerland, 2004; Besson et al, 2008; Chu et al, 2008; Razavipour et al, 2020). Recently, it has been found that prolonged amino acid deprivation can induce noncanonical translocation of murine p27Kip1 to lysosomes, where it inhibits mTORC1 assembly in mouse embryo fibroblasts (MEFs) (Nowosad et al, 2020). However, the underlying mechanisms for intrinsic regulation of lysosomal P27KIP1-mTOR axis are not well understood.

Noncoding RNAs (ncRNAs) were previously defined as a class of RNA molecules that do not encode amino acid products. The gradual discovery of micropeptides—small open reading frames (sORFs) encoded from ncRNAs and less than 100 amino acids in length—has changed the traditional definition of ncRNAs (Andrews and Rothnagel, 2014; Makarewich and Olson, 2017; Wu et al, 2020b). Recent studies have indicated that micropeptides

[1]Department of Geriatrics, Gerontology Institute of Anhui Province, Centre for Leading Medicine and Advanced Technologies of IHM, The First Affiliated Hospital of USTC, Division of Life Sciences and Medicine, University of Science and Technology of China, Hefei, Anhui, China. [2]Anhui Provincial Key Laboratory of Tumor Immunotherapy and Nutrition Therapy, Hefei, Anhui, China. [3]Department of Hepatobiliary Surgery, Centre for Leading Medicine and Advanced Technologies of IHM, The First Affiliated Hospital of USTC, Division of Life Sciences and Medicine, University of Science and Technology of China, Hefei, Anhui, China. [4]Department of Pathology, Tangshan Gongren Hospital, Tangshan, Hebei, China. [5]Division of Life Sciences and Medicine, University of Science and Technology of China, Hefei, Anhui, China. [6]School of Sports Medicine, Wuhan Institute of Physical Education, Wuhan, Hubei, China. [7]Department of Biomedical Informatics, Centre for Noncoding RNA Medicine, State Key Laboratory of Vascular Homeostasis and Remodeling, School of Basic Medical Sciences, Peking University, Beijing, China. [8]Yunnan Key Laboratory of Breast Cancer Precision Medicine, Academy of Biomedical Engineering, Kunming Medical University, Kunming, Yunnan, China. [9]Yunnan Key Laboratory of Breast Cancer Precision Medicine, Yunnan Cancer Hospital, The Third Affiliated Hospital of Kunming Medical University, Peking University Cancer Hospital Yunnan, Kunming, Yunnan, China. [10]Anhui Provincial Key Laboratory of Molecular Enzymology and Mechanism of Major Metabolic Diseases, Anhui Provincial Engineering Research Centre for Molecular Detection and Diagnostics, College of Life Sciences, Anhui Normal University, Wuhu, Anhui, China. [11]These authors contributed equally: Hua Lu, Yao Liu, Jiabei Wang, Qingan Xia, Xiangmin Shi. ✉E-mail: stao@ahnu.edu.cn; Liulx@ustc.edu.cn; wangxt11@ustc.edu.cn

encoded by long noncoding RNAs (lncRNAs) functionally impact cancer development by regulating cell proliferation, angiogenesis, and metastasis (Huang et al, 2017; Li et al, 2023; Li et al, 2022; Li et al, 2020; Morgado-Palacin et al, 2023; Pang et al, 2020; Pei et al, 2023; Sun et al, 2021; Wang et al, 2020; Wu et al, 2020b; Wu et al, 2020c; Xiang et al, 2021; Xie et al, 2022; Xu et al, 2020; Yang et al, 2023; Zhang et al, 2018).

Recently, Matsumoto et al, found that *LINC00961* also has the capacity to encode functional micropeptide known as small regulatory polypeptide of amino acid response (SPAR) (Matsumoto et al, 2017; Spencer et al, 2020). *LINC00961* is located on human chromosome 9. Previous studies have shown that *LINC00961* functions as an lncRNA to regulate cell proliferation, invasion, and apoptosis in different cancer cells (Chen et al, 2019; Huang et al, 2018; Jiang et al, 2018; Lu et al, 2018; Mehrpour Layeghi et al, 2020; Mu et al, 2022; Pan and Sun, 2019; Wu et al, 2020a; Yin et al, 2019; Zhang et al, 2019). SPAR is conserved between mouse and human, and shows certain degree of tissue specificity (Matsumoto et al, 2017). A lysosomal-localization has been observed for SPAR in mouse muscle cells, HEK293T cells, cervical cancer HeLa cells and prostate cancer PC3 cells (Matsumoto et al, 2017). In amino acid re-stimulated HEK293T cells, lysosomal-localized SPAR tightly bound with V-ATPase and stabilized V-ATPase's interaction with the Ragulator complex to hamper the proper mTORC1 assembly (Matsumoto et al, 2017). Amino acid re-stimulation-dependent inhibitory effect of SPAR on mTOR signaling has also been observed in mouse myoblast C2C12 cells. By constructing SPAR knockout mice, SPAR downregulating enables efficient activation of mTORC1 and promotes muscle regeneration. Moreover, upon acute muscle injury induced by Cardiotoxin (CTX) on the wild-type and knockout mice, SPAR has been proven to interact with V-ATPase, inhibit mTORC1 signaling activity, and suppress muscle regeneration (Matsumoto et al, 2017). However, the function of human SPAR (hSPAR) in cancer is unexplored.

Here, we demonstrate that human SPAR (hSPAR) acts as a tumor suppressor in breast cancer cells through P27KIP1-dependent/V-ATPase-independent mTOR inactivation mechanism. Specifically, hSPAR fulfills its functions by two coordinate pathways: On one hand, hSPAR stabilizes the protein level of P27KIP1 in the cytoplasm by blocking TRIM21-mediated ubiquitin-proteasome degradation of P27KIP1. On the other hand, hSPAR acts as a glutamine inhibitor and triggers the translocation of accumulated P27KIP1 from the cytoplasm to lysosomes. The large amount of lysosomal P27KIP1 competitively binds to LAMTOR1, the critical component of the Ragulator complex and required for mTOR activation, thus almost completely blocking the cancerous mTORC1 complex assembly and mTOR signaling activity. Importantly, we have shown that these regulatory effects of hSPAR are absent in HEK293T cells. Furthermore, our study highlights the cytoplasmic C-terminal domain of hSPAR (hSPAR-C) is the key to convey such regulatory effects. Chemically synthesized TAT-hSPAR-C peptide administered by tail vein injection dramatically prevents the growth of xenograft breast tumors in mice. Collectively, our work uncovers the regulatory mechanisms of lysosomal P27KIP1-mTOR axis by identifying hSPAR as a promoter for P27KIP1 protein stability and intrinsic glutamine metabolism inhibitor to trigger lysosomal-localization of P27KIP1.

# Results

## The human micropeptide hSPAR acts as a tumor suppressor independently of its parental *LINC00961* in breast cancer

Upon analyzing The Cancer Genome Atlas (TCGA) database, we found that the expression level of *LINC00961* was significantly downregulated in breast invasive carcinoma (BRCA) (Fig. 1A), with similar patterns in all 4 listed subtypes (triple-negative, HER2 + , Luminal-A, and Luminal-B) (Fig. 1B). Consistently, we found that the expression of *LINC00961* was significantly lower in cultured Luminal-A cell line (MCF7), Luminal-B cell line (BT474), and three triple-negative breast cancer cell lines (MDA-MB-468, MDA-MB-231, and MDA-MB-453) than in the human breast epithelial cell line (MCF10A) (Fig. 1C).

Apart from encoding the lncRNA product, the human locus encoding *LINC00961* on chromosome 9 also harbors two putative ORFs with initiating ATG codons. ORF1 (starting from ATG1) encodes a putative micropeptide of 90 amino acids, with an extra 15 aa at the N-terminus that is not present in mouse SPAR (Fig. EV1A). ORF2 (starting from ATG2) is in-frame of ORF1 (Fig. EV1A) and encodes a putative micropeptide of 75 aa. To detect whether and how human SPAR (hSPAR) expresses in breast cancers, we generated polyclonal antibodies against the full-length hSPAR encoded from ORF1. Immunoblotting showed the expected ~10 KD band for endogenous hSPAR in the cultured cells, and the hSPAR levels in all the tested breast cancer cell lines were significantly reduced compared to the MCF10A (Fig. 1D). Upon conducting immunohistochemical (IHC) staining of 25 resected breast tumor tissues covering clinical subtypes including triple-negative, HER2 + , Luminal-A, and Luminal-B, we found that the hSPAR levels were dramatically reduced in all the tested breast tumoral tissues as compared to non-tumoral tissues (Figs. 1E and EV1B). The cellular and clinical data suggest that hSPAR is also expressed and downregulated in breast cancer.

To study the potential impacts of *LINC00961* and hSPAR on breast cancer, we built three hSPAR mutant constructs fused with C-terminal Flag tags (ΔATG1, ΔATG2, and ΔATG1 + 2) by mutating the ATG codon(s) to ATT on ORF1 or/and ORF2 (Fig. EV1C). Our results showed that only the ΔATG1 + 2 construct resulted in complete blockage of hSPAR translation (Fig. EV1D). Therefore, ΔATG1 + 2 was used in our subsequent experiments to distinguish the effects of the lncRNA *LINC00961* and the micropeptide hSPAR. Next, we conducted EdU incorporation assays in MDA-MB-231 cells transfected with siRNAs for *LINC00961* knockdown (targeting 3 discrete regions) or a scramble control siRNA. Knockdown of *LINC00961* by all three siRNAs led to significantly increased rates of cell proliferation (Figs. 1F,G and EV1E). To determine whether this effect was conferred by *LINC00961* or hSPAR, we transfected MDA-MB-231 cells with either control plasmid, translation-defective ΔATG1 + 2, or translation-competent Flag-tagged hSPAR (Fig. 1H,J) and subsequently conducted EdU incorporation assays. Compared to the control plasmid, expression of Flag-tagged hSPAR conferred a strong inhibitory effect on MDA-MB-231 cell proliferation (Fig. 1J,K). In contrast, no difference was detected between the cells transfected with control and ΔATG1 + 2 (Fig. 1J,K).

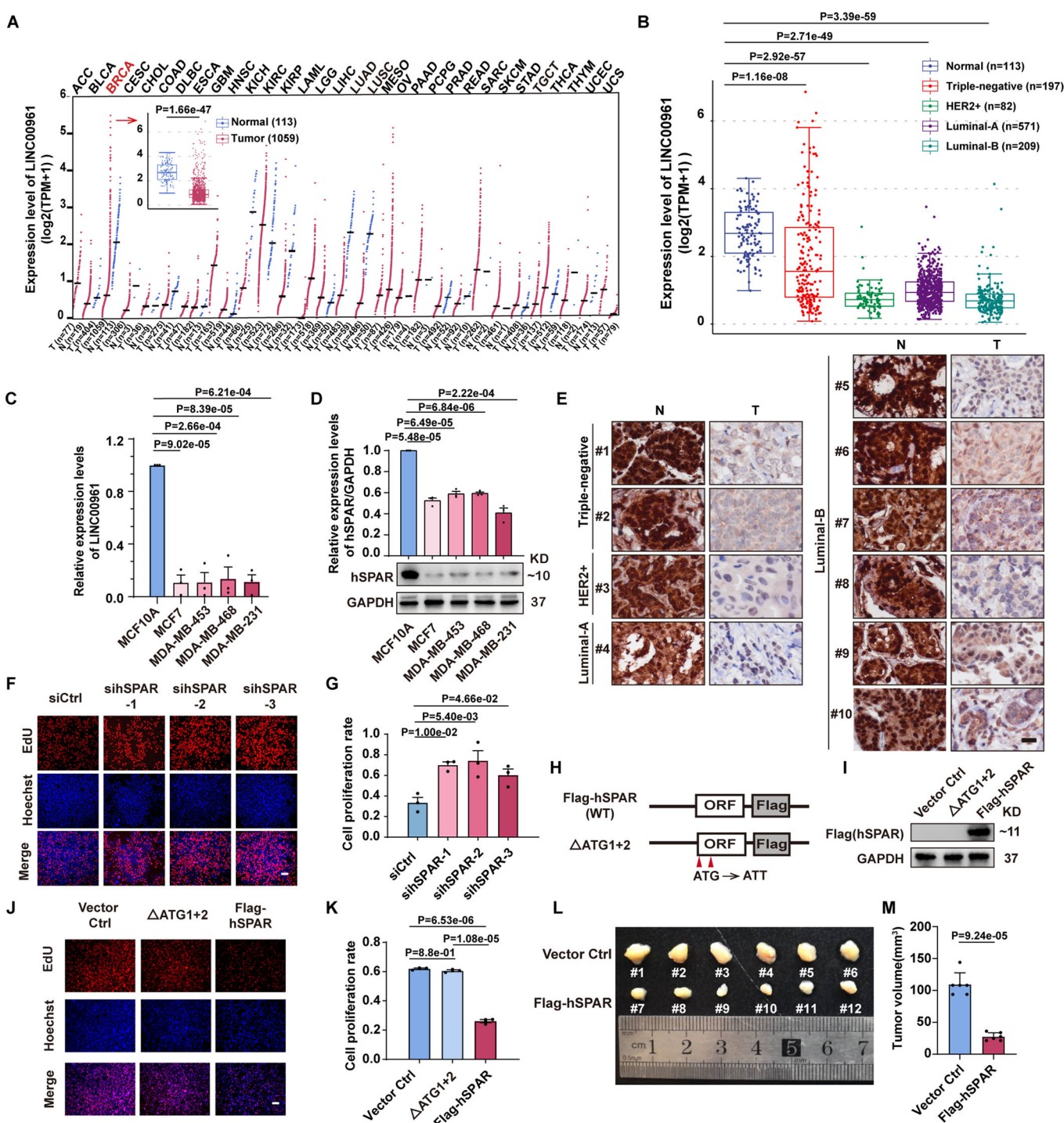

Consistent with the EdU results, immunostaining against β-Tubulin followed by quantification of mitotic phase cells showed that cells expressing Flag-tagged hSPAR failed to enter mitosis (Fig. EV1F,G). We also conducted a cell cycle analysis and found that hSPAR, but not *LINC00961*, triggered G1-S arrest in MDA-MB-231 cells (Fig. EV2B). By inoculating nude mice with MDA-MB-231 cells expressing control or Flag-tagged hSPAR, we found that the growth of breast cancer xenografts was significantly reduced for the samples expressing Flag-tagged hSPAR (Fig. 1L,M).

These results support that the micropeptide hSPAR confers a tumor suppressive function against breast cancer independently of its lncRNA.

## hSPAR suppresses cancer cell proliferation through mTOR signaling in a *LINC00961*-independent manner

To determine the subcellular localization of hSPAR in breast cancer cells, we used the hSPAR antibody to label endogenous

◀ **Figure 1. The micropeptide hSPAR acts as a tumor suppressor independently of its parental lncRNA *LINC00961* in breast cancer.**

See also Fig. EV1. (**A**) The RNA levels of *LINC00961* analyzed in different cancer tissues and their corresponding normal tissues from TCGA database. The RNA levels of *LINC00961* in breast cancer tissues and their corresponding normal tissues are shown in inserted figure (red arrow). The statistical graphs are generated using the GEPIA2 website (http://gepia2.cancer-pku.cn) and analyzed using one-way ANOVA. The box plots show the interquartile range (box limits), median (center line), min and max values with whiskers. The box plot information of normal tissues: Maximum: 4.31; Upper quartile: 3.3; Median: 2.68; Lower quartile: 2.09; Minimum: 0.99; Upper whisker: 1.01; Lower whisker: 1.10. The box plot information of breast cancer tissues: Maximum: 2.21; Upper quartile: 1.26; Median: 0.87; Lower quartile: 0.62; Minimum: 0.05; Upper whisker: 0.95; Lower whisker: 0.57. "*n*" represents the number of independent biological samples from tumoral tissues (T) or non-tumoral tissues (N). (**B**) The RNA levels of *LINC00961* analyzed in different subtypes (triple-negative, HER2 + , Luminal-A and Luminal-B) of breast cancer tissues and their corresponding normal tissues from TCGA database. The data are analyzed using one-way ANOVA. The box plots show the interquartile range (box limits), median (center line), min and max values with whiskers. The box plot information of normal tissues: Maximum: 4.31; Upper quartile: 3.3; Median: 2.68; Lower quartile: 2.09; Minimum: 0.99; Upper whisker: 1.01; Lower whisker: 1.10. The box plot information of triple-negative breast cancer tissues: Maximum: 5.81; Upper quartile: 2.86; Median: 1.56; Lower quartile: 0.80; Minimum: 0.08; Upper whisker: 2.95; Lower whisker: 0.72. The box plot information of HER2+ breast cancer tissues: Maximum: 1.3; Upper quartile: 0.91; Median: 0.72; Lower quartile: 0.53; Minimum: 0.17; Upper whisker: 0.39; Lower whisker: 0.36. The box plot information of Luminal-A breast cancer tissues: Maximum: 2.09; Upper quartile: 1.25; Median: 0.93; Lower quartile: 0.67; Minimum: 0.14; Upper whisker: 0.84; Lower whisker: 0.53. The box plot information of Luminal-B breast cancer tissues: Maximum: 1.48; Upper quartile: 0.88; Median: 0.68; Lower quartile: 0.48; Minimum: 0.05; Upper whisker: 0.60; Lower whisker: 0.43. "*n*" represents the number of independent biological samples from tumoral tissues (T) or non-tumoral tissues (N). (**C**) The RNA levels of *LINC00961* were analyzed by qPCR in the human breast epithelial cell MCF10A and indicated breast cancer cell lines (*n* = 3 independent biological samples). Data are presented as the mean ± SEM and analyzed using one-way ANOVA with Dunnett' multiple comparisons test. (**D**) The hSPAR levels assessed by immunoblotting with the anti-hSPAR antibody in MCF10A and indicated breast cancer cell lines (*n* = 3 independent biological samples). Data are presented as the mean ± SEM and analyzed using one-way ANOVA with Dunnett' multiple comparisons test. (**E**) The protein expression of hSPAR detected by immunohistochemistry in the breast cancer tumoral tissues (including triple-negative, HER2 + , Luminal-A and Luminal-B) and their adjacent non-tumoral tissues (*n* = 10 independent biological samples). More results are shown in Fig. EV1B. Scale bar, 20 µm. (**F**) Representative images of EdU assay in the presence of siCtrl or indicated si*hSPAR*s in triple-negative breast cancer MDA-MB-231 cells (*n* = 3 independent biological samples). Scale bar, 50 µm. (**G**) Quantification of cell proliferation rate from panel (**F**) (*n* = 3 independent biological samples). Data are presented as the mean ± SEM and analyzed using one-way ANOVA with Dunnett' multiple comparisons test. (**H**) Diagram of Flag-tagged hSPAR and its translation-defective construct (ΔATG1 + 2). WT, wild-type. (**I**) The Flag-hSPAR level assessed by immunoblotting with the anti-Flag antibody after transfection with Vector Ctrl, ΔATG1 + 2 or Flag-hSPAR in MDA-MB-231 cells (*n* = 3 independent biological samples). (**J**) Representative images of EdU assay in the presence of Vector Ctrl, ΔATG1 + 2 or Flag-hSPAR constructs in MDA-MB-231 cells (*n* = 3 independent biological samples). Scale bar, 50 µm. (**K**) Quantification of cell proliferation rate from panel (**J**) (*n* = 3 independent biological samples). Data are presented as the mean ± SEM and analyzed using one-way ANOVA with Dunnett' multiple comparisons test. (**L**) Images of xenografts from Balb/c (nu/nu) mice injected with Vector Ctrl or Flag-hSPAR expressed stable MDA-MB-231 cells (*n* = 12 independent biological samples). (**M**) Quantification of xenograft tumor volume from panel (**L**) (*n* = 12 independent biological samples). Data are presented as the mean ± SEM and analyzed using two-tailed Student's *t* test with Welch's correction. Source data are available online for this figure.

hSPAR. Confocal imaging revealed an aggregated distribution adjacent to the cell nucleus (Fig. 2A), and a subcellular fractionation analysis showed hSPAR enrichment in the membrane fraction (Fig. 2B). We further cloned the hSPAR ORF1 sequence in-frame with a C-terminal GFP epitope tag (Fig. 2C) and expressed this GFP-hSPAR fusion protein in MDA-MB-231 cells. Live-cell imaging analysis showed that GFP-labeled hSPAR co-localized with Lyso-Tracker labled lysosomes (Fig. 2D). The similar lysosomal distribution was supported by our immuno-fluorescence analysis, showing the co-localization of the Flag-hSPAR with the lysosomal marker LAMP1 signals, or the Flag-hSPAR with the lysosomal marker Lyso-Tracker signals (Figs. 2E and EV2A).

mTOR is frequently activated in diverse carcinomas and also widely accepted as a master regulator for cell proliferation control (Bar-Peled and Sabatini, 2014; Ben-Sahra and Manning, 2017; Mossmann et al, 2018). Thus, we started to explore the potential impacts of hSPAR on mTOR activation in breast cancer cells, by investigating downstream consequences of mTOR signaling in MDA-MB-231 cells in the presence of hSPAR. While there were no differences in the levels of mTOR, S6K, or S6, we observed significantly reduced levels of p-mTOR, p-S6K, and p-S6 in cells expressing Flag-hSPAR (Fig. 2F,G). Importantly, the ΔATG1 + 2 failed to alter levels of p-mTOR, p-S6K, and p-S6 (Fig. 2F,G). These results suggest that micropeptide hSPAR suppresses mTOR signaling activation in breast cancer cells independently of its lncRNA. Consistently, we collected clinical breast cancer specimens and detected the signals of hSPAR and p-mTOR, p-S6K and p-S6. Our results showed a negative correlation between the hSPAR signal intensity and the signals for p-mTOR, p-S6K, and p-S6 in breast tumoral tissues and non-tumoral tissues (Fig. 2H,I).

To further investigate whether hSPAR suppresses breast cancer cell proliferation through mTOR signaling, we transfected control, ΔATG1 + 2, or Flag-hSPAR plasmids into MDA-MB-231 cells in which we blocked mTOR signaling using rapamycin, followed by EdU incorporation assays. We observed the complete loss of hSPAR's effects under rapamycin treatment (Fig. 2J–L). These data support the notion that hSPAR suppresses cancer cell proliferation through mTOR signaling in a *LINC00961*-independent manner.

## The E3 ubiquitin ligase TRIM21 interacts with hSPAR and mediates hSPAR's effects on mTOR signaling and cell proliferation

Aiming to identify the underlying mechanism for hSPAR's suppression of mTOR activation, we profiled the hSPAR inter-actome based on a co-immunoprecipitation (Co-IP) analysis followed by mass spectrometry in MDA-MB-231 cells. Briefly, Co-IP samples were resolved using SDS-PAGE followed by silver staining, and the bands uniquely present in the hSPAR Co-IP samples were excised and analyzed with mass spectrometry. Tripartite motif-containing 21 (TRIM21), an E3 ubiquitin ligase (Kiss et al, 2021; Pan et al, 2016; Wang et al, 2021; Zhu et al, 2022) was among the top-ranking candidate hSPAR-interacting proteins (Fig. 3A). Subsequently, the interaction of hSPAR and TRIM21 was successfully validated by Co-IP followed by western blot analysis (Fig. 3B).

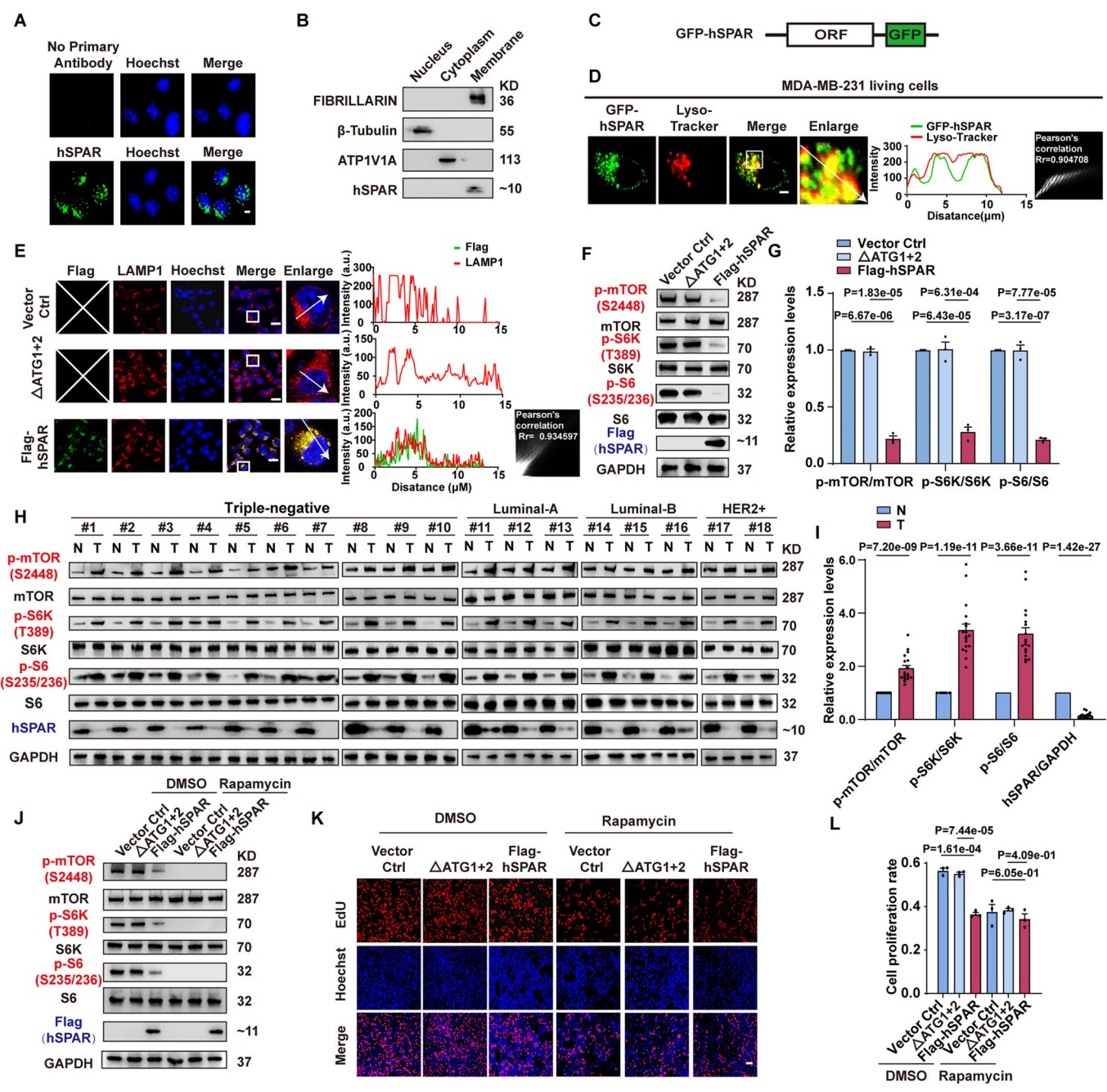

To investigate the potential functions of TRIM21 in the observed hSPAR-mediated suppression of mTOR activation and cell proliferation, we investigated downstream targets of mTOR signaling in MDA-MB-231 cells in the presence of both Flag-hSPAR and GFP-TRIM21. We found that co-expression of GFP-TRIM21 rescued the Flag-hSPAR-mediated changes in mTOR signaling (Fig. 3C,D), cell proliferation (Fig. 3E,F), and cell cycle progression (Fig. EV2B). PI3K-AKT is a previously reported upstream regulatory pathway of mTOR signaling. Our results showed that overexpressing hSPAR did not affect serine 473 phosphorylation of AKT, an indicator of the PI3K activation (Appendix Fig. S1). These results demonstrate that the inhibitory roles of hSPAR in mTOR signaling and cell proliferation require TRIM21.

## hSPAR stabilizes P27KIP1 by disrupting TRIM21-dependent ubiquitination of P27KIP1

Interestingly, in our Flag-hSPAR-expressing cells, we observed a decreased TRIM21-P27KIP1 interaction and a protein level upregulation of P27KIP1, which is a known substrate for E3 ubiquitin ligase TRIM21 (Sabile et al, 2006), compared to Vector control or ΔATG1 + 2 cells (Fig. 4A). These findings suggest that the micropeptide hSPAR disrupts TRIM21's capacity to recognize

**Figure 2.  hSPAR suppresses mTOR signaling and is oppositely correlated with mTOR activation in breast cancer patients.**

See also Fig. EV2. (**A**) Immunofluorescence analysis of hSPAR using an anti-hSPAR antibody in MDA-MB-231 cells ($n = 3$ independent biological samples). Scale bar, 5 μm. (**B**) Immunoblotting against hSPAR, FIBRILLARIN (nuclear marker), β-Tubulin (cytoplasmic marker), and ATP1V1A (membrane marker) in nuclear, cytoplasmic (membrane components removed), and membrane fractions prepared from MDA-MB-231 cells ($n = 3$ independent biological samples). (**C**) Diagram of GFP-tagged hSPAR. (**D**) Living-cell images of GFP-hSPAR and lysosomes (Red, labeled with Lyso-Tracker) in MDA-MB-231 cells. The graphs display the fluorescence intensity (arbitrary units) of GFP-hSPAR and Lyso-Tracker over the distance from adjacent image (depicted by the arrow). The value of Pearson's correlation Rr of SPAR and Lyso-Tracker is 0.907408. Scale bar, 5 μm, ($n = 3$ independent biological samples). (**E**) Co-immunofluorescence staining of Flag (green) and the lysosomal marker LAMP1 (red) in MDA-MB-231 cells transfected with Vector Ctrl, ΔATG1 + 2, or Flag-hSPAR. Nuclei were stained with Hoechst (blue). The Vector Ctrl and ΔATG1 + 2 are negative control groups, and Flag proteins are not expressed (The white "X" in Vector Ctrl and ΔATG1 + 2 groups indicates no green fluorescent signals). The graphs display the fluorescence intensity (arbitrary units) of Flag and the lysosomal marker LAMP1 over the distance from adjacent image (depicted by the arrows). The value of Pearson's correlation Rr of Flag-hSPAR and LAMP1 is 0.934597. Scale bar, 25 μm ($n = 3$ independent biological samples). (**F**) Immunoblotting against p-mTOR, mTOR, p-S6K, S6K, p-S6, S6, Flag and GAPDH in extracts from MDA-MB-231 cells transfected with Vector Ctrl, ΔATG1 + 2, or Flag-hSPAR ($n = 3$ independent biological samples). (**G**) Quantified relative levels of p-mTOR/mTOR, p-S6K/S6K, and p-S6/S6 from panel (**F**) ($n = 3$ independent biological samples). Data are presented as the mean ± SEM and analyzed using one-way ANOVA with Dunnett' multiple comparisons test. (**H**) Immunoblotting against p-mTOR, mTOR, p-S6K, S6K, p-S6, S6, hSPAR, and GAPDH in extracts from the breast cancer tumoral tissues and their adjacent non-tumoral tissues ($n = 18$ independent biological samples). (**I**) Quantified relative levels of p-mTOR/mTOR, p-S6K/S6K, p-S6/S6, and hSPAR/GAPDH from panel (**H**) ($n = 18$ independent biological samples). Data are presented as the mean ± SEM and analyzed using two-tailed Student's $t$ test with Welch's correction. (**J**) Immunoblotting against p-mTOR, mTOR, p-S6K, S6K, p-S6, S6, Flag, and GAPDH in extracts from MDA-MB-231 cells transfected with Vector Ctrl, ΔATG1 + 2, or Flag-hSPAR, with or without Rapamycin treatment (an mTOR inhibitor; 10 μM for 12 h) ($n = 3$ independent biological samples). (**K**) Representative images of EdU assay in the presence of Vector Ctrl, ΔATG1 + 2 or Flag-hSPAR, with or without Rapamycin treatment (10 μM for 12 h) ($n = 3$ independent biological samples). Scale bar, 50 μm. (**L**) Quantification of cell proliferation rate from panel (**K**) ($n = 3$ independent biological samples). Data are presented as the mean ± SEM and analyzed using one-way ANOVA with Dunnett' multiple comparisons test. The hSPAR-regulated proteins shown by immunoblotting are marked by red text. Source data are available online for this figure.

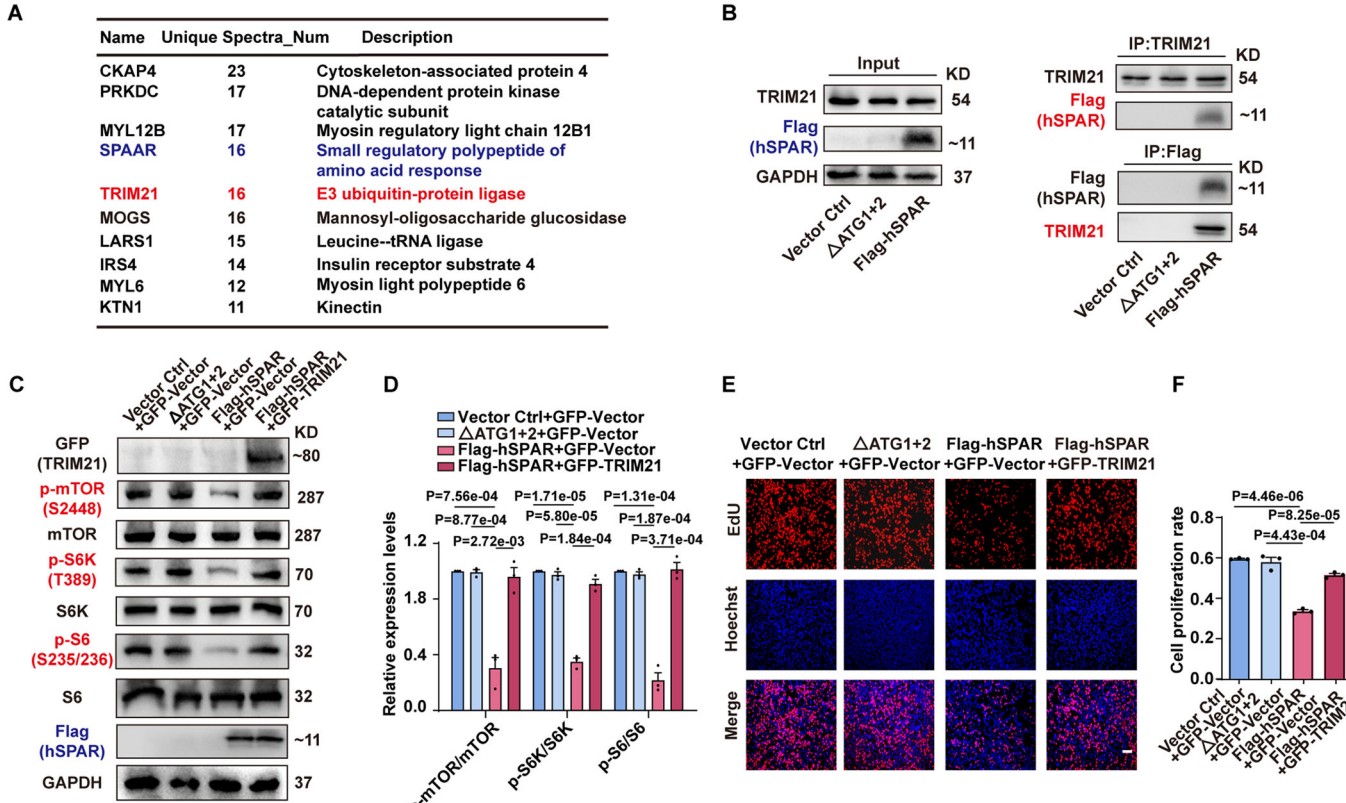

**Figure 3.   Identification of E3 ligase TRIM21 as an hSPAR-interacting protein and target to mediate hSPAR's effects in MDA-MB-231 cells.**

See also Fig. EV2. (**A**) List of the top10 hSPAR-interacting proteins identified by hSPAR Co-IP analysis and mass spectrometry. (**B**) Interaction of TRIM21 and Flag-hSPAR detected by Co-IP and immunoblotting from MDA-MB-231 cells transfected with the indicated constructs ($n = 3$ independent biological samples). (**C**) Immunoblotting against GFP, p-mTOR, mTOR, p-S6K, S6K, p-S6, S6, Flag, and GAPDH for extracts from MDA-MB-231 cells transfected with the indicated constructs ($n = 3$ independent biological samples). (**D**) Quantified relative levels of p-mTOR/mTOR, p-S6K/S6K, and p-S6/S6 from panel (**C**) ($n = 3$ independent biological samples). Data are presented as the mean ± SEM and analyzed using one-way ANOVA with Dunnett' multiple comparisons test. (**E**) Representative images of EdU assay in MDA-MB-231 cells transfected with the indicated constructs ($n = 3$ independent biological samples). Scale bar, 50 μm. (**F**) Quantification of cell proliferation rate from panel (**E**) ($n = 3$ independent biological samples). Data are presented as the mean ± SEM and analyzed using one-way ANOVA with Dunnett' multiple comparisons test. The hSPAR-regulated proteins shown by immunoblotting are marked by red text. Source data are available online for this figure.

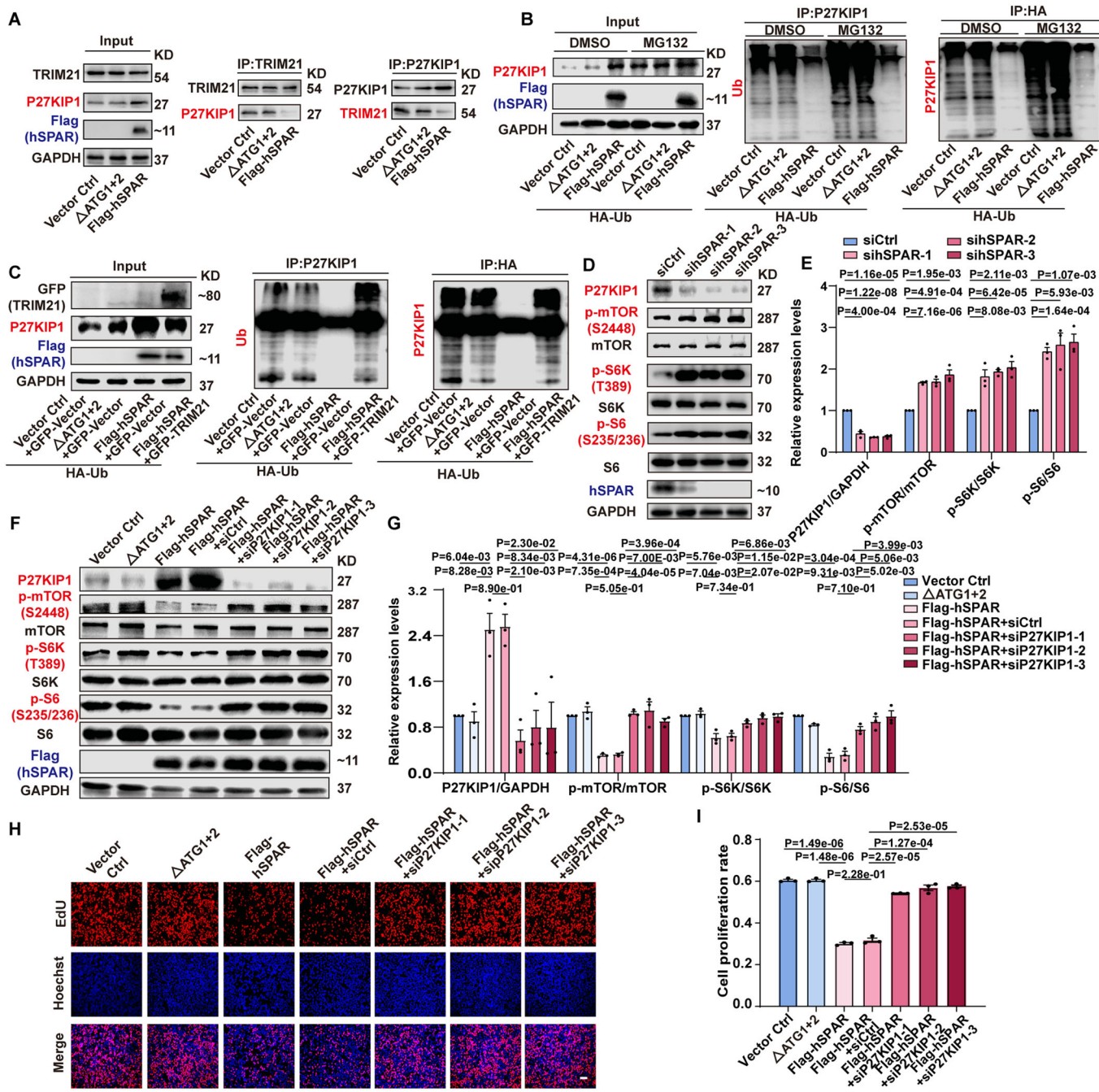

and ubiquitinate P27KIP1 for degradation. Supporting that, we observed the poly-ubiquitination level of P27KIP1 was greatly reduced in Flag-hSPAR-expressing cells by immunoprecipitation assay (Fig. EV2C). In addition, treatment with the proteasome inhibitor MG132 resulted in the accumulation of ubiquitylated P27KIP1 in Vector Ctrl and ΔATG1 + 2 cells, but not Flag-hSPAR cells (Fig. 4B). Moreover, we showed that the Flag-hSPAR-reduced ubiquitination of P27KIP1 could be rescued upon co-expression of GFP-TRIM21 (Fig. 4C). Consistent with the results from hSPAR overexpression, we showed that hSPAR knockdown could result in a significant reduction in the protein level of P27KIP1 (Fig. 4D,E).

TRIM21's E3 ubiquitin ligase activity depends on forming a SCF$^{skp2}$ complex with SKP1, SKP2, and Cul1 (Sabile et al, 2006). The SCF$^{skp2}$-mediated ubiquitination degradation process of P27KIP1 consists of two key steps: (1) The phosphorylation-modified P27KIP1 on threonine residue 187 is recognized by SKP2; (2) SKP2 binds to TRIM21 and mediates the ubiquitination of P27KIP1 by TRIM21. Based on these steps, we further explored the effect of hSPAR on the phosphorylation level of P27KIP1 and the interaction of TRIM21 and SKP2 by immunoblotting and Co-IP. We observed that overexpression of hSPAR did not affect the phosphorylation level of P27KIP1 (Fig. EV2D,E). However, over-expression of hSPAR significantly decreased TRIM21-SKP2

**Figure 4.   hSPAR stabilizes P27KIP1 by disrupting TRIM21-dependent ubiquitination on P27KIP1 in MDA-MB-231 cells.**

See also Fig. EV2. (**A**) Interaction of TRIM21 and P27KIP1 detected by Co-IP and immunoblotting from MDA-MB-231 cells transfected with the indicated constructs ($n = 3$ independent biological samples). (**B**) Changes of the ubiquitination level of P27KIP1 detected by Co-IP and immunoblotting from MDA-MB-231 cells transfected with Vector Ctrl, $\Delta$ATG1 + 2 or Flag-hSPAR, together with HA-Ub, followed by treatment with DMSO or the proteasome inhibitor MG132 (5 µM). Left, immunoblotting of inputs. Middle, immunoblotting using antibody against ubiquitin following IP of P27KIP1. Right, immunoblotting using the antibody against P27KIP1 following IP of HA ($n = 3$ independent biological samples). (**C**) Changes of the ubiquitination level of P27KIP1 after co-transfection with GFP-TRIM21 in the presence of Flag-hSPAR and HA-Ub detected by Co-IP and immunoblotting from MDA-MB-231 cells. Left, immunoblotting of inputs. Middle, immunoblotting using antibody against ubiquitin following IP of P27KIP1. Right, immunoblotting using the antibody against P27KIP1 following IP of HA ($n = 3$ independent biological samples). (**D**) Immunoblotting against P27KIP1, p-mTOR, mTOR, p-S6K, S6K, p-S6, S6, hSPAR, and GAPDH in extracts from MDA-MB-231 cells in the presence of siCtrl or indicated si*hSPAR*s ($n = 3$ independent biological samples). (**E**) Quantified relative levels of P27KIP1/GAPDH, p-mTOR/mTOR, p-S6K/S6K, and p-S6/S6 from panel (**D**) ($n = 3$ independent biological samples). Data are presented as the mean ± SEM and analyzed using one-way ANOVA with Dunnett' multiple comparisons test. (**F**) Immunoblotting against P27KIP1, p-mTOR, mTOR, p-S6K, S6K, p-S6, S6, Flag, and GAPDH for extracts from MDA-MB-231 cells transfected with the indicated controls, and Flag-hSPAR with or without P27KIP1 knockdown ($n = 3$ independent biological samples). (**G**) Quantified relative levels of P27KIP1/GAPDH, p-mTOR/mTOR, p-S6K/S6K and p-S6/S6 from panel (**F**) ($n = 3$ independent biological samples). Data are presented as the mean ± SEM and analyzed using one-way ANOVA with Dunnett' multiple comparisons test. (**H**) Representative images of EdU assay in the indicated controls, and Flag-hSPAR with or without P27KIP1 knockdown ($n = 3$ independent biological samples). Scale bar, 50 µm. (**I**) Quantification of cell proliferation rate from panel (**H**) ($n = 3$ independent biological samples). Data are presented as the mean ± SEM and analyzed using one-way ANOVA with Dunnett' multiple comparisons test. The hSPAR-regulated proteins shown by immunoblotting are marked by red text. Source data are available online for this figure.

interaction (Fig. EV2F). These results suggest that hSPAR inhibits TRIM21-dependent ubiquitination of P27KIP1 by affecting the interaction between TRIM21 and SKP2 to hinder the formation of the SCF$^{skp2}$ complex.

P27KIP1 is also a CDK inhibitor protein known to function in regulating the G1-S transition, and P27KIP1 deficiency has been clinically associated with poor outcomes in breast cancer patients (Alkarain and Slingerland, 2004; Besson et al, 2008; Chu et al, 2008; Razavipour et al, 2020). Thus, we next investigated the potential involvement of hSPAR-mediated P27KIP1 upregulation on mTOR signaling inactivation and cell proliferation inhibition. Our results showed that mTOR signaling activation and down-regulation of P27KIP1 correspondingly occurred in hSPAR knockdown cells (Fig. 4D,E). Further, when we knocked down P27KIP1 in Flag-hSPAR-expressing MDA-MB-231 cells, we observed P27KIP1 knockdown completely restored the activation of mTOR signaling inhibited by Flag-hSPAR (Fig. 4F,G). P27KIP1 knockdown also reversed the cell proliferation/cell cycle effects suppressed by Flag-hSPAR (Figs. 4H,I and EV2G). Thus, our results demonstrate that hSPAR can stabilize the cytosol protein level of P27KIP1 by disrupting TRIM21-dependent ubiquitination of P27KIP1. The accumulated P27KIP1 further mediates the roles of hSPAR in mTOR signaling inactivation and cell proliferation suppression.

## hSPAR inhibits the glutamine transporter SLC38A2 and induces glutamine-associated lysosomal-localization of P27KIP1

Next, we investigated whether hSPAR affected the lysosomal localization of P27KIP1 in breast cancer cells. Our data showed that Flag-hSPAR overexpressing did not change P27KIP1 signals in the nucleus fraction (Fig. 5A), but triggered dramatically increased signals of P27KIP1 in the membrane fraction (Fig. 5A) and the lysosomal fraction in MDA-MB-231 cells (Fig. 5B,C). Given that TRIM21 is also the upstream expression regulator of P27KIP1 (Fig. 4A–C), we sought to determine whether TRIM21 affects the lysosomal-localization of P27KIP1. Subcellular fractionation ana-lysis showed that knockdown of TRIM21 resulted in increased expression of P27KIP1 in cytoplasm fraction without causing changes of P27KIP1's lysosomal localization (Fig. 5D). These

results suggest that the lysosome translocation of P27KIP1 is independent of TRIM21.

It is worth noting that the hSPAR's effects on P27KIP1 in our system were observed under complete culturing media, whereas the reported lysosomal-localization of p27Kip1 was found upon prolonged amino acid deprivation in MEFs (Nowosad et al, 2020). We therefore hypothesized that hSPAR may possess TRIM21-independent roles in regulating amino acid metabolism to promote P27KIP1's lysosomal localization observed in Fig. 5A–C. We started to test our hypothesis by performing metabolomics. As expected, we found that, glutamine, one of the strongest activators of mTOR signaling (Guo et al, 2023), was among the top differentially downregulated metabolites after expressing Flag-hSPAR in MDA-MB-231 cells (Fig. 5E; Table EV1). Interestingly, metabolomics analysis in HEK293T cells showed no effects on amino acid metabolites (Fig. 5E). Supportively, we found that Flag-hSPAR expressing MDA-MB-231 cells resulted in barely detectable level of glutamine 24 h after transfection (Fig. 5F). In contrast, Flag-hSPAR expressing HEK293T cells and MCF10A cells showed no changes on cellular glutamine level (Figs. 5F and EV3A).

To determine whether glutamine controls lysosomal-localization of P27KIP1, we performed immunofluorescence and subcellular fractionation assays with or without glutamine supplementation in MDA-MB-231 cells. After removing soluble P27KIP1, confocal images of P27KIP1 showed significant co-localization of P27KIP1 with lysosomes upon glutamine starvation (Figs. 5G and EV3B), which was further confirmed by subcellular fractionation analysis (Fig. 5H). Next, we sought to explore the underlying mechanism by which hSPAR controls the cellular glutamine uptake. Transporter SLC38A2 represents one of the major promoters of glutamine entry into cells under physiological conditions and is frequently over-expressed in cancer cells (Guo et al, 2023; Morotti et al, 2021). We found that the protein levels of SLC38A2 were significantly decreased in Flag-hSPAR expressing MDA-MB-231 cells, while no changes of SLC38A2 were observed in HEK293T cells (Fig. 5I). Consistently, knockdown of hSPAR in MDA-MB-231 cells significantly increased the cellular glutamine level, which could be restored by co-transfecting SLC38A2 siRNAs (Fig. 5J). The subcellular fractionation experiments following with immunoblot-ting further confirmed that knockdown of SLC38A2 promoted the

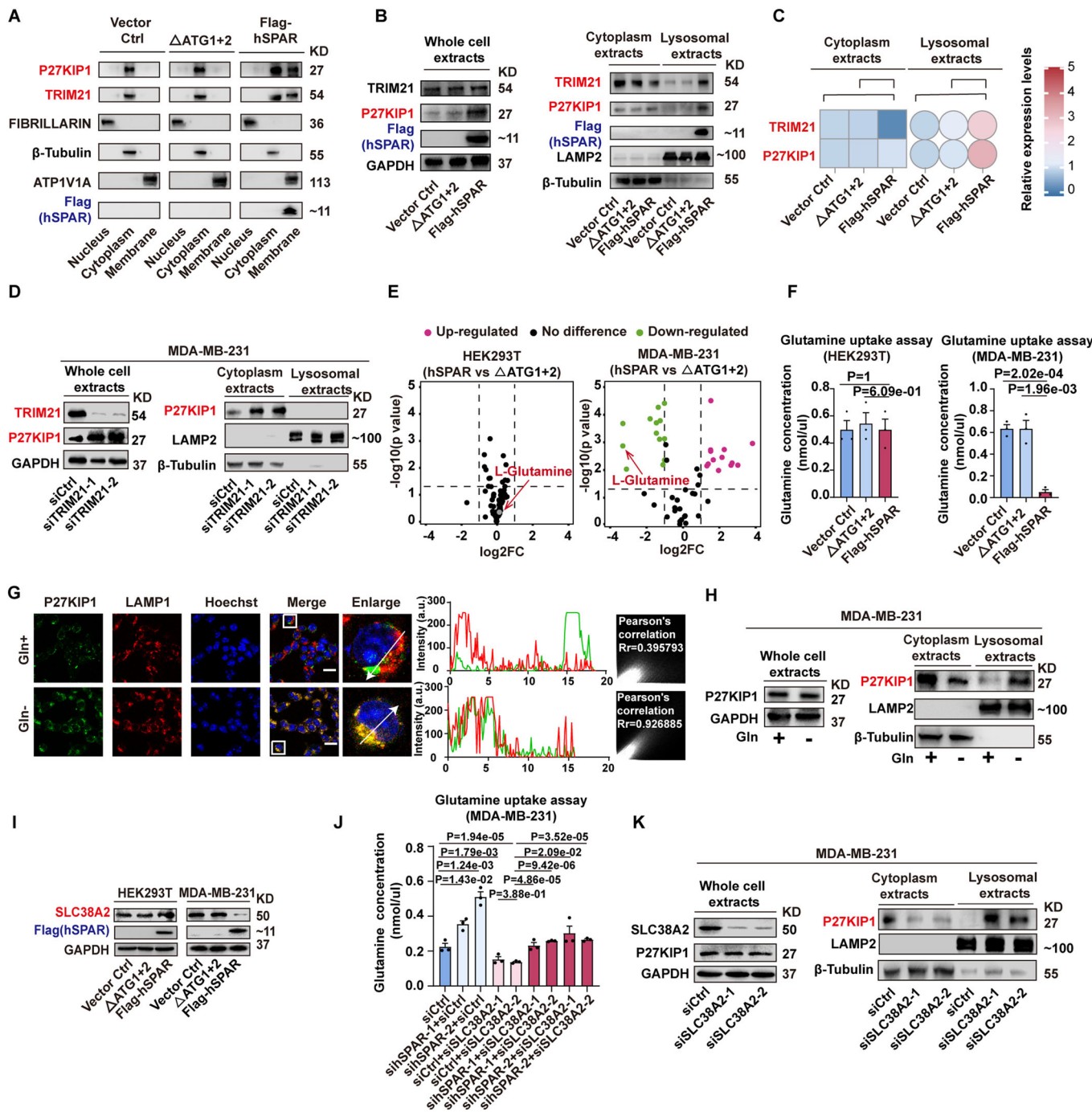

lysosomal recruitment of P27KIP1 without affecting the overall expression of P27KIP1 in MDA-MB-231 cells (Fig. 5K).

In addition to glutamine, arginine, and leucine are also key amino acids to activate mTOR signaling (Sancak et al, 2010). Therefore, we further investigated whether hSPAR would affect the expression of arginine and leucine transporters, and whether arginine or leucine deficiency would affect the lysosomal localization of P27KIP1. Our results

showed that hSPAR overexpressing had no effect on the expression levels of arginine transporter SLC7A1 or leucine transporter SLC7A5 (Fig. EV3C,D). The subcellular localization of P27KIP1 remained unaltered in MDA-MB-231 cells following arginine or leucine starvation treatment (Fig. EV3E). Collectively, our data suggest that hSPAR can downregulate SLC38A2 expression, deprive cellular glutamine level, and trigger P27KIP1's lysosomal translocation in breast cancer cells.

**Figure 5. hSPAR induces P27KIP1's lysosomal localization by downregulating SLC38A2 expression and depriving cellular glutamine in MDA-MB-231 cells.**

See also Fig. EV3. (A) Immunoblotting of nuclear, cytoplasmic (membrane components removed), and membrane fractions prepared from MDA-MB-231 cells transfected with the indicated constructs against P27KIP1, TRIM21, FIBRILLARIN (nuclear marker), β-Tubulin (cytoplasmic marker), ATP1V1A (membrane marker), and Flag ($n = 3$ independent biological samples). (B) Immunoblotting of whole-cell extracts (left panel), cytoplasmic (lysosome components removed) and lysosomal extracts (right panel) prepared from MDA-MB-231 cells transfected with the indicated constructs against TRIM21, P27KIP1, Flag, GAPDH, LAMP2 (lysosomal marker), and β-Tubulin (cytoplasmic marker) ($n = 3$ independent biological samples). (C) Heatmap shows the relative expression analysis of the displayed experiment for TRIM21 and P27KIP1 to β-Tubulin (cytoplasmic fractions) or LAMP2 (lysosomal fractions) from panel (B) ($n = 3$ independent biological samples). (D) Immunoblotting of whole-cell extracts (left panel), cytoplasmic (lysosome components removed) and lysosomal extracts (Right panel) prepared from MDA-MB-231 cells transfected with siCtrl or indicated siTRIM21s against TRIM21, P27KIP1, GAPDH, LAMP2 (lysosomal marker), and β-Tubulin (cytoplasmic marker) ($n = 3$ independent biological samples). (E) Volcano plot showing differential amino acid metabolite levels between MDA-MB-231 (right panel) or HEK293T cells (left panel) transfected with ΔATG1 + 2 and Flag-hSPAR ($n = 3$ independent biological samples). The regulated metabolites between ΔATG1 + 2 and Flag-hSPAR are determined by fold change analysis. The significance is presented as *P* value and analyzed using hypergeometric test. Red dots, significantly upregulated metabolites. Green dots, significantly downregulated metabolites. Black dots, non-different metabolites. (F) Levels of glutamine in MDA-MB-231 or HEK293T cells transfected with Vector Ctrl, ΔATG1 + 2 or Flag-hSPAR ($n = 3$ independent biological samples). Data are presented as the mean ± SEM and analyzed using one-way ANOVA with Dunnett' multiple comparisons test. (G) Co-immunofluorescence staining of P27KIP1 (green) and the lysosomal marker LAMP1 (red) in MDA-MB-231 cells cultured with or without glutamine. Cells were permeabilized with digitonin to remove the soluble P27KIP1. Nuclei were stained with Hoechst (blue). The graphs display the fluorescence intensity (arbitrary units) of P27KIP1 and LAMP1 over the distance from adjacent image (depicted by the arrows). The graphs display the values of Pearson's correlation Rr of P27KIP1 and LAMP1 with or without glutamine. Scale bar, 25 μm ($n = 3$ independent biological samples). (H) Immunoblotting of whole-cell extracts (Left panel), cytoplasmic (lysosome components removed) and lysosomal extracts (right panel) prepared from MDA-MB-231 cells cultured with or without glutamine against P27KIP1, GAPDH, LAMP2 (lysosomal marker) and β-Tubulin (cytoplasmic marker) ($n = 3$ independent biological samples). (I) The protein levels of SLC38A2 detected by immunoblotting with the anti-SLC38A2 antibody after transfection with Vector Ctrl, ΔATG1 + 2 or Flag-hSPAR in MDA-MB-231 and HEK293T cells. GAPDH, loading control ($n = 3$ independent biological samples). (J) Levels of glutamine in MDA-MB-231 cells in the presence of siCtrl, si*hSPAR*s, siSLC38A2s, or si*hSPAR*s together with siSLC38A2s ($n = 3$ independent biological samples). Data are presented as the mean ± SEM and analyzed using one-way ANOVA with Dunnett' multiple comparisons test. (K) Immunoblotting of whole-cell extracts (left panel), cytoplasmic (lysosome components removed) and lysosomal extracts (right panel) prepared from MDA-MB-231 cells transfected with siCtrl or indicated siSLC38A2s against SLC38A2, P27KIP1, GAPDH, LAMP2 (lysosomal marker), and β-Tubulin (cytoplasmic marker) ($n = 3$ independent biological samples). The hSPAR-regulated proteins shown by immunoblotting are marked by red text. Source data are available online for this figure.

## Lysosomal localized P27KIP1 competes with Ragulator-complex subunits LAMTOR2-5 for LAMTOR1 binding to prevent mTORC1 assembly complex in breast cancer cells

Next, we attempted to explore the impact of lysosomal localized P27KIP1 on the mTOR signaling. Our subcellular fractionation experiments revealed the reduced lysosomal localizations of LAMTOR2-5 in Flag-hSPAR overexpressing cells compared to control and ΔATG1 + 2 cells (Fig. 6A,B). Moreover, Co-IP of extracts from MDA-MB-231 cells further confirmed that Flag-hSPAR induced the P27KIP1–LAMTOR1 interaction, resulting in a significant decrease in the interaction between LAMTOR1 and LAMTOR2–5 (Fig. 6C). Known that LAMTOR1 is one of Ragulator-complex subunits, which plays a crucial role in responding to amino acids and regulating the activation of mTOR on the lysosomal surface, and provides lysosomal anchor and scaffold for other Ragulator-complex subunits (LAMTOR2–5) and Rag GTPases (Rags) (Nowosad et al, 2020; Sancak et al, 2010; Sancak et al, 2008; Su et al, 2017). We also detected LAMTOR1-mTOR and LAMTOR1-RagA interactions in cells expressing Flag-hSPAR. Our results showed a significantly decreased LAMTOR1-mTOR and LAMTOR1-RagA interactions in cells expressing Flag-hSPAR (Fig. 6C). These data strongly suggest that hSPAR-induced lysosomal P27KIP1 inhibits the assembly of Ragulator-Rags–mTORC1 complex.

Considering our observation that P27KIP1 is not shown as an hSPAR-interacting protein (Fig. 3A) and that dissociation of TRIM21 with P27KIP1 in the hSPAR overexpressing cells (Fig. 4A), we proposed that the lysosomal hSPAR and TRIM21 are located separately with P27KIP1 and LAMTOR1 complex on the lysosome. As expected, our Co-IP assays using lysosomal extracts showed that P27KIP1 did not interact with TRIM21 but was bound to LAMTOR1 (Fig. EV3F).

Next, we proceeded to investigate whether combination of TRIM21 knockdown and glutamine deprivation would mimic the regulatory impact of hSPAR on mTOR. The subcellular fractionation experiments showed that siTRIM21-accumulated P27KIP1 in the cytoplasm was indeed translocated to lysosome upon siSLC38A2 co-treatment (Fig. 6D,E). In sharp contrast, Ragulator subunits (LAMTOR2–5), RagA, the total mTOR and active p-mTOR were decreased in the lysosomal fraction upon TRIM21 and SLC38A2 double-knockdown treatment, compared to control or TRIM21 single-knockdown treatment (Fig. 6D,E). Remarkably, treating MDA-MB-231 cells with either siTRIM21 or siSLC38A2 individually only led to a partial reduction in mTOR signaling activity, with a maximum decrease of 50% (Fig. 6F,G). However, simultaneous knockdown of both TRIM21 and SLC38A2 resulted in nearly complete inactivation of mTOR signaling (Fig. 6F,G), akin to the inhibitory effect of hSPAR observed in Figs. 2F, 3C, and 4F. Collectively, these results suggest that coordination of increased stability and glutamine-deprivation-induced lysosomal-localization of P27KIP1 license P27KIP1's inhibitory effect on the assembly of Ragulator-Rags-mTORC1 complex and mTOR activity. Similar to MDA-MB-231 cells, the effects of hSPAR on SLC38A2, P27KIP1, mTOR signaling, and cell proliferation were also observed in breast cancer MCF7 cells (Luminal-A subtype) (Fig. EV4I–L) and MDA-MB-468 cells (triple-negative subtype) (Fig. EV4E-H), but not in HEK293T cells (Fig. EV4A–D). These results suggest that hSPAR possesses distinct regulatory effects in breast cancer cells and HEK293T cells.

## The cytoplasmic domain of hSPAR is sufficient to trigger TRIM21/glutamine-P27KIP1-mTOR signaling and suppress xenograft tumor growth

Analysis of the predicted hSPAR structure by Alphafold2 suggested that hSPAR ORF1 comprises a lumenal N-terminal domain, a

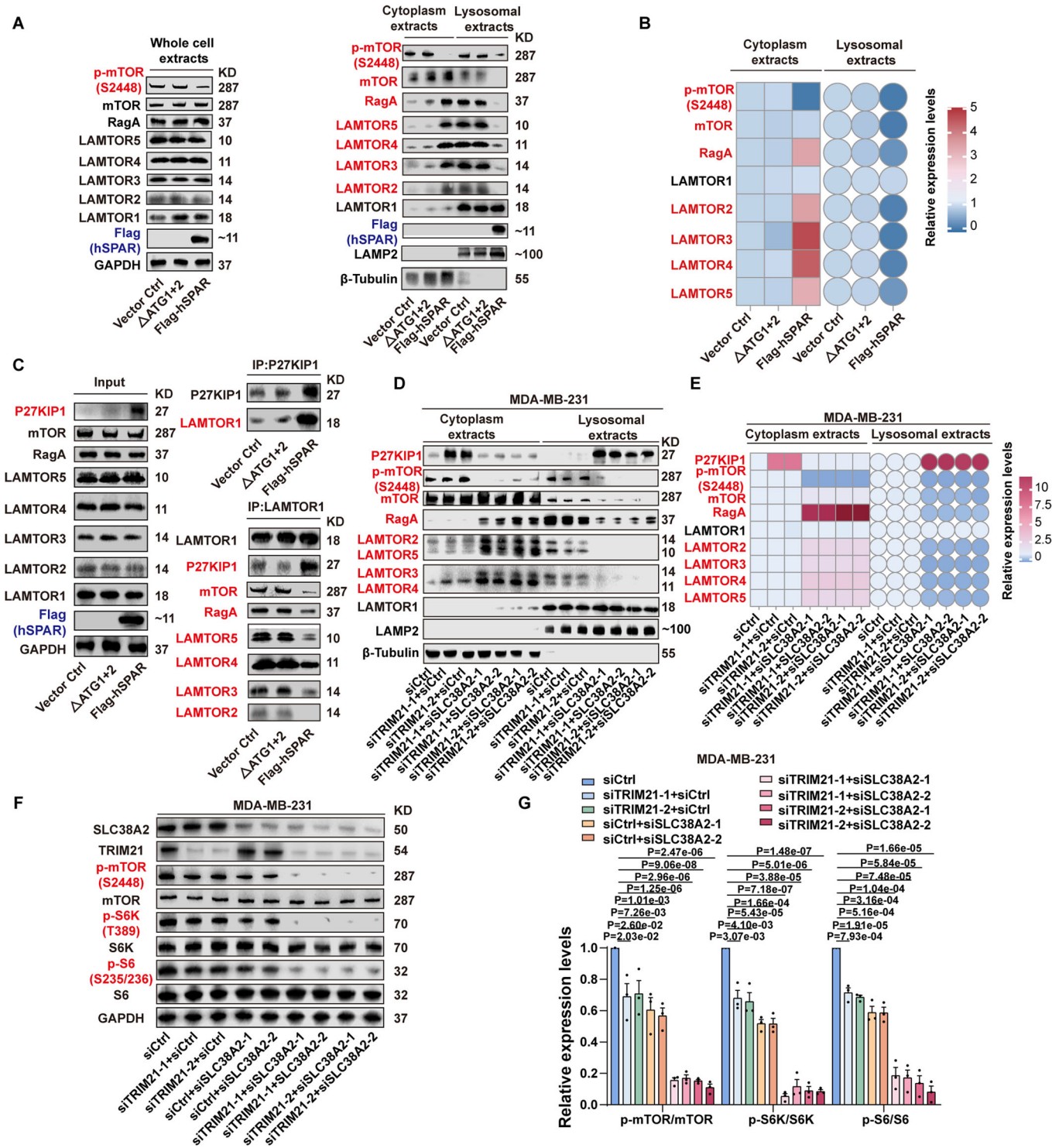

transmembrane domain (TM), and a cytoplasmic C-terminal domain (Fig. EV5A). Considering that the TRIM21-mediated ubiquitination exists in the cytoplasm, the cytoplasmic C-terminal domain of hSPAR might be the functional domain for TRIM21 interaction. To test this hypothesis, we generated an hSPAR's C-terminal construct (Flag-hSPAR-C). Co-IP analysis revealed that Flag-hSPAR-C was sufficient to interact with TRIM21, and to

reduce the interaction between TRIM21 and P27KIP1 in MDA-MB-231 cells (Fig. 7A).

To further explore the necessity of hSPAR-C, we generated three Flag-tagged hSPAR domain deletion variants based on ORF1 (Fig. EV5B) and expressed them or full-length Flag-hSPAR in MDA-MB-231 cells. Supporting our subcellular fractionation results shown in Fig. 5, full-length Flag-hSPAR, instead of control

**Figure 6. hSPAR guarantees the coordination of increased stability and glutamine-associated lysosomal-localization of P27KIP1 to inactivate mTOR signaling specifically in breast cancer cells.**

See also Figs. EV3 and 4. (A) Immunoblotting of whole-cell extracts (left panel), cytoplasmic (lysosome components removed) and lysosomal extracts (right panel) prepared from MDA-MB-231 cells transfected with the indicated constructs against p-mTOR, mTOR, RagA, LAMTOR1-5, Flag, GAPDH, LAMP2 (lysosomal marker), and β-Tubulin (cytoplasmic marker) ($n = 3$ independent biological samples). (B) Heatmap shows the relative expression analysis of the displayed experiment for each protein to β-Tubulin (cytoplasmic fractions) or LAMP2 (lysosomal fractions) from panel (A) ($n = 3$ independent biological samples). (C) Interaction of LAMTOR1 with P27KIP1, LAMTOR2-5, RagA and mTOR, or P27KIP1 with LAMTOR1 detected by Co-IP and immunoblotting from MDA-MB-231 cells transfected with the indicated constructs. Left: immunoblotting of inputs. Upper right: immunoblotting using antibodies against LAMTOR1 and P27KIP1 after IP of P27KIP1. Lower right: immunoblotting using antibodies against P27KIP1, LAMTOR1-5, RagA and mTOR after IP of LAMTOR1 ($n = 3$ independent biological samples). (D) Immunoblotting of cytoplasmic (lysosome components removed) and lysosomal extracts (right panel) prepared from MDA-MB-231 cells transfected with siCtrl, indicated siTRIM21s, or siTRIM21s together with siSLC38A2s against P27KIP1, TRIM21, SLC38A2, p-mTOR, mTOR, RagA, LAMTOR1-5, LAMP2 (lysosomal marker) and β-Tubulin (cytoplasmic marker) ($n = 3$ independent biological samples). (E) Heatmap shows the relative expression analysis of the displayed experiment for each protein to β-Tubulin (cytoplasmic fractions) or LAMP2 (lysosomal fractions) from panel (D) ($n = 3$ independent biological samples). (F) Immunoblotting against TRIM21, SLC38A2, p-mTOR, mTOR, p-S6K, S6K, p-S6, S6 and GAPDH for extracts from MDA-MB-231 cells transfected with siCtrl, indicated siTRIM21s or siSLC38A2s, or siTRIM21s together with siSLC38A2s ($n = 3$ independent biological samples). (G) Quantified relative levels of p-mTOR/mTOR, p-S6K/S6K, and p-S6/S6 from panel (F) ($n = 3$ independent biological samples). Data are presented as the mean ± SEM and analyzed using one-way ANOVA with Dunnett' multiple comparisons test. The hSPAR-regulated proteins shown by immunoblotting are marked by red text. Source data are available online for this figure.

or ΔATG1 + 2 plasmid, promoted dramatic lysosomal accumulation of P27KIP1 observed by immunofluorescence (Figs. 7B and EV5C). In ΔC-hSPAR expressing cells, P27KIP1 failed to localize to the lysosomes, whereas cells expressing ΔTM-hSPAR or ΔN-hSPAR retained a similar ability as the full-length hSPAR for P27KIP1's lysosomal-localization (Figs. 7B and EV5C). Next, we found that only ΔC-hSPAR, but not ΔTM-hSPAR or ΔN-hSPAR failed to decrease lysosomal distribution of mTOR (Figs. 7C and EV5D) or inactivate mTOR signaling (Figs. 7D and EV5E). Consistently, we found that Flag-hSPAR-C was sufficient to reduce mTOR activation similar as the full-length hSPAR (Figs. 7E and EV5F). We also synthesized a TAT-fused hSPAR-C peptide (TAT-hSPAR-C). Similar to the Flag-hSPAR-C plasmid, our data showed that giving TAT-hSPAR-C peptide in MDA-MB-231 cells could effectively decrease the expression level of SLC38A2, increase protein level of P27KIP1, and inactivate mTOR signaling (Figs. 7F and EV5G).

To further examine the role of hSPAR's C-terminal domain on cancer cell proliferation, we conducted EdU assays. Our results showed that ΔC-hSPAR, but not ΔN-hSPAR or ΔTM-hSPAR, lost the inhibitory effect of hSPAR on cell proliferation (Figs. 7G and EV5H). Consistently, treatment of TAT-hSPAR-C peptide also dramatically suppressed the MDA-MB-231 cell proliferation (Figs. 7H and EV5I). We next attempted to investigate the in vivo anti-tumor activity of hSPAR-C. Using breast cancer xenograft mouse model, we tail vein injected either TAT or TAT-hSPAR-C peptide. IHC staining with anti-hSPAR antibody showed strong signals of hSPAR in TAT-hSPAR-C-treated samples compared to the background signals of the TAT-treated samples (Fig. EV5J). Our results further demonstrated that TAT-hSPAR-C remarkably suppressed the tumor growth (Fig. 7I–K) with reduced the signal of Ki67 (Fig. EV5K). Moreover, the effects of hSPAR on its downstream targets were all reproduced in the TAT-hSPAR-C-treated xenografts: decreased SLC38A2 and cellular glutamine (Fig. 7M,L), increased P27KIP1 distributed as punctate aggregates similar to lysosomes by IHC staining (Fig. 7N), as well as dramatically reduced mTOR signaling activities (Figs. 7O and EV5L). Collectively, our data clearly demonstrate that TAT-hSPAR-C is necessary and sufficient for hSPAR-mediated cell proliferation inhibition and regulation on TRIM21/glutaimine-P27KIP1-mTOR axis in breast cancer cells.

## Discussion

As summarized in Fig. 7P, in this study, we have identified that hSPAR/hSPAR-C exhibits robust inhibitory effects on mTOR signaling activation through coordinate regulations by promoting the stability of P27KIP1 protein in the cytoplasm and inducing glutamine deprivation-dependent lysosomal-localization of P27KIP1 specifically in cancer cells.

Glutamine, one of the most abundant amino acids in cells, is essential in the synthesis of amino acids, purines, pyrimidines, and fatty acids to maintain the proliferation of both non-cancerous and cancerous cells (DeBerardinis and Cheng, 2010; Fan et al, 2013; Hensley et al, 2013; Ling et al, 2023; van Geldermalsen et al, 2016; Yang et al, 2014). Specifically targeting cancerous glutamine metabolism has become an attractive approach for anti-cancer treatment (Gross et al, 2014; Guo et al, 2023; Hassanein et al, 2015; Korangath et al, 2015; Robinson et al, 2007; Wang et al, 2010; Yang et al, 2017). In the present work, we find that micropeptide hSPAR acts as an intrinsic regulator of glutamine metabolism by repressing the glutamine transporter SLC38A2 and demonstrate that glutamine deprivation is the key to mediating the anti-tumor effect of hSPAR. However, it is worth emphasizing that the effect of hSPAR on mTOR inactivation is about 3–9 times stronger than directly targeting SLC38A2 (Fig. 5L,M). The underlying mechanism lies in that, in addition to blocking the cellular uptake of glutamine, hSPAR can stabilize cytosol P27KIP1 and result in much greater amount of P27KIP1 accumulated to lysosomes than solely knockdown of SLC38A2. Importantly, such effects of hSPAR do not exist in the HEK293T or MCF10A cells. In addition to glutamine, our metabolomics data indicate that overexpression of hSPAR results in alterations of various amino acids in MDA-MB-231 cells (Table EV1), suggesting that hSPAR plays a crucial role in amino acid metabolism processing. The underlying mechanisms await further exploration.

It has been reported that phosphorylation of threonine residues at position 157 and 198 of P27KIP1 can promote its localization in the cytoplasm (Liang et al, 2002; Shin et al, 2002; Viglietto et al, 2002). While phosphorylation of serine residue at position 10 of P27KIP1 can promote its localization in the nucleus (Ishida et al, 2015; Rodier et al, 2001). Our data show that a large amount of P27KIP1 is translocated to the lysosomes in glutamine-deficient

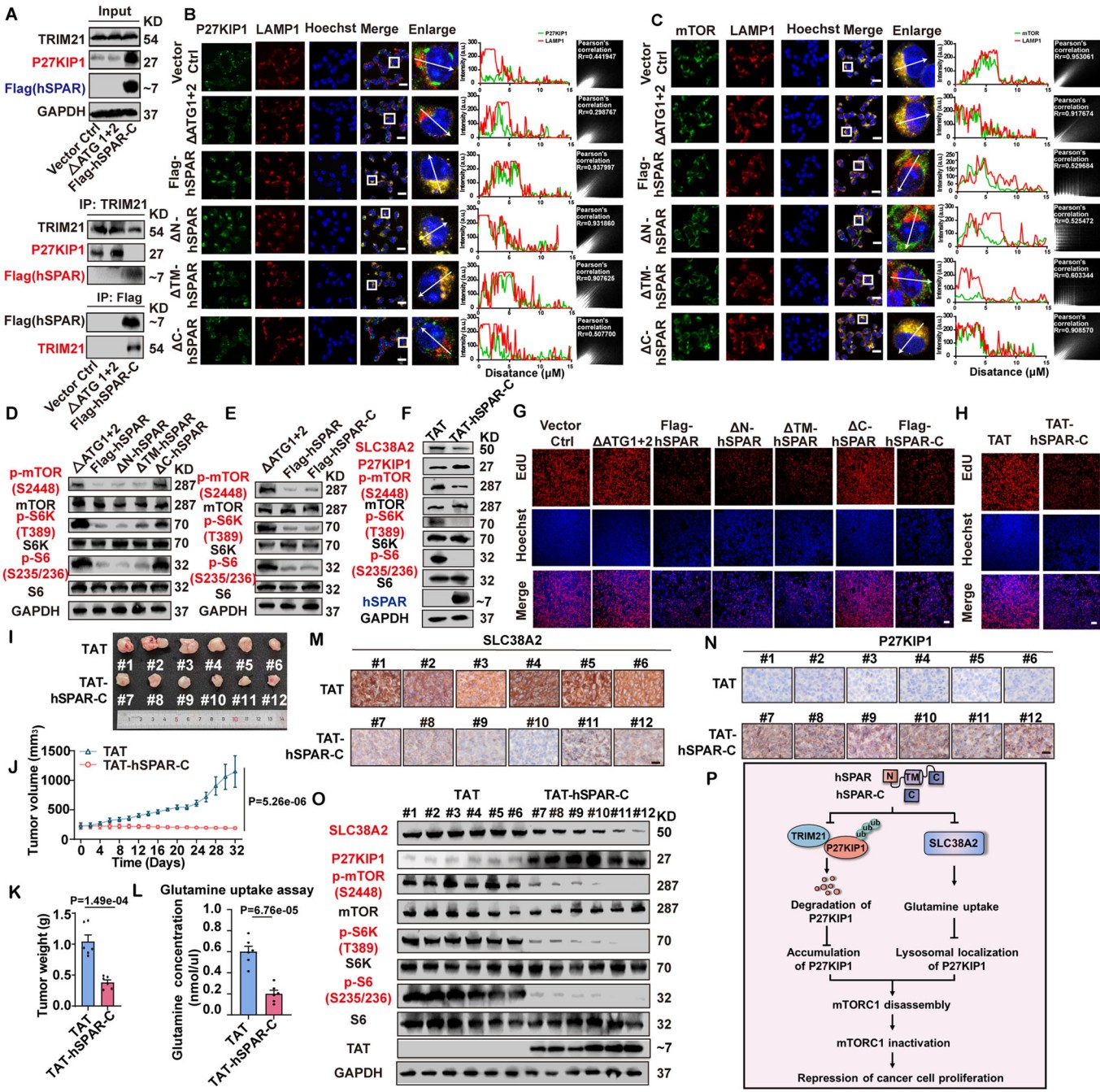

MDA-MB-231 cells. It would be of great interest to explore whether certain posttranslational modifications of P27KIP1 are involved in P27KIP1's lysosomal accumulation in future investigations.

Numerous studies have demonstrated the indispensable roles of amino acids (particularly glutamine, arginine and leucine) in mTOR signaling activation (Bar-Peled and Sabatini, 2014; Mossmann et al, 2018; Sancak et al, 2010; Takahara et al, 2020). Treating cells with amino acid deprivation media disrupts the appropriate assembly of mTORC1 on lysosomes and inactivates mTOR signaling. While supplementing amino acids promotes the

lysosomal mTORC1 assembly and reactivates mTOR signaling. mTORC1 assembly is under precise control. For example, the Ragulator complex, comprised of LAMTOR1-5 subunits, is anchored to lysosomes through LAMTOR1. Upon amino acid stimulation, LAMTOR2-5 components sequentially interact with LAMTOR1 to assemble the Ragulator complex on lysosomes, thereby activating mTOR signaling (Sancak et al, 2010; Saxton and Sabatini, 2017). While V-ATPase complex is well-known for its inhibitory effect to mTORC1's lysosomal recruitment by competitively interacting with Ragulator upon amino acid starvation (Ratto et al, 2022; Zoncu et al, 2011). Recently, the CDK inhibitor

**Figure 7. The cytoplasmic domain of hSPAR is sufficient to trigger TRIM21/glutamine-P27KIP1-mTOR signaling in MDA-MB-231 cells and suppress xenograft tumor growth.**

See also Fig. EV5. (A) Changes of interaction between TRIM21 and P27KIP1, TRIM21, and Flag-hSPAR-C detected by Co-IP and immunoblotting from MDA-MB-231 cells transfected with Vector Ctrl, ΔATG1 + 2 or Flag-hSPAR-C. Upper, immunoblotting of inputs. Middle, immunoblotting using antibodies against TRIM21, P27KIP1 and Flag following IP of TRIM21. Lower, immunoblotting using antibodies against Flag and TRIM21 following IP of Flag ($n = 3$ independent biological samples). (B) Co-immunofluorescence staining of P27KIP1(green) and the lysosomal marker LAMP1 (red) in MDA-MB-231 cells transfected with the indicated constructs. Cells were permeabilized with digitonin to remove the soluble P27KIP1. Nuclei were stained with Hoechst (blue). The graphs display the fluorescence intensity (arbitrary units) of P27KIP1 and LAMP1 over the distance from adjacent image (depicted by the arrows). The graphs display the values of Pearson's correlation Rr of P27KIP1 and LAMP1 with the indicated constructs. Scale bar, 25 μm ($n = 3$ independent biological samples). (C) Co-immunofluorescence staining of mTOR (green) and the lysosomal marker LAMP1 (red) in MDA-MB-231 cells transfected with the indicated constructs. Cells were permeabilized with digitonin to remove the soluble mTOR. Nuclei were stained with Hoechst (blue). The graphs display the fluorescence intensity (arbitrary units) of mTOR and LAMP1 over the distance from adjacent image (depicted by the arrows). The graphs display the values of Pearson's correlation Rr of mTOR and LAMP1 with the indicated constructs. Scale bar, 25 μm ($n = 3$ independent biological samples). (D) Immunoblotting against p-mTOR, mTOR, p-S6K, S6K, p-S6, and S6 in extracts from MDA-MB-231 cells transfected with ΔATG1 + 2, Flag-tagged full-length hSPAR or three indicated hSPAR domain deletion variants. Statistical analysis is shown in Fig. EV5E ($n = 3$ independent biological samples). (E) Immunoblotting against p-mTOR, mTOR, p-S6K, S6K, p-S6, and S6 in extracts from MDA-MB-231 cells transfected with ΔATG1 + 2, Flag-hSPAR or Flag-hSPAR-C (hSPAR's C-terminal domain). Statistical analysis is shown in Fig. EV5F ($n = 3$ independent biological samples). (F) Immunoblotting against p-mTOR, mTOR, p-S6K, S6K, p-S6, S6, P27KIP1, SLC38A2 and TAT in extracts from MDA-MB-231 cells treated with TAT-hSPAR-C (150 nM) at the indicated time. Statistical analysis is shown in Fig. EV5G ($n = 3$ independent biological samples). (G) Representative images of EdU assay detected in MDA-MB-231 cells transfected with Vector Ctrl, ΔATG1 + 2, Flag-hSPAR, ΔN-hSPAR, ΔTM-hSPAR, ΔC-hSPAR, or Flag-hSPAR-C. Scale bar, 50 μm. Statistical analysis is shown in Fig. EV5H ($n = 3$ independent biological samples). (H) Representative images of EdU assay detected in MDA-MB-231 cells treated with TAT or TAT-hSPAR. Scale bar, 50 μm. Statistical analysis is shown in Fig. EV5I ($n = 3$ independent biological samples). (I) Images of tumors from breast cancer xenograft mice with injection of TAT ($n = 6$ independent biological samples) or TAT-hSPAR-C peptide ($n = 6$ independent biological samples). (J) Time-course analysis of tumor volume in the xenografts from panel (I) ($n = 12$ independent biological samples). Data are presented as the mean ± SEM and analyzed using two-tailed Student's $t$ test with Welch's correction. (K) Tumor weight in the xenografts from panel (I) ($n = 12$ independent biological samples). Data are presented as the mean ± SEM and analyzed using two-tailed Student's $t$ test with Welch's correction. (L) Levels of glutamine in the xenografts from panel (I) ($n = 12$ independent biological samples). Data are presented as the mean ± SEM and analyzed using two-tailed Student's $t$ test with Welch's correction. (M) Immunohistochemistry of SLC38A2 detected by immunohistochemistry with the anti-SLC38A2 antibody in the xenografts from panel (I) ($n = 12$ independent biological samples). Scale bar, 20 μm. (N) Immunohistochemistry of P27KIP1 detected by immunohistochemistry with the anti-P27KIP1 antibody in the xenografts from panel (I) ($n = 12$ independent biological samples). Scale bar, 20 μm. (O) Immunoblotting against SLC38A2, P27KIP1, p-mTOR, mTOR, p-S6K, S6K, p-S6, S6, TAT, and GAPDH in the xenografts from panel (I). Statistical analysis is shown in Fig. EV5L ($n = 12$ independent biological samples). (P) Working model: in cancer cells or xenograft tumors, hSPAR or hSPAR-C interacts with E3 ligase TRIM21 and disrupts TRIM21-P27KIP1 interaction, resulting escape of P27KIP1 from ubiquitin-proteasome degradation and increase of P27KIP1 expression level in cytosol. Coordinately, acting as an inhibitor for glutamine uptake, hSPAR results in glutamine deprivation by suppressing SLC38A2, and promotes P27KIP1 lysosomal-localization. Lysosome-localized P27KIP1 competitively interacts with LAMTOR1 to disassemble mTORC1 complex and inhibit cancer cell proliferation. The hSPAR-regulated proteins shown by immunoblotting are marked by red text. Source data are available online for this figure.

---

p27Kip1 has been found as a lysosomal-localized protein and noncanonical inhibitor to mTORC1 assembly when the MEFs were treated upon prolonged amino acid starvation condition (Nowosad et al, 2020). Previously, Matsumoto et al have reported that SPAR interacts with V-ATPase and facilitates V-ATPase's inhibitory role on mTORC1 assembly in HEK293T cells upon amino acid re-stimulation (Matsumoto et al, 2017). In our work, hSPAR-inhibited effect on mTOR activity occurs in breast cancer cells under complete culturing media and mice with regular diet. Under these conditions, hSPAR shows no interaction with V-ATPase or other reported lysosomal proteins. Instead, hSPAR fulfills its mTOR inhibitory effect through acting as a cancer-specific metabolism regulator and P27KIP1 stabilizer as we discussed above. Therefore, we have identified a new regulatory mechanism of hSPAR in breast cancer cells, distinct from HEK293T and MCF10A cells. The underlying mechanisms for such differential in different types of cells await further exploration.

P27KIP1 has been previously reported as a substrate of TRIM21 in cervical cancer HeLa cells and glioblastoma T98G cells (Sabile et al, 2006). In the present work, we confirm the role of TRIM21 as the E3 ligase to P27KIP1 in breast cancer cells. Intriguingly, although directly knocking down of TRIM21 increases as comparable extent of P27KIP1 protein level as hSPAR overexpression, the accumulated P27KIP1 stays in cytosol instead and has slight impact on mTOR signaling activation when solely knocking down

TRIM21. These findings provide strong evidence for the biological significance of the lysosomal-localized P27KIP1 on mTOR inhibition. Paradoxically, co-expression of TRIM21 interrupted the Flag-hSPAR-mediated inhibitory effects on mTOR signaling and cancer proliferation, suggesting the indispensable role of TRIM21 in hSPAR's anti-tumor functions. The underlying mechanism here lies in that, without the large amount of P27KIP1 accumulated through blocking TRIM21, only small amount of P27KIP1 would be loaded to lysosomes and has limited influence on mTOR signaling, as observed in single-knockdown of SLC38A2. In addition, cancer-related micropeptides have been shown as promising drug-developing targets (Bakhti and Latifi-Navid, 2022). In this work, we further demonstrate the 48aa-long hSPAR-C as the functional domain for hSPAR's inhibitory effect on glutamine level control, mTOR signaling and cancer proliferation on both cultured cells and xenografic mice. Identification of its functional domain further makes hSPAR as an attractive target for therapeutical development.

In summary, we have discovered micropeptide hSPAR as an intrinsic metabolism regulator that integrates three critical pathways—cellular glutamine-level control, P27KIP1 protein stability, and noncanonical lysosomal-translocation of P27KIP1—to generate a robust inhibitory effect on mTOR signaling and cancer cell proliferation. Our identifications of hSPAR-TRIM21/glutamine-P27KIP1-mTOR cascade and its specificity in cancer cells expand the current understanding on P27KIP1-mTOR axis and cancer cell metabolism.

# Methods

### Reagents and tools table

| Reagent/resource | Reference or source | Identifier or catalog number |
| --- | --- | --- |
| **Experimental models** | | |
| NOD.Cg-*Prkdc^scid^Il2rg^em1cya^*/Cya (*Mus musculus*) | Cyagen, China | Cat #C001316 |
| BALB/cAnSmoc-*Foxn1^nu^*/Smoc (*Mus musculus*) | Nanjing Biomedical Research Institute of Nanjing University, China | Cat #SM-014 |
| HEK293T (*Homo sapiens*) | Procell, China | Cat #CL-0005 |
| MCF10A (*Homo sapiens*) | Procell, China | Cat #CL-0525 |
| MDA-MB-231 (*Homo sapiens*) | Procell, China | Cat #CL-0150 |
| MDA-MB-468 (*Homo sapiens*) | Procell, China | Cat #CL-0290 |
| MDA-MB-453 (*Homo sapiens*) | Procell, China | Cat #CL-0152 |
| MCF7 (*Homo sapiens*) | Procell, China | Cat #CL-0149 |
| **Recombinant DNA** | | |
| pcDNA3.1 (+) | Invitrogen, USA | Cat #V790-20 |
| pEGFP-C1 | BD Biosciences, USA | Cat #6084-1 |
| **Antibodies** | | |
| Rabbit anti-ATP1V1A | Proteintech, USA | Cat #14418-1-AP |
| Rabbit anti-phospho-AKT(Ser473) | CST, USA | Cat #4060 |
| Rabbit anti-AKT | CST, USA | Cat #9272 |
| Rabbit anti-Beta Tubulin | Proteintech, USA | Cat #10068-1-AP |
| Rabbit anti-FBL | Proteintech, USA | Cat #16021-1-AP |
| Mouse anti-GAPDH | Proteintech, USA | Cat #60004-1-Ig |
| Rabbit anti-HBXIP (LAMTOR3) | Proteintech, USA | Cat #14492-1-AP |
| Rabbit anti-hSPAR | HUABIO, China | This study |
| Rabbit anti-LAMP1 | CST, USA | Cat #15665 |
| Rabbit anti-LAMP2 | CST, USA | Cat #49067 |
| Rabbit anti-LAMTOR1/C11orf59 | CST, USA | Cat #8975 |
| Rabbit anti-LAMTOR2/ROBLD3 | CST, USA | Cat #8145 |
| Rabbit anti-LAMTOR4/C7orf59 | CST, USA | Cat #13140 |
| Rabbit anti-MAPKSP1 | Proteintech, USA | Cat #11937-1-AP |
| Rabbit anti-mTOR | CST, USA | Cat #2983 |
| Rabbit anti-phospho-mTOR(Ser2448) | CST, USA | Cat #5536 |

| Reagent/resource | Reference or source | Identifier or catalog number |
| --- | --- | --- |
| Rabbit anti-p70 S6 Kinase (49D7) | CST, USA | Cat #2708 |
| Rabbit anti-phospho-P27KIP1 | Abcam, USA | Cat #ab75908 |
| Rabbit anti-phospho-p70 S6 Kinase (Thr389) | CST, USA | Cat #9234 |
| Rabbit anti-S6 ribosomal Protein | CST, USA | Cat #2217 |
| Rabbit anti-phospho-S6 Ribosomal Protein (Ser235/236) | CST, USA | Cat #2211 |
| Rabbit anti-RagA | CST, USA | Cat #4357 |
| Rabbit anti-SNAT2 (SLC38A2) | ImmunoWay, USA | Cat #YT4354 |
| Rabbit anti-SLC7A1 | Proteintech, USA | Cat #14195-1-AP |
| Rabbit anti-SLC7A5 | Proteintech, USA | Cat #28670-1-AP |
| Rabbit anti-Ubiquitin | Abcam, UK | Cat #ab134953 |
| Goat anti-Rabbit IgY_H&L (Alexa Fluor Plus 488) | Invitrogen, USA | Cat #A32731 |
| Goat anti-Mouse IgG_H&L (Alexa Fluor Plus 555) | Invitrogen, USA | Cat #A32727 |
| Mouse anti-DDDDK tag | Abcam, UK | Cat #ab125243 |
| Mouse anti-P27KIP1 | CST, USA | Cat #3698 |
| Mouse anti-TRIM21 | Proteintech, USA | Cat #67136-1-Ig |
| Rabbit anti-DDDDK tag | Abcam, UK | Cat #ab205606 |
| Rabbit anti-P27KIP1 | Proteintech, USA | Cat #25614-1-AP |
| Rabbit anti-TRIM21/SS | Abcam, UK | Cat #91423 |
| **Oligonucleotides and other sequence-based reagents** | | |
| siRNAs | This study | Table EV2 |
| PCR primers | This study | Table EV3 |
| **Chemicals, enzymes, and other reagents** | | |
| Acryl/Bis 30% Solution (29:1) | BBI, China | Cat #B546017 |
| BSA | BioFroxx, Germany | Cat #4240GR500 |
| Certified Fetal Bovine Serum (FBS) | Biological Industries, Israel | Cat #04-011-1A |
| DEPC Treated Water | BBI, China | Cat #B501005 |
| Digition | Sigma-Aldrich, USA | Cat #D141 |
| Dimethyl sulfoxide (DMSO) | Sigma-Aldrich, USA | Cat #D8418 |
| Dulbecco's Modified Eagle Medium (DMEM) | Biological Industries, Israel | Cat #C3113-0500 |
| DTT | Sangon, China | Cat #A300862 |

| Reagent/resource | Reference or source | Identifier or catalog number |
| --- | --- | --- |
| EDTA | BBI, China | Cat #EB0185 |
| EGTA | BBI, China | Cat #A600077 |
| Ethanol absolute | BBI, China | Cat #A500737 |
| Glycerol | Sangon, China | Cat #A501745 |
| Glycine | BBI, China | Cat #A610235 |
| G418 | Wisent, China | Cat #450-130-ZL |
| Hoechst 33342 | Invitrogen, USA | Cat #H3570 |
| Immobilon-P PVDF Membrane | Millipore, USA | Cat #IPVH00010 |
| IP Lysis Buffer | Thermo Scientific, USA | Cat #87788 |
| Lipofectamine 3000 Transfection Reagent | Invitrogen, USA | Cat #L3000015 |
| LysoTracker™ Deep Red | Invitrogen, USA | Cat #L12492 |
| MCF7 Cell Medium | Procell, China | Cat #CM-0149 |
| MCF10A Cell Medium | Procell, China | Cat #CM-0525 |
| Methanol | BBI, China | Cat #A601617 |
| MG132 | MCE, USA | Cat #HY-13259 |
| MgCl2 | BBI, China | Cat #A100288 |
| NaCl | Sangon, China | Cat #A501218 |
| Paraformaldehyde | BBI, China | Cat #A500684 |
| Phenylmethyl sulfonyl fluoride (PMSF) | BBI, China | Cat #A610425 |
| PIPES | BBI, China | Cat #A600719 |
| Poly-L-lysine solution | Sigma-Aldrich, USA | Cat #P4707 |
| Potassium dihydrogen phosphate | Sangon, China | Cat #A501211 |
| Potassium chloride | Sangon, China | Cat #A610440 |
| Proteinase Inhibitor Cocktail | Sigma-Aldrich, USA | Cat #P8340 |
| Protein A/G Agarose beads | Santa Cruz, USA | Cat #sz-2003 |
| protein A/G Magnetic Beads | BiolinkedIn, China | Cat #L-1004 |
| SDS | Vetec, USA | Cat #V900859 |
| Skim milk powder (Blotting Grade) | Beyotime, China | Cat #P0216 |
| Sodium phosphate, dibasic, dodecahydrate | Sangon,China | Cat #A501725 |
| Sodium citrate tribasic dihydrate | BBI, China | Cat #A610035 |
| Tali™ Cell Cycle Solution | Invitrogen, USA | Cat #A10798 |

| Reagent/resource | Reference or source | Identifier or catalog number |
| --- | --- | --- |
| Triton X-100 | BBI, China | Cat #TB0198 |
| Tris (hydroxymethyl) aminomethane | BBI, China | Cat #A600194 |
| Tween-20 | BBI, China | Cat #A600560 |
| TRIzol Reagent | Invitrogen, USA | Cat #15596018 |
| Tali™ Cell Cycle Solution | Invitrogen, USA | Cat #A10798 |
| Trypsin EDTA Solution A (0.25%), EDTA (0.02%) | Biological Industries, Israel | Cat #03-050-1A |
| TAT peptide | QYAOBIO, China | This study |
| TAT-hSPAR C peptide | QYAOBIO, China | This study |
| **Software** | | |
| ImageJ software | National Institutes of Health, USA | N/A |
| GraphPad Prism 8.0 | GRAPHPAD SOFTWARE, LLC,USA | N/A |
| Image Pro Plus 6.0 | MEDIA CYBERNETICS, USA | N/A |
| FCS4 Express Cytometry | De Novo Software, USA | N/A |
| **Other** | | |
| Cell-Light EdU Apollo567 In Vitro Kit (100 T) | RIBOBIO, China | Cat #C10310-1 |
| Glutamine Assay Kit (Colorimetric) | Abcam, UK | Cat #ab197011 |
| HiPure Gel Pure DNA Mini Kit | Magen, China | Cat #D2111-03 |
| HiScriptII 1st Strand cDNA Synthesis Kit | Vazyme, China | Cat #R211-01 |
| MaxPure Plasmid EF Mini Kit | Magen, China | Cat #P1220-03 |
| Minute™ Lysosome Isolation Kit for Mammalian Cells/Tissues | Invent, USA | Cat #LY-034 |
| Mut Express MultiS Fast Mutagenesis Kit V2 | Vazyme, China | Cat #C215-01 |
| Mouse and Rabbit Specific HRP/DAB (ABC) Detection IHC Kit | Abcam, UK | Cat #ab64264 |
| Nuc-Cyto-Mem Preparation Kit | Applygen, China | Cat #P1201 |
| Rapid Silver Staining Kit | Beyotime, China | Cat #P0017S |

## Cell culture

The HEK293T, MDA-MB-231, MDA-MB-468, MDA-MB-453 and MCF7 cells were obtained from Procell (Wuhan, China). The HEK293T, MDA-MB-231, MDA-MB-468 and MDA-MB-453 cells were cultured in Dulbecco's modified Eagle's medium (DMEM) (vivacell, Shanghai, China), containing 10% fetal bovine serum (vivacell, Shanghai, China) and 1% penicillin–streptomycin (WISENTInc., CA). The MCF10A cells were cultured in DMEM/F12 medium (Procell, China), containing 5% horse serum, 20 ng/mL EGF, 0.5 μg/mL hydrocortisone, 10 μg/mL

insulin, 1% non-essential amino acids (NEAA) and 1% penicillin–streptomycin. MCF7 cells were cultured in Minimum Essential Medium medium (Procell, China), containing 0.01 mg/mL insulin, 10% fetal bovine serum and 1% penicillin–streptomycin. BT474 cells were cultured in RPMI-1640 medium (Procell, China), containing 10 μg/mL insulin, 2 mM L-glutamine, 20% fetal bovine serum and 1% penicillin–streptomycin. For glutamine starvation, the MDA-MB-231 cells were cultured in glutamine-deficient medium (Gibco, USA), containing 10% fetal bovine serum (vivacell, Shanghai, China) and 1% penicillin–streptomycin (WISENTInc., CA). All cells were cultured under humidified atmosphere of 5% $CO_2$ at 37 °C.

## Clinical samples and TCGA data analysis

The IHC specimens and frozen patient tissues (including the breast cancer tumoral tissues and non-tumoral tissues) were collected from the Division of Life Sciences and Medicine, University of Science and Technology of China (Figs. 1E and 2H,I, Approval number 2021KY286) and Tangshan Gongren Hospital (Fig. EV1B, Approval number 2023YL12). All participants signed and informed consent prior to sample collection, the experiments conformed to the principles set out in the WMA Declaration of Helsinki and the Department of Health and Human Services Belmont Report. The RNA expression data of different cancer tissues and their adjacent non-cancerous tissues were downloaded from TCGA database (https://portal.gdc.cancer.gov/), and all statistical graphs were generated using the GEPIA2 website (http://gepia2.cancer-pku.cn).

## Plasmid construction, RNA interference, and cell transfection

To generate Flag fusion protein constructs with the hSPAR ORF (Flag-hSPAR) or the cytoplasmic C-terminal domain of hSPAR (hSPAR-C), the hSPAR ORF sequence or hSPAR-C sequence was amplified using RT-PCR and cloned into the pcDNA3.1(+) vector. Mutation constructs (ΔATG1, ΔATG2, ΔATG1 + 2) were generated by using Mut Express MultiS Fast Mutagenesis Kit V2 (Vazyme, China) to mutate the Flag-hSPAR's ATG to ATT. The Flag-tagged hSPAR truncated constructs (ΔN-hSPAR, ΔTM-hSPAR, ΔC-hSPAR) were generated by General Biol (China). GFP-hSPAR was constructed using the pEGFP-C1 vector.

SiRNAs were designed and synthesized from RiboBio (China). The siRNA and PCR primers used in this study are listed in Tables EV2 and EV3.

## RNA isolation and quantitative PCR with reverse transcription

RNA was extracted from cell samples by using Trizol Reagent (Ambion, USA) according to the manufacturer's instructions. One Drop® OD-1000 Spectrophotometer (Nanjing Wuyi Corporation, China) was used to measure RNA concentration and purity. The RNA was reverse-transcribed using HiScript II One Step RT-PCR Kit (Vazyme, China) according to the manufacturer's instructions. The following quantitative PCR (qPCR) analysis was performed using SYBR® Green Master Mix (Vazyme, China) on Light Cycle® 96 (Roche, USA). GAPDH RNA was used as internal control for all the RT-qPCRs. The primers for the real-time PCR are listed in

Table EV3. Fold changes were determined using the relative quantification $2^{-\Delta\Delta CT}$ method.

## Immunoblotting

The 3×Sample Buffer (1 M Tris-HCl pH 7.4, 6% SDS, 0.03% bromophenol blue, 34.5% glycerol) was diluted to the 1.5×sample buffer. Cells were lysed in 1.5×sample buffer for 30 min on ice and boiled for 10 min at 100 °C. After quantification, whole-cell lysates or cell fractions were separated by SDS-PAGE under denaturing conditions and transferred to PVDF membranes (Millipore, USA). The membranes were blocked in 5% BSA (Sangon, China) for 1 h at room temperature, and then incubated at 4 °C overnight with primary antibody GAPDH (Proteintech, USA, 60004-1-Ig), hSPAR (HuaBio, China), Flag (Abcam, UK, ab205606), β-Tubulin (Proteintech, USA, 10068-1-AP), FIBRILLARIN (Proteintech, USA, 16021-1-AP), ATPV1A (Proteintech, USA, 14418-1-AP), LAMP2 (CST, USA, 49067), TRIM21 (Proteintech, USA, 67136-1-Ig), P27KIP1 (Proteintech, USA, 25614-1-AP), phospho-P27KIP1 (Abcam, USA, ab75908), SKP2 (Proteintech, USA, 15010-1-AP), SLC7A1 (Proteintech, USA, 14195-1-AP), SLC7A5 (Proteintech, USA, 28670-1-AP), phospho-mTOR (CST, USA, 5536), mTOR (CST, USA, 2983), phospho-S6K (CST, USA, 9234), S6K (CST, USA, 2708), phospho-S6 (CST, USA, 2211), S6 (CST, USA, 2217), phospho-AKT (CST, USA, 4060), AKT (CST, USA, 9272), Ubiquitin (Abcam, UK, ab134953), SLC38A2 (ImmunoWay, USA, YT4354), LAMTOR1(CST, USA, 8975), LAMTOR2 (CST, USA, 8145), LAMTOR3 (Proteintech, USA, 14492-1-AP), LAM-TOR4 (CST, USA, 13140), LAMTOR5 (Proteintech, USA,11937-1-AP), RagA (CST, USA, 4357S) and TAT (Abcam, UK, ab42359). The PVDF membranes were then incubated with secondary antibodies conjugated with horseradish peroxidase (Proteintech, USA, SA00001-1 or SA00001-2). Immunoreactive proteins were visualized using the SuperSignal® West Femto Maximum Sensitivity Substrate (Thermo Scientific, USA) on Tanon 5200Muti chemiluminescence gel imaging system (Tanon, China). For breast cancer patient tissue, every 100 mg of frozen tissues was added 1 mL of pre-cooled RIPA lysis buffer and lysed on ice by glass homogenizer, then the homogenate was then boiled for 10 min at 100 °C. For tumor tissues of breast cancer xenograft mice, the frozen tumors were ground into powder and then lysed in RIPA lysis buffer for 1 h at 4 °C. The protein extracts were collected and boiled for 10 min at 100 °C. After quantification, whole tumor lysates were continued for immunoblotting as described above.

## Animal experiments

All animal studies were conducted with the approval from the Animal Research Ethics Committee of the University of Science and Technology of China (Approval number USTCACUC-1801020). Female BALB/C nude mice at the age of 5–6 weeks (Nanjing Biomedical Research Institute of Nanjing University, China) were used for xenograft mouse model construction. Briefly, differently treated tumor cells were trypsinized and harvested, then $1 \times 10^8$ MDA-MB-231 cells in serum-free medium containing 20% matrigel with a total volume of 0.2 ml were injected subcutaneously to the inguinal of the mice. Four weeks later, the mice were sacrificed and noticeable tumors were immediately excised. The

tumor volumes were measured with a caliper and calculated using the following formula $0.5 \times \text{length} \times \text{width}^2$.

For efficacy of chemosynthetic hSPAR's C-terminal domain studies, TAT peptides and TAT-hSPAR-C peptides were synthetized by Fmoc solid-phase peptide synthesis (QYAOBIO, China), and MDA-MB-231 cells were subcutaneously injected into the C-NKG mice generated by (Cyagen, China). When the volume of xenograft reached 100 mm³, mice were randomly assigned into two groups and received intravenous (IV) injections with 10 mg/kg dosage of TAT-hSPAR-C or TAT peptides every 2 days. The length and width of the mice tumors were measured every 2 days, and the tumor volume was calculated using the above formula. After 32 days, the mice were sacrificed, the tumors were removed, and the tumor weight was weighed.

All of the aforementioned animal experiments were conducted by using single blinding trial.

## Transfection and stable cell line establishment

All the transfections were performed using Lipofectamine 3000 reagent (Invitrogen, USA) according to the manufacturer's instructions. Stable cell lines were constructed after transfection with the pcDNA3.1-Flag-hSPAR plasmid and grew in the presence of 400 μg/ml G418 (WISENT Inc., CA). Samples with 2–10 times overexpression rate were selected for further tests. The PCR primers are listed in Table EV3.

## Cell proliferation assay

Cell proliferation assay was performed using Cell-Light EdU Apollo567 In Vitro Kit (100 T) (RiboBio, China). Briefly, cells were seeded in a 96-well plate with 100 ml of complete medium per well. Edu reagent was added into each well 2 h prior to measurement according to the manufacturer's instructions.

## Cell cycle analysis

For cell cycle analysis, cells were synchronized to G0 phase by treating with serum-free medium for 24 h, and subsequently cells were cultured in complete medium containing 10% fetal bovine serum for 24 h before analysis. Cell samples were then harvested and fixed with ice-cold 70% ethanol in distilled water overnight. Then, cells were incubated with Tali™ Cell Cycle Kit containing propidium iodide (Invitrogen, USA) in the dark for 30 min at room temperature. All the samples for cell cycle analysis were performed on a Cellometer Vision Image Cytometer (Nexcelom Bioscience, USA), and Data were analyzed by FCS4 Express Cytometry (De Novo Software, USA).

## Subcellular fractionation

The nucleocytoplasmic fractionation and lysosome isolation were conducted with Nuc-Cyto-Mem Preparation Kit (Applygen, China) and Minute™ Lysosome Isolation Kit (Invent, USA) according to the manufacturer's instructions.

## Lysosome labeling in living cell

For lysosome labeling, the 1 mM Lyso-Tracker probe (Invitrogen, USA) stock solution was diluted to the 75 nM in complete medium.

Cells were incubated in the probe-containing medium for 2 h before living cell imaging or immunofluorescence.

## Immunofluorescence

Cells were seeded on coverslips and cultured to 80–90% confluency. The cells were rinsed with PBS and fixed with 2% PFA in PBS for 20 min at 37 °C, followed by 100% methanol for 5 min at −20 °C. For immunostaining, the cells were permeabilized for 3 min with PBS containing 0.2% Triton X-100, rinsed three times for 5 min in PBS and incubated for 20 min in blocking solution (PBS, 3% BSA, 0.05% Tween-20), followed by the desired primary antibodies diluted in blocking solution for 1 h at 37 °C. After three washes of 5 min in PBS, the cells were incubated for 30 min at 37 °C with fluorescent secondary antibodies, conjugated to Alexa Fluor Plus 488 (Invitrogen, USA, A32731) and/or 555 (Invitrogen, USA, A32727). For double immunofluorescence, after three washes of 5 min in PBS, cells were incubated in the second primary antibody diluted in blocking solution for 1 h at 37 °C. After three washes of 5 min in PBS, the cells were incubated for 30 min at 37 °C with the other fluorescent secondary antibody. Finally, the cell nuclei were stained with Hochst33342 (Thermo, USA). Images of Figs. 2E, 5G, and 7B,C were taken by confocal fluorescence microscope (DM6B, Leica, Germany). Images of Figs. 2D and EV3B were taken by confocal fluorescence microscope (SpinSR, Olympus, Japan). Images of Figs. EV2A and EV5C,D were taken by confocal fluorescence microscope (FV1200MPE-share, Olympus, Japan). The co-localization images were analyzed by image J and Image Pro Plus 6.0 software. The antibodies used for immunofluorescence include: hSPAR (HuaBio, China), Flag (Abcam, UK, ab205606), P27KIP1 (Proteintech, USA, 25614-1-AP), mTOR (CST, USA, 2983) and LAMP1 (CST, USA, 15665).

As previously described (Nowosad et al, 2020), in order to remove the soluble P27KIP1or mTOR, the cells were treated as previously reported. Briefly, cells were permeabilized for 3 min with 40 μg/ml digitonin (Sigma-Aldrich, USA, D141) in PHEM (60 mM PIPES, 25 mM HEPES pH 6.9, 5 mM EGTA and 1 mM MgCl₂) for 3 min at room temperature. The cells were then rinsed with PBS, fixed in 2% PFA for 20 min at room temperature, and continued for immunostaining as described above.

## Co-immunoprecipitation (Co-IP)

Cells were harvested and lysed in IP lysis buffer (20 mM Tris-HCl pH 8.0, 137 mM NaCl, 1% Nonidet P-40 and 2 mM EDTA) supplemented with protease inhibitor cocktail (Sigma-Aldrich, USA, P8340) and PMSF for 1 h at 4 °C. The supernatants were collected by centrifugation at 12,000 rpm for 20 min at 4 °C and pre-cleared with 50 μl Protein A/G Agarose beads at 4 °C for 1 h on a rotator. In all, 10% of the pre-cleared supernatants was saved as input and the rest was subjected for incubation with non-specific control IgG (Abcam, UK, ab172730) or the antibodies used for IP: TRIM21 (Proteintech, USA, 67136-1-Ig), Flag (Abcam, UK, ab205606), P27KIP1 (Proteintech, USA, 25614-1-AP), HA (Abcam, UK, ab18181), or LAMTOR1 (CST, USA, 8975) overnight at 4 °C. The next day, protein A/G Agarose beads were added to the reactions for 4 h at room temperature. The protein-antibody-beads complex was then conducted to IP Wash Buffer (10 mM Tris pH7.4, 1 mM EDTA, 1 mM EGTA pH 8.0, 150 mM NaCl, 1%

TritonX-100) containing PMSF and Proteinase Inhibitor Cocktail. Next, the resulted complex was dissolved in the 3×sample buffer and subjected to SDS-PAGE for immunoblotting.

## Immunohistochemistry (IHC)

Sections containing human breast cancer tumoral and non-tumoral tissues, or tumor tissues of breast cancer xenograft mice were deparaffinized, subjected to target/antigen retrieval boiling for 3 min in antigen retrieval buffer (10 mM sodium citrate, 0.05% Tween-20, pH 6.0), cooled to room temperature, rinsed with wash buffer and blocked for 10 min. Sections were then incubated with 1:500 antibody to hSPAR (HuaBio, China), P27KIP1 (Proteintech, USA, 25614-1-AP), SLC38A2 (ImmunoWay, USA, YT4354) or Ki67 (Proteintech, USA, 28074-1-AP) for 1 h. HRP/DAB detection was then performed using the Mouse and Rabbit Specific HRP/DAB (ABC) Detection IHC kit (Abcam, USA) according to the manufacturer's instructions. Sections were counterstained with hematoxylin, washed, and sealed with neutral resin.

## Glutamine uptake assay

Cellular glutamine level was measured using Glutamine Assay kit (Colorimetric) (Abcam, USA). Briefly, for cell samples, cells were collected and suspended at a concentration of 1 million/ml in PBS. After repeated freezing and thawing, the cells were under centrifugation at 4 °C for 10,000 × g, 5 min. The supernatants were then collected and conducted for the test of glutamine concentration according to the manufacturer's instructions on Multi-Mode Microplate Reader (CLARIOstar, BMG LABTECH, GERMANY). For tissue samples, 10–20 mg of tissues were homogenized on ice using 10× (v/w) Assay Buffer XXIX/Hydrolysis Buffer. The homogenate was then centrifuged at 4 °C for 10,000 × g, 10 min. The supernatants were then collected and conducted for the test of glutamine concentration as described above.

## Metabolomics

MDA-MB-231 cells or HEK293T cells were transfected with ΔATG1 + 2 or Flag-hSPAR for 48 h. The metabolites were extracted for metabolite quantification by LC/MS analysis at PTMBio (Hangzhou, China).

## Statistical analysis

All data are presented as the mean ± SEM. Statistical analyses were performed using the GraphPad Prism Software. The RNA or protein differential expression data were analyzed using two-tailed Student's $t$ test with Welch's correction or one-way ANOVA with Dunnett' multiple comparisons test. $P$ value < 0.05 was considered significant.

## Ethics statement

The IHC specimens and frozen patient tissues (including the breast cancer tumoral tissues and non-tumoral tissues) were collected from the Division of Life Sciences and Medicine, University of Science and Technology of China (Figs. 1E and 2H,I, Approval number 2021KY286) and Tangshan Gongren Hospital (Fig. EV1B,

Approval number 2023YL12). All participants signed and informed consent prior to sample collection.

All animal studies were conducted with the approval from the Animal Research Ethics Committee of the University of Science and Technology of China (Approval number USTCACUC-1801020).

## Study design

No blinding was done during data collection or analysis. Quantification of processed samples was done at the same time and analyzed with the same software settings. No data was excluded. Data were only excluded for failed experiments. Failed experiments were determined by positive and negative control experiments. In vitro experiments were performed using three biologically independent replicates per experimental group, and in vivo studies were performed using three or more biologically independent animals per experimental group to ensure sufficient statistical power. Sample sizes were determined based on previous experience and reference to existing literature.

## Data availability

All data generated or analyzed during this study are included in this published article, its supplementary information files and public available repositories. For Fig. 1A,B, the statistical graphs were generated using the GEPIA2 website (http://gepia2.cancer-pku.cn). The metabolomics data from HEK293T and MDA-MB-231 cells that support the findings of this study are uploaded in Metabolomics Workbench (https://www.metabolomicsworkbench.org/). The Study ID for the metabolomics data of MDA-MB-231 cells is ST003531 (URL link: https://dev.metabolomicsworkbench.org:22222/data/DRCCMetadata.php?Mode=Study&StudyID=ST003531&Access=YfuK4995). The Study ID for the metabolomics data of HEK293T cells is: ST003532 (URL link: https://dev.metabolomicsworkbench.org:22222/data/DRCCMetadata.php?Mode=Study&StudyID=ST003532&Access=XfrI6289).

The source data of this paper are collected in the following database record: biostudies:S-SCDT-10_1038-S44318-024-00359-z.

## Peer review information

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

## Acknowledgements

The authors thank Xiao Liu from Capital Normal University, Yingchuan Qi from ShanghaiTech University, Yuxing Chen and Zhiyong Zhang from University of Science and Technology of China (USTC). The authors thank Fanghong Shao from USTC for bioinformatics suggestions. The authors thank Lili Qu from USTC for graphic suggestions. The authors thank all members from the Laboratory of LncRNA and Diseases. The authors are grateful to the technical support from the instrument sharing center and laboratory animal center of USTC. This work was supported by National Key R&D Program of China (Grant No. 2019YFA0709300 to LL), National Natural Science Foundation of China (Grant No. 32170557 to XW and 32300450 to TS), Research Funds of Centre for Leading Medicine and Advanced Technologies of IHM (2023IHM01030 to XW and YL), and National Natural Science Foundation of China (U19A2008 to LL and 62025102 to QC).

## Author contributions

**Yan Huang**: Conceptualization; Data curation; Software; Formal analysis; Validation; Investigation; Visualization; Methodology; Writing—original draft; Project administration; Writing—review and editing. **Hua Lu**: Data curation;

Formal analysis; Validation; Visualization; Writing—review and editing. **Yao Liu**: Resources; Writing—review and editing. **Jiabei Wang**: Resources; Writing—review and editing. **Qingan Xia**: Resources; Writing—review and editing. **Xiangmin Shi**: Data curation; Software; Formal analysis; Validation; Visualization; Methodology; Writing—review and editing. **Yan Jin**: Data curation; Methodology; Writing—review and editing. **Xiaolin Liang**: Data curation; Methodology; Writing—review and editing. **Wei Wang**: Data curation; Methodology; Writing—review and editing. **Xiaopeng Ma**: Resources; Methodology; Writing—review and editing. **Yangyi Wang**: Resources; Methodology; Writing—review and editing. **Meng Gong**: Methodology; Writing—review and editing. **Canjun Li**: Methodology; Writing—review and editing. **Chunlei Cang**: Methodology; Writing—review and editing. **Qinghua Cui**: Funding acquisition; Methodology; Writing—review and editing. **Ceshi Chen**: Methodology; Writing—review and editing. **Tao Shen**: Supervision; Funding acquisition; Methodology; Writing—original draft; Project administration; Writing—review and editing. **Lianxin Liu**: Resources; Supervision; Funding acquisition; Methodology; Writing—review and editing. **Xiangting Wang**: Conceptualization; Supervision; Funding acquisition; Writing—original draft; Project administration; Writing—review and editing.

Source data underlying figure panels in this paper may have individual authorship assigned. Where available, figure panel/source data authorship is listed in the following database record: biostudies:S-SCDT-10_1038-S44318-024-00359-z.

## Disclosure and competing interests statement

The authors declare no competing interests.

# Expanded View Figures

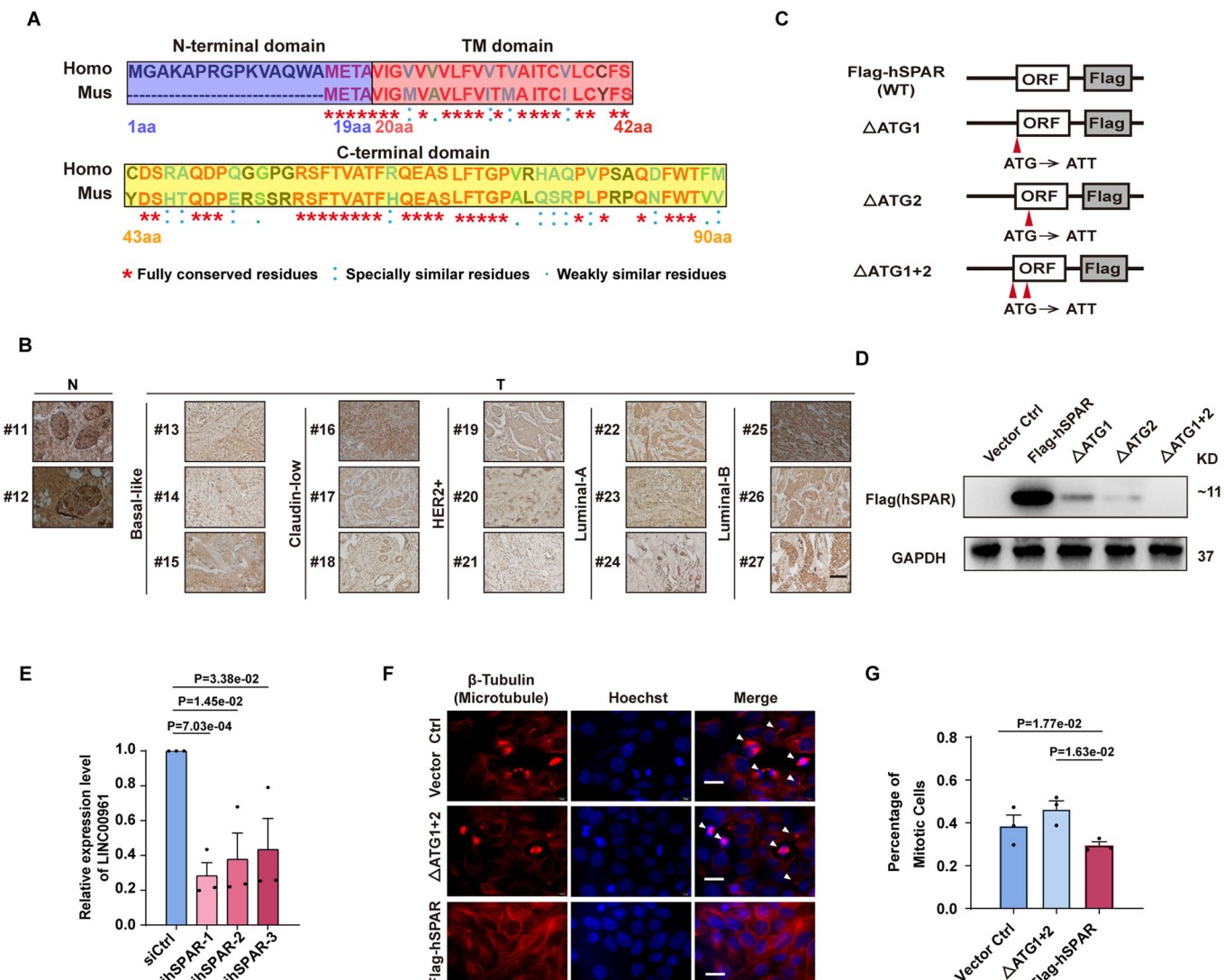

**Figure EV1.  Related to Fig. 1. The micropeptide hSPAR acts as a tumor suppressor independently of its parental lncRNA *LINC00961* in breast cancer.**

(A) The fully conserved residues are indicated by red '*', the specially similar residues are indicated by blue ':', and the weakly similar residues are indicated by green '.'. N-terminal domain is highlighted with blue box, TM domain is highlighted with red box, C-terminal domain is highlighted with yellow box. (B) The expression levels of hSPAR were detected by immunohistochemistry in the breast cancer tissues (including triple-negative (Basal-like and Claudin-low), HER2 + , Luminal-A and Luminal-B) (*n* = 15 independent biological samples) and non-tumoral tissues (*n* = 2 independent biological samples). Scale bar, 20 μm. (C) Diagram of Flag-tagged hSPAR and three translation defective constructs (ΔATG1, ΔATG2, ΔATG1 + 2). (D) Immunoblotting against Flag in extracts from MDA-MB-231 cells transfected with Vector Ctrl, ΔATG1, ΔATG2, ΔATG1 + 2, or Flag-hSPAR. GAPDH, loading control (*n* = 3 independent biological samples). (E) Knockdown efficiency of *LINC00961* analyzed by qPCR in MDA-MB-231 cells (*n* = 3 independent biological samples). Data are presented as the mean ± SEM and analyzed using one-way ANOVA with Dunnett' multiple comparisons test. (F) Immunofluorescence staining of MDA-MB-231 cells transfected with the indicated constructs for β-Tubulin (red). Nuclei were stained with Hoechst (blue). The white arrow heads indicate cells undergoing division (*n* = 3 independent biological samples). Scale bar, 10 μm. (G) Quantification of the cell number in mitosis phase from panel (F) (*n* = 3 independent biological samples). Data are presented as the mean ± SEM and analyzed using one-way ANOVA with Dunnett' multiple comparisons test. Source data are available online for this figure.

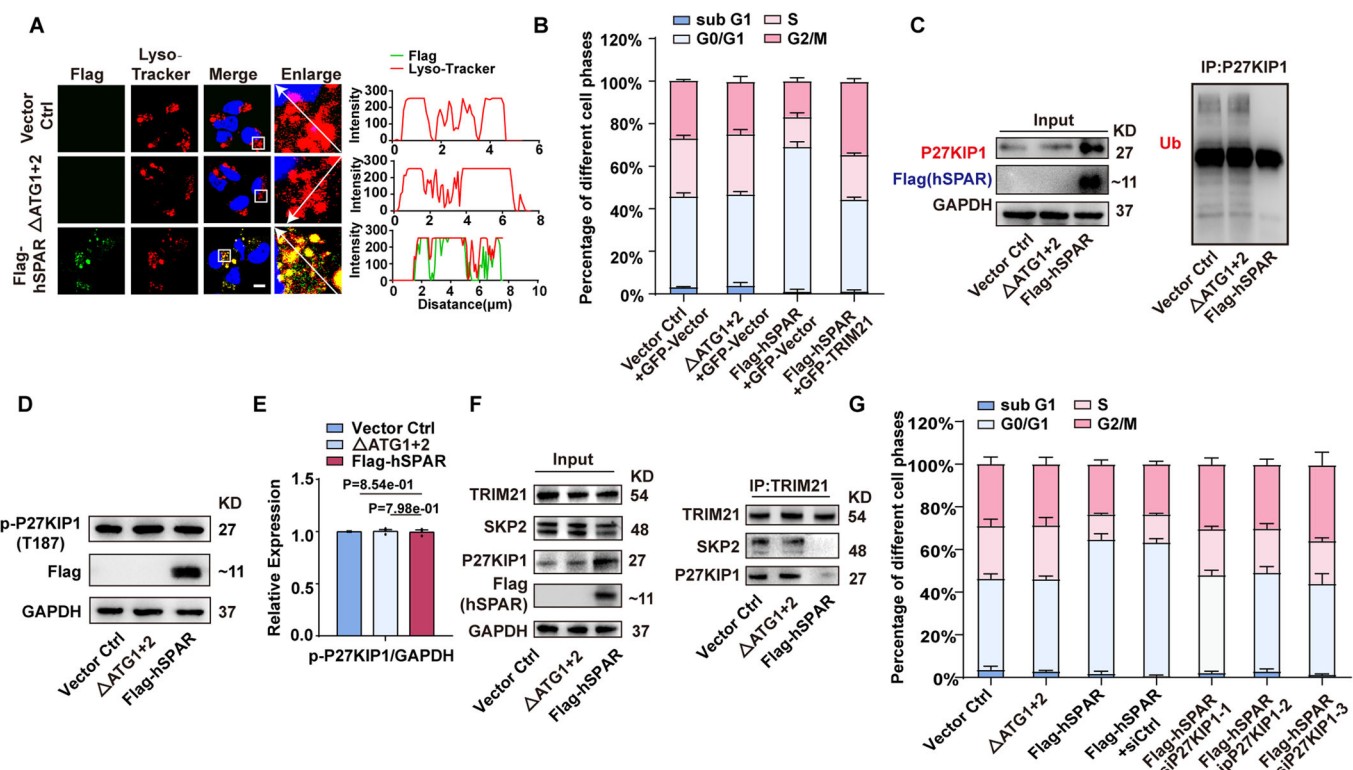

**Figure EV2. Related to Figs. 2, 3, and 4. Lysosome localized hSPAR triggers G1-S arrest in MDA-MB-231 cells in TRIM21- and P27KIP-dependent manner.**

(A) Co-immunofluorescence staining of Flag (green) and Lyso-Tracker (red) in MDA-MB-231 cells transfected with the indicated constructs. Nuclei were stained with Hoechst (blue). The graphs display the fluorescence intensity (arbitrary units) of Flag and Lyso-Tracker over the distance from adjacent image (depicted by the arrows). Scale bar, 5 μm ($n = 3$ independent biological samples). (B) Changes of cell ratio at different cell cycle phases in the presence of indicated controls and Flag-hSPAR with or without GFP-TRIM21 ($n = 3$ independent biological samples). Data are presented as the mean ± SEM. (C) Changes of the ubiquitination level of P27KIP1 were detected by Co-IP and immunoblotting from MDA-MB-231 cells transfected with Vector Ctrl, ΔATG1 + 2 or Flag-hSPAR. Left, immunoblotting of inputs. Right, immunoblotting using antibody against ubiquitin following IP of P27KIP1 ($n = 3$ independent biological samples). (D) Immunoblotting against p-P27KIP1(T187), Flag and GAPDH in extracts from MDA-MB-231 cells transfected with Vector Ctrl, ΔATG1 + 2, or Flag-hSPAR ($n = 3$ independent biological samples). (E) Quantified relative levels of p-P27KIP1(T187)/ GAPDH from panel (D) ($n = 3$ independent biological samples). Data are presented as the mean ± SEM and analyzed using one-way ANOVA with Dunnett' multiple comparisons test. (F) Interaction of TRIM21 with SKP2 and P27KIP1 detected by Co-IP and immunoblotting from MDA-MB-231 cells transfected with the indicated constructs. Left, immunoblotting of inputs. Right, immunoblotting using antibodies against TRIM21, SKP2 and P27KIP1 following IP of TRIM21 ($n = 3$ independent biological samples). (G) Changes of cell ratio at different cell cycle phases in the presence of indicated controls and Flag-hSPAR with or without P27KIP1 knockdown ($n = 3$ independent biological samples). Data are presented as the mean ± SEM. Source data are available online for this figure.

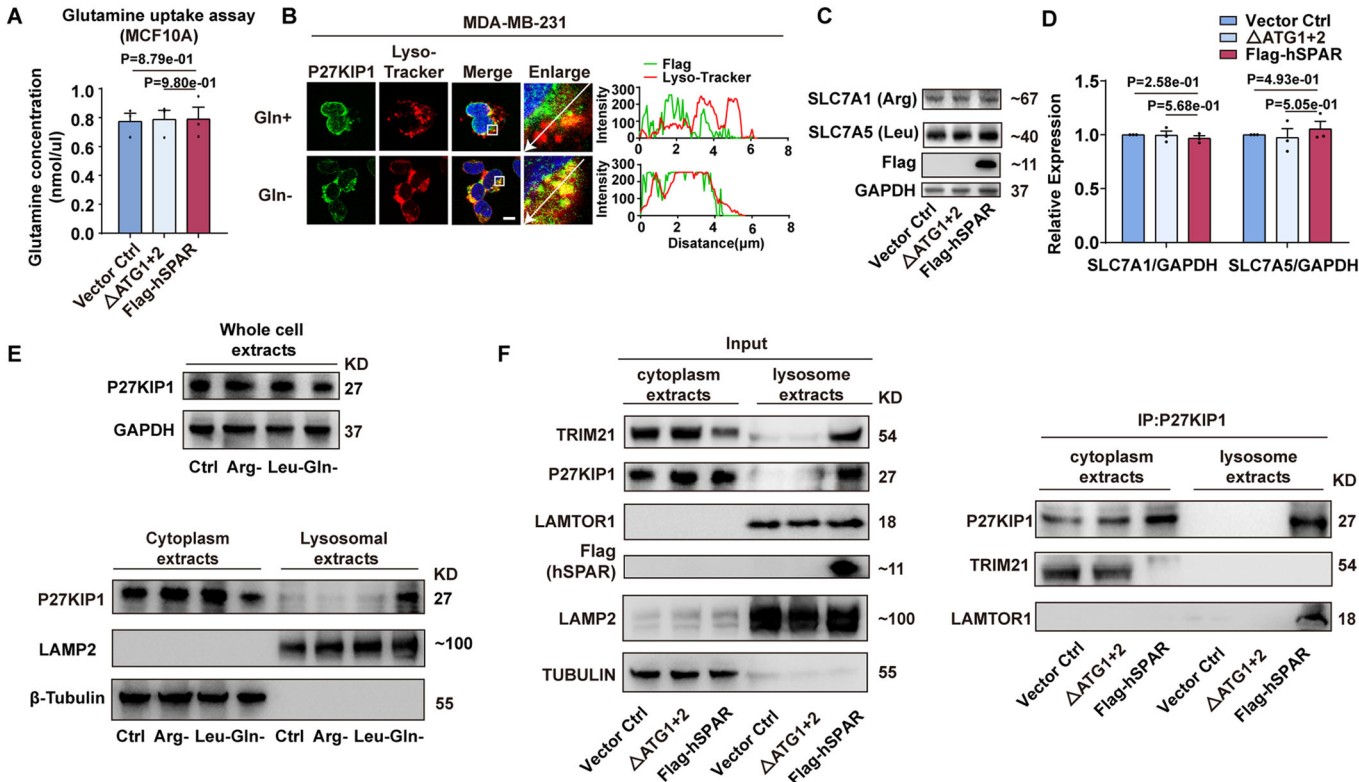

**Figure EV3.   Related to Figs. 5 and 6. The arginine and leucine not associated with the hSPAR-TRIM21-P27KIP1-mTOR pathway in MDA-MB-231 cells.**

(A) Levels of glutamine in MCF10A cells transfected with Vector Ctrl, ΔATG1 + 2 or Flag-hSPAR ($n = 3$ independent biological samples). Data are presented as the mean ± SEM and analyzed using one-way ANOVA with Dunnett' multiple comparisons test. (B) Co-immunofluorescence staining of P27KIP1 (green) and Lyso-Tracker (red) in MDA-MB-231 cells cultured with or without glutamine. Cells were permeabilized with digitonin to remove the soluble P27KIP1. Nuclei were stained with Hoechst (blue). The graphs display the fluorescence intensity (arbitrary units) of P27KIP1 and Lyso-Tracker over the distance from adjacent image (depicted by the arrows). Scale bar, 5 μm ($n = 3$ independent biological samples). (C) Immunoblotting against SLC7A1, SLC7A5, Flag and GAPDH in extracts from MDA-MB-231 cells transfected with Vector Ctrl, ΔATG1 + 2, or Flag-hSPAR ($n = 3$ independent biological samples). (D) Quantified relative levels of SLC7A1/GAPDH and SLC7A5/GAPDH from panel (C) ($n = 3$ independent biological samples). Data are presented as the mean ± SEM and analyzed using one-way ANOVA with Dunnett' multiple comparisons test. (E) Immunoblotting of whole-cell extracts (upper panel), cytoplasmic (lysosome components removed) and lysosomal extracts (lower panel) prepared from MDA-MB-231 cells cultured with or without arginine, leucine or glutamine against P27KIP1, GAPDH, LAMP2 (lysosomal marker) and β-Tubulin (cytoplasmic marker) ($n = 3$ independent biological samples). (F) Changes of interaction between P27KIP1 and TRIM21, P27KIP1 and LAMTOR1 were detected by Co-IP and immunoblotting in the indicated fractions extracts from MDA-MB-231 cells after transfection with the indicated constructs. Left, immunoblotting of inputs. Right, immunoblotting using antibodies against P27KIP1, TRIM21 and LAMTOR1 following IP of P27KIP1 ($n = 3$ independent biological samples). Source data are available online for this figure.

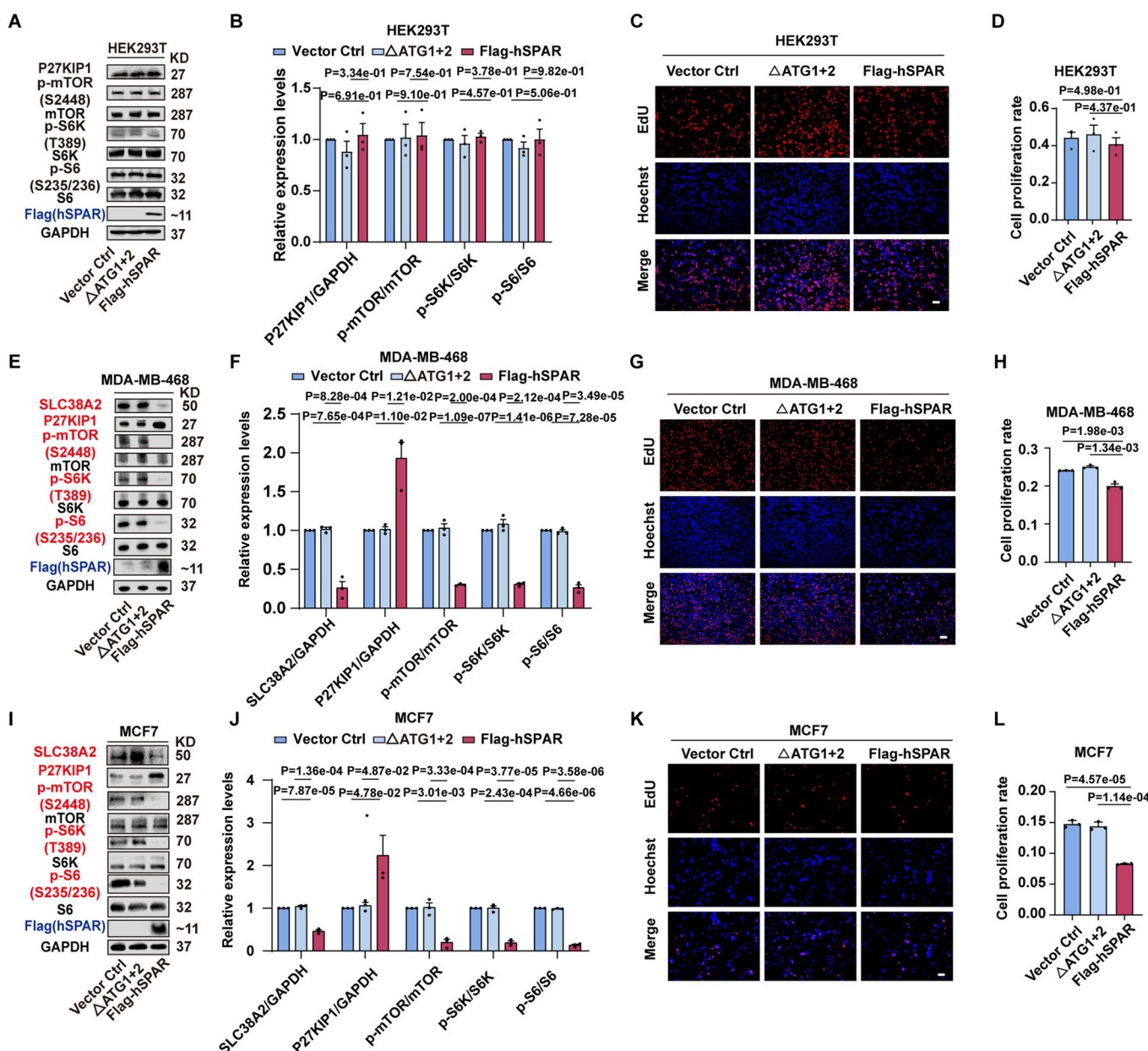

**Figure EV4. Related to Fig. 6. hSPAR inhibits P27KIP1-regulated mTOR signaling and cell proliferation in MDA-MB-468 and MCF7 cells, but not in HEK293T cells.**

(A) Immunoblotting against P27KIP1, p-mTOR, mTOR, p-S6K, S6K, p-S6, S6, Flag and GAPDH for extracts from HEK293T cells transfected with Vector Ctrl, ΔATG1 + 2 or Flag-hSPAR (n = 3 independent biological samples). (B) Quantified relative levels of P27KIP1/GAPDH, p-mTOR/mTOR, p-S6K/S6K and p-S6/S6 from panel (A) (n = 3 independent biological samples). Data are presented as the mean ± SEM and analyzed using one-way ANOVA with Dunnett' multiple comparisons test. (C) Representative images of EdU assay HEK293T cells transfected with Vector Ctrl, ΔATG1 + 2 or Flag-hSPAR (n = 3 independent biological samples). Scale bar, 50 μm. (D) Quantification of cell proliferation rate from panel (C) (n = 3 independent biological samples). Data are presented as the mean ± SEM and analyzed using one-way ANOVA with Dunnett' multiple comparisons test. (E) Immunoblotting against SLC38A2, P27KIP1, p-mTOR, mTOR, p-S6K, S6K, p-S6, S6, Flag and GAPDH for extracts from MDA-MB-468 cells transfected with Vector Ctrl, ΔATG1 + 2 or Flag-hSPAR (n = 3 independent biological samples). (F) Quantified relative levels of SLC38A2/GAPDH, P27KIP1/GAPDH, p-mTOR/mTOR, p-S6K/S6K and p-S6/S6 from panel (E) (n = 3 independent biological samples). Data are presented as the mean ± SEM and analyzed using one-way ANOVA with Dunnett' multiple comparisons test. (G) Representative images of EdU assay MDA-MB-468 cells transfected with Vector Ctrl, ΔATG1 + 2 or Flag-hSPAR (n = 3 independent biological samples). Scale bar, 50 μm. (H) Quantification of cell proliferation rate from panel (G) (n = 3 independent biological samples). Data are presented as the mean ± SEM and analyzed using one-way ANOVA with Dunnett' multiple comparisons test. (I) Immunoblotting against SLC38A2, P27KIP1, p-mTOR, mTOR, p-S6K, S6K, p-S6, S6, Flag and GAPDH for extracts from MCF7 cells transfected with Vector Ctrl, ΔATG1 + 2 or Flag-hSPAR (n = 3 independent biological samples). (J) Quantified relative levels of SLC38A2/GAPDH, P27KIP1/GAPDH, p-mTOR/mTOR, p-S6K/S6K and p-S6/S6 from panel (I) (n = 3 independent biological samples). Data are presented as the mean ± SEM and analyzed using one-way ANOVA with Dunnett' multiple comparisons test. (K) Representative images of EdU assay MCF7 cells transfected with Vector Ctrl, ΔATG1 + 2 or Flag-hSPAR (n = 3 independent biological samples). Scale bar, 50 μm. (L) Quantification of cell proliferation rate from panel (K) (n = 3 independent biological samples). Data are presented as the mean ± SEM and analyzed using one-way ANOVA with Dunnett' multiple comparisons test. The hSPAR-regulated proteins shown by immunoblotting are marked by red text. Source data are available online for this figure.

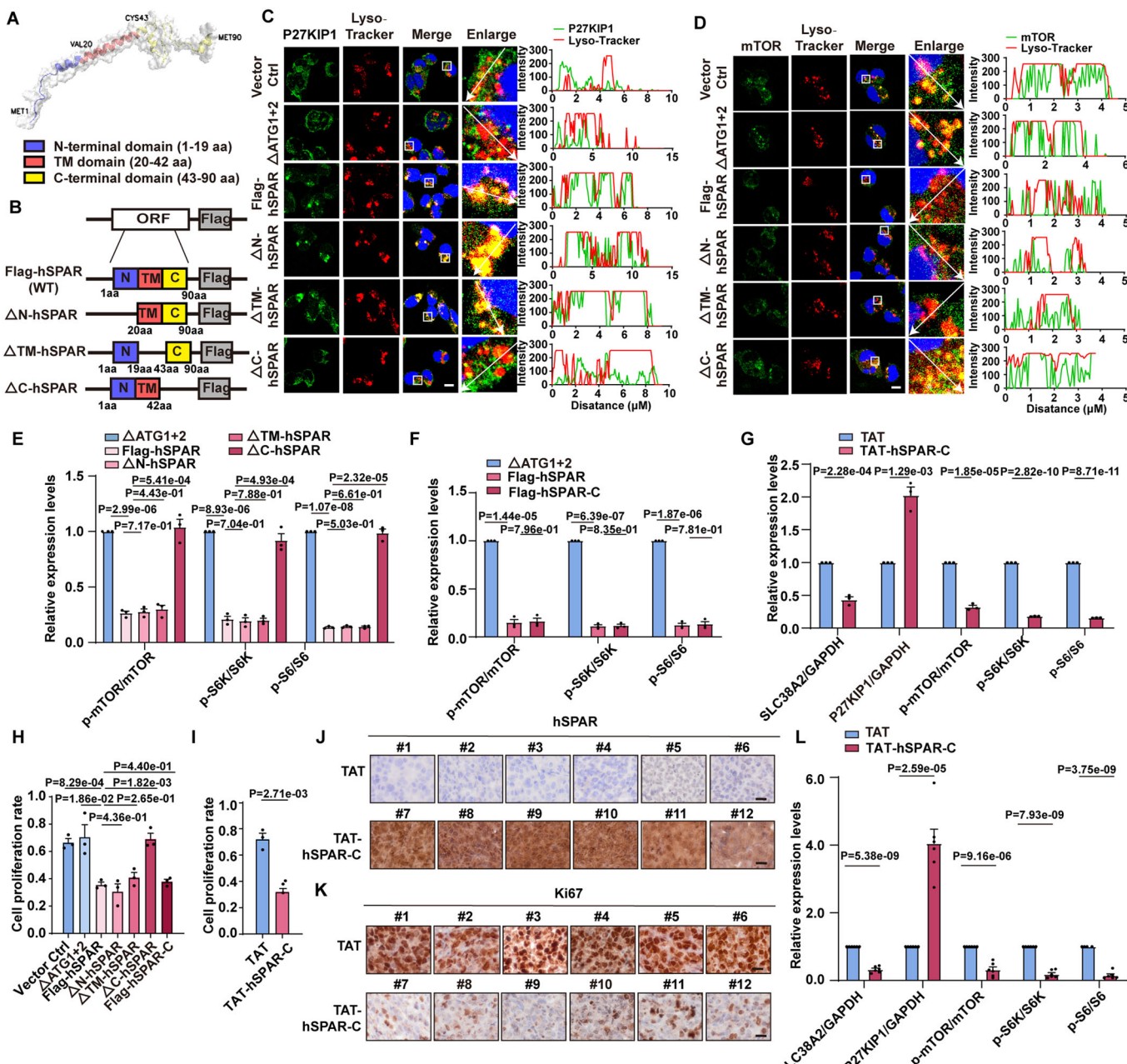

◄ **Figure EV5. Related to Fig. 7. SPAR-C inhibits mTOR signaling, cell proliferation, and tumors growth in MDA-MB-231 xenograft mice.**

(A) hSPAR is composed of an N-terminal domain, a transmembrane (TM) domain and a C-terminal domain as predicted by Alphafold2. (B) Diagram of Flag-tagged full-length hSPAR and three hSPAR domain deletion variants. (C) Co-immunofluorescence staining of P27KIP1 (green) and Lyso-Tracker (red) in MDA-MB-231 cells transfected with the indicated constructs. Cells were permeabilized with digitonin to remove the soluble P27KIP1. Nuclei were stained with Hoechst (blue). The graphs display the fluorescence intensity (arbitrary units) of P27KIP1 and Lyso-Tracker over the distance from adjacent image (depicted by the arrows). Scale bar, 5 µm ($n = 3$ independent biological samples). (D) Co-immunofluorescence staining of mTOR (green) and Lyso-Tracker (red) in MDA-MB-231 cells transfected with the indicated constructs. Cells were permeabilized with digitonin to remove the soluble mTOR. Nuclei were stained with Hoechst (blue). The graphs display the fluorescence intensity (arbitrary units) of mTOR and Lyso-Tracker over the distance from adjacent image (depicted by the arrows). Scale bar, 5 µm ($n = 3$ independent biological samples). (E) Quantified relative levels of p-mTOR/mTOR, p-S6K/S6K and p-S6/S6 from panel (Fig. 7D) ($n = 3$ independent biological samples). Data are presented as the mean ± SEM and analyzed using one-way ANOVA with Dunnett' multiple comparisons test. (F) Quantified relative levels of p-mTOR/mTOR, p-S6K/S6K and p-S6/S6 from panel (Fig. 7E) ($n = 3$ independent biological samples). Data are presented as the mean ± SEM and analyzed using one-way ANOVA with Dunnett' multiple comparisons test. (G) Quantified relative levels of SLC38A2/GAPDH, P27KIP1/GAPDH, p-mTOR/mTOR, p-S6K/S6K and p-S6/S6 from panel (Fig. 7F) ($n = 3$ independent biological samples). Data are presented as the mean ± SEM and analyzed using two-tailed Student's *t* test with Welch's correction. (H) Quantification of cell proliferation rate from panel (Fig. 7G) ($n = 3$ independent biological samples). Data are presented as the mean ± SEM and analyzed using one-way ANOVA with Dunnett' multiple comparisons test. (I) Quantification of cell proliferation rate from panel (Fig. 7H) ($n = 3$ independent biological samples). Data are presented as the mean ± SEM and analyzed using two-tailed Student's *t* test with Welch's correction. (J) Immunohistochemistry of TAT-hSPAR-C detected by immunohistochemistry with the anti-hSPAR antibody in the xenografts from panel (Fig. 7I) ($n = 12$ independent biological samples). Scale bar, 20 µm. (K) Immunohistochemistry of cell proliferation marker Ki67 in the xenografts from panel (Fig. 7I) ($n = 12$ independent biological samples). Scale bar, 20 µm. (L) Quantified relative levels of SLC38A2/GAPDH, P27KIP1/GAPDH, p-mTOR/mTOR, p-S6K/S6K and p-S6/S6 from panel (Fig. 7O) ($n = 12$ independent biological samples). Data are presented as the mean ± SEM and analyzed using two-tailed Student's *t* test with Welch's correction. Source data are available online for this figure.

