## [Peer Review File · The EMBO Journal]

Micropeptide hSPAR regulates glutamine levels and suppresses mammary tumor growth via a TRIM21-P27KIP1-mTOR axis

Yan Huang, Hua Lu, Yao Liu, Jiabei Wang, Qingan Xia, Xiangmin Shi, Yan Jin, Xiaolin Liang, Wei Wang, Xiaopeng Ma, Yangyi Wang, Meng Gong, Canjun Li, Chunlei Cang, Qinghua Cui, Ceshi Chen, Tao Shen, Lianxin Liu, and Xiangting Wang

Corresponding authors: Xiangting Wang (wangxt11@ustc.edu.cn) , Lianxin Liu (liulx@ustc.edu.cn), Tao Shen (stao@ahnu.edu.cn)

Review Timeline:

Submission Date:	14th Jun 24
Editorial Decision:	17th Aug 24
Revision Received:	16th Oct 24
Editorial Decision:	18th Nov 24
Revision Received:	26th Nov 24
Accepted:	4th Dec 24

Editor: Daniel Klimmeck

Transaction Report:

Dear Dr Wang,

Thank you again for the submission of your manuscript (EMBOJ-2024-118035) to The EMBO Journal. Please accept my apologies for getting back to you with unusual delay due to protracted referee input and detailed discussion in the editorial team. As mentioned earlier, your study was assessed by two reviewers with expertise in tumor biology and TOR signaling, whose comments are enclosed below.

As you will see from the experts' reports, the referees acknowledge the analysis and potential interest of your results. However, they also express major concerns regarding completeness and endogenous relevance of the findings, which need to be addressed thoroughly to make them supportive of publication in the EMBO Journal. The reviewers also raise a number of issues related to the data presentation, additional controls and improved methods annotation required, statistics applied and overall discussion of related literature, that would need to be conclusively addressed to achieve the level of robustness and clarity needed for The EMBO Journal.

Given the overall interest stated and broader angle of your findings, we are able to invite you to revise your manuscript experimentally to address the referees' comments. I need to stress though that we do require strong support from the referees on a revised version of the study in order to move on to publication of the work.

In light of the extensive experimentation requested, I would appreciate if you could contact me during the next weeks for exchange e.g. a video call to discuss your perspective on the comments and potential plan for revisions.

Please feel free to contact me if you have any questions or need further input on the referee comments.

When submitting your revised manuscript, please carefully review the instructions below.

Please feel free to approach me any time should you have additional questions related to this.

Thank you for the opportunity to consider your work for publication.

I look forward to your revision.

Kind regards,

Daniel Klimmeck

Daniel Klimmeck, PhD
Senior Editor
The EMBO Journal

Instruction for the preparation of your revised manuscript:

- 1) a .docx formatted version of the manuscript text (including legends for main figures, EV figures and tables). Please make sure that the changes are highlighted to be clearly visible.
- 2) individual production quality figure files as .eps, .tif, .jpg (one file per figure).
- 3) a .docx formatted letter INCLUDING the reviewers' reports and your detailed point-by-point response to their comments. As part of the EMBO Press transparent editorial process, the point-by-point response is part of the Review Process File (RPF), which will be published alongside your paper.
- 4) a complete author checklist, which you can download from our author guidelines (<https://wol-prod-cdn.literatumonline.com/pb->

assets/embo-site/Author Checklist%20-%20EMBO%20J-1561436015657.xlsx). Please insert information in the checklist that is also reflected in the manuscript. The completed author checklist will also be part of the RPF.

6) It is mandatory to include a 'Data Availability' section after the Materials and Methods. Before submitting your revision, primary datasets produced in this study need to be deposited in an appropriate public database, and the accession numbers and database listed under 'Data Availability'. Please remember to provide a reviewer password if the datasets are not yet public (see <https://www.embopress.org/page/journal/14602075/authorguide#datadeposition>).

7) Our journal encourages inclusion of *data citations in the reference list* to directly cite datasets that were re-used and obtained from public databases. Data citations in the article text are distinct from normal bibliographical citations and should directly link to the database records from which the data can be accessed. In the main text, data citations are formatted as follows: "Data ref: Smith et al, 2001" or "Data ref: NCBI Sequence Read Archive PRJNA342805, 2017". In the Reference list, data citations must be labeled with "[DATASET]". A data reference must provide the database name, accession number/identifiers and a resolvable link to the landing page from which the data can be accessed at the end of the reference. Further instructions are available at .

8) At EMBO Press we ask authors to provide source data for the main and EV figures. Our source data coordinator will contact you to discuss which figure panels we would need source data for and will also provide you with helpful tips on how to upload and organize the files.

Numerical data can be provided as individual .xls or .csv files (including a tab describing the data). For 'blots' or microscopy, uncropped images should be submitted (using a zip archive or a single pdf per main figure if multiple images need to be supplied for one panel). Additional information on source data and instruction on how to label the files are available at .

9) We replaced Supplementary Information with Expanded View (EV) Figures and Tables that are collapsible/expandable online (see examples in <https://www.embopress.org/doi/10.15252/emj.201695874>). A maximum of 5 EV Figures can be typeset. EV Figures should be cited as 'Figure EV1, Figure EV2' etc. in the text and their respective legends should be included in the main text after the legends of regular figures.

11) For data quantification: please specify the name of the statistical test used to generate error bars and P values, the number (n) of independent experiments (specify technical or biological replicates) underlying each data point and the test used to calculate p-values in each figure legend. The figure legends should contain a basic description of n, P and the test applied. Graphs must include a description of the bars and the error bars (s.d., s.e.m.).

The revision must be submitted online within 90 days; please click on the link below to submit the revision online before 15th Nov 2024.

Referee #1:

Comments to Authors:

The manuscript entitled, "Micropeptide hSPAR, a glutamine regulator, suppresses tumor growth via TRIM21-P27KIP1-mTOR pathway" reports that hSPAR inhibits the E3 ligase TRIM21-mediated P27KIP1 degradation, resulting in the cytoplasmic accumulation of P27KIP1 in breast cancer cells. hSPAR also functions as a glutamine transport inhibitor, decreasing glutamine levels in cancer cells, which promotes the translocation of P27KIP1 to the lysosome where it disrupts the Ragulator complex and mTORC1 activity. Overall, the authors conclude that hSPAR inhibits mTOR activity by blocking glutamine entry into the cell and simultaneously decreasing P27KIP1 ubiquitination, resulting in increased P27KIP1 lysosomal localization and disruption of the mTORC1 complex. The study provides insights into mTOR signaling, identifies an important role for P27KIP1 lysosomal localization in regulating mTORC1, and would be of interest to the readership of EMBO. The study was for the most part thorough and well designed; however, there are several aspects that should be considered and addressed prior to publication. While the authors' findings adequately demonstrate that TRIM21-hSPAR interactions decrease TRIM21-P27KIP1 interactions and, thereby, prevent P27KIP1 ubiquitination, the mechanism by which this occurs is unclear and would benefit from further investigation. These and other points that should be considered prior to publication are listed below.

Major comments:

1. TRIM21 functions as part of an E3 ubiquitin ligase complex that is like SCF(Skp2) and consists of TRIM21, Skp2, Skp1, and Cul1. In this complex, Skp2 captures the Thr187-phosphorylated P27. Skp1 then couples P27 to the complex putting it into proximity of the catalytic core (Cul1 and TRIM21). Thus, one possible explanation for the decrease in TRIM21-dependent ubiquitination of P27KIP1 is that hSPAR sequesters TRIM21 and prevents the assembly of the SCF(Skp2)-like complex. One way to examine this is to perform co-IPs in vector control, Δ ATG1+2, and Flag-hSPAR expressing cells to probe for TRIM21-Skp2 interactions. However, hSPAR also appears to place TRIM21 in proximity to P27KIP1 at the lysosome (Fig. 5B), raising the question of whether hSPAR is simply preventing direct TRIM21-P27KIP1 interactions or if it is also blocking TRIM21's catalytic activity in some way. An initial way to probe this is to utilize proximity ligation assays (PLA) to examine the TRIM21-P27KIP1 and TRIM21-hSPAR interactions in-situ, which could give more precise insight to the co-localization and interactions of these proteins. Additionally, a Ubiquitin Chain Formation Assay (PMID: 33619271) using hSPAR-bound TRIM21 as the E3 ligase could shed light on the impact of hSPAR on TRIM21 catalytic activity.
2. The authors conclude that hSPAR inhibits glutamine uptake, successfully demonstrating that this is due to hSPAR-mediated downregulation of the glutamine transport protein, SLC38A2. However, the metabolomics study indicated that hSPAR expression is correlated with the dysregulation of other metabolites in addition to glutamine. It would be useful to include a table that summarizes the analyzed metabolites and their trends as well as a discussion point regarding what the trends implicate.
3. How is glutamine availability causing the translocation of P27KIP1 to the lysosome? Given the importance of this finding, ideally this would be further investigated or at the least discussed in more detail with some ideas about the possible mechanism involved.
4. The activity (i.e., phosphorylation) of P27KIP1 should be examined. Given phosphorylation of P27KIP1 by AKT impairs P27 nuclear import and opposes P27-mediated G1 arrest (PMID: 12244302), it would also be helpful to examine nuclear localization and accumulation of P27KIP1 relative to cytoplasmic. This will be particularly helpful in Figure 5.
5. Previous reports of P27KIP1 lysosomal localization did not specify glutamine as the only amino acid involved in this process. Further, arginine and leucine are both also linked to mTORC1 activity. Thus, if arginine and leucine were also dysregulated in response to hSPAR expression, it would be interesting to further probe the role of hSPAR in regulating their uptake by first probing the effect of hSPAR on their entry transport proteins.
6. Conclusions throughout the manuscript from imaging studies are supported by representative images of live cell images/immunofluorescence and single line-scans, which would benefit from additional analysis. Further quantification of signal co-localization such as Pearson's correlation coefficient or Mander's overlap coefficient would be helpful to support claims of localization. Additional techniques such as in situ proximity ligation assay (PLA) could be used to visualize co-localization and protein-protein interactions and lysosomal localization, and would significantly strengthen the manuscript.
7. To better compare differences cancer cells to normal, healthy cells, MCF10A mammary cells should be used as normal cell controls for MDA-MB-231 cells (for example in Fig. 5e).
8. This raises the question, why are the effects of hSPAR not observed in non-cancerous cells; further discussion of this is warranted?

9. Where does the ability of SPAR to bind the v-ATPase and hamper the assembly of mTORC1 fit into the story?

Minor comments:

1. It would be useful to include a discussion of why TRIM21 was selected out of the list of hSPAR-interacting proteins.
2. LysoTracker probes are fluorescent acidotropic probes for labeling and tracking acidic organelles in live cells. It is unusual to have LysoTracker used in live cells followed by immuno-fluorescence staining. Why wasn't LAMP1/2 immunostaining performed here?
3. Please indicate the phosphorylation sites for phospho-proteins evaluated via Western blotting on all figures.
4. Quantification of Western blots should be normalized to Ctrl conditions so that changes in relative expression levels between treatments can be more easily visualized by the reader.
5. Please include amino acid numbers on construct diagrams to improve the clarity of the construct design.
6. Please increase the size of all scale bars. Currently, they are too small to read and even see in some images.
7. No citation was included for the previously reported treatment for the removal of soluble P27KIP1 or mTOR from cells.
8. There were several typos throughout the manuscript, including on line 164, 222, 228, 240, 290, and 359.

Referee #2:

This paper focuses on the negative regulation of mTORC1 by the micropeptide, hSPAR. It is topical, because, there is growing realisation that micropeptides, derived from what were previously defined as 'non-coding RNAs', have important functions. A role for hSPAR downregulation in mTORC1 signalling and muscle regeneration has previously been shown (Matsumoto et al., 2017 Nature). It was postulated to involve an inhibitory interaction with the V-ATPase, which recruits mTORC1 to the lysosomal membrane on acute addition of amino acids in HEK293 cells, but the mechanism mediating its long-term effect on mTORC1 during muscle regeneration was not characterised. To some extent, the current study builds on these findings, providing mechanistic insights into the role for hSPAR in cancer mediated via its C-terminus.

The authors identify two additional mechanisms by which hSPAR inhibits mTORC1. They show that the C-terminus of hSPAR (hSPAR-C) promotes the non-canonical role of P27KIP1 as an inhibitor of the mTORC1 axis by stabilising cytoplasmic P27KIP1, and also leads to glutamine deprivation-induced lysosomal localisation of P27KIP1. Furthermore, they show that hSPAR-C inhibits the amino acid transporter SLC38A2, therefore decreasing cellular glutamine levels, and blocking mTORC1 through this combinatorial mechanism more strongly than suppressing the two mechanisms separately. This logical analysis, involving overexpression and knockdown experiments, sheds light on the mechanisms underlying their findings at the beginning of the manuscript that hSPAR acts as a tumour suppressor in breast cancer cells, which might not be considered entirely surprising given the previous publication on its mTORC1-mediated role in muscle cell regeneration. The schematic in Figure 7p nicely illustrates the final model and could be the basis for a helpful graphical abstract.

I have some concerns that I think need to be addressed to reflect the current state of the field, appropriately present the data, and to support the conclusions:

1. I am not an expert in this specific field, but it appears that the authors have omitted a number of publications on the lncRNA associated with hSPAR, LINC00961. They cite two references, Matsumoto et al., 2017, and Spencer et al., 2020, which identifies differing roles for hSPAR and LINC00961 in endothelial cells.

There are, however, also several other papers where roles have been identified for LINC00961, but the possible contribution of hSPAR has not been fully studied and some of these are relevant to cancer, so I am surprised that none of these have been cited eg.

Jiang B, Liu J, Zhang YH, Shen D, Liu S, Lin F, Su J, Lin QF, Yan S, Li Y, Mao WD, Liu ZL. Long noncoding RNA LINC00961 inhibits cell invasion and metastasis in human non-small cell lung cancer. *Biomed Pharmacother.* 2018 Jan;97:1311-1318. doi: 10.1016/j.biopha.2017.11.062.

Huang Z, Lei W, Tan J, Hu HB. Long noncoding RNA LINC00961 inhibits cell proliferation and induces cell apoptosis in human non-small cell lung cancer. *J Cell Biochem.* 2018 Nov;119(11):9072-9080. doi: 10.1002/jcb.27166.

Lu XW, Xu N, Zheng YG, Li QX, Shi JS. Increased expression of long noncoding RNA LINC00961 suppresses glioma metastasis and correlates with favorable prognosis. *Eur Rev Med Pharmacol Sci.* 2018 Aug;22(15):4917-4924. doi: 10.26355/eurrev_201808_15630.

Chen D, Zhu M, Su H, Chen J, Xu X, Cao C. LINC00961 restrains cancer progression via modulating epithelial-mesenchymal transition in renal cell carcinoma. *J Cell Physiol.* 2019 May;234(5):7257-7265. doi: 10.1002/jcp.27483.

Yin J, Liu Q, Chen C, Liu W. Small regulatory polypeptide of amino acid response negatively relates to poor prognosis and controls hepatocellular carcinoma progression via regulating microRNA-5581-3p/human cardiolipin synthase 1. *J Cell Physiol.* 2019 Aug;234(10):17589-17599. doi: 10.1002/jcp.28383.

Zhang L, Shao L, Hu Y. Long noncoding RNA LINC00961 inhibited cell proliferation and invasion through regulating the Wnt/ β -catenin signaling pathway in tongue squamous cell carcinoma. *J Cell Biochem.* 2019 Aug;120(8):12429-12435. doi: 10.1002/jcb.28509.

Pan LN, Sun YR. LINC00961 suppresses cell proliferation and induces cell apoptosis in oral squamous cell carcinoma. *Eur Rev Med Pharmacol Sci.* 2019 Apr;23(8):3358-3365. doi: 10.26355/eurrev_201904_17699.

Mu X, Mou KH, Ge R, Han D, Zhou Y, Wang LJ. Linc00961 inhibits the proliferation and invasion of skin melanoma by targeting the miR-367/PTEN axis. *Int J Oncol.* 2019 Sep;55(3):708-720. doi: 10.3892/ijo.2019.4848.

Wu H, Dai Y, Zhang D, Zhang X, He Z, Xie X, Cai C. LINC00961 inhibits the migration and invasion of colon cancer cells by sponging miR-223-3p and targeting SOX11. *Cancer Med.* 2020 Apr;9(7):2514-2523. doi: 10.1002/cam4.2850. Epub 2020 Feb 11. PMID: 32045135; PMCID: PMC7131851.

Most relevant to the manuscript, Mu et al., 2022 and other groups looking at the normal function of LINC00961 have postulated an effect on PTEN/PI3K, which modulates mTORC1 activity. In this case, the authors not only should cite and discuss the reference, but it would be straightforward to check whether the genetic manipulations they undertake affect PI3K signalling in addition to mTORC1. For example, are there differences in PI3K regulation when LINC00961 is knocked down but not when hSPAR is overexpressed?

One additional clinical study was directly related to breast cancer and so should also be discussed.

Mehrpour Layeghi S, Arabpour M, Esmaeili R, Naghizadeh MM, Tavakkoly Bazzaz J, Shakoori A. Evaluation of the potential role of long non-coding RNA LINC00961 in luminal breast cancer: a case-control and systems biology study. *Cancer Cell Int.* 2020 Oct 2;20:478. doi: 10.1186/s12935-020-01569-1.

2. Based on their data, the authors conclude that hSPAR-C only affects glutamine levels and cell growth specifically in cancer cells. This conclusion seems to be based on their work in breast cancer cell lines compared to HEK-293T cells, which the authors term 'non-cancerous', but, as a cell line, they are far from normal. The conclusion made by the authors from the data in Fig. S5 is that hSPAR has no effect on mTORC1 in non-cancerous cells, but the Matsumoto et al., 2017 paper suggests that it does regulate steady-state mTORC1 activity in regenerating muscle over a period of days. Unless other data are provided, I think the previously published data need to be discussed and this conclusion toned down.

3. The co-localisation data analysis in Figs. 5g and 7c appears extremely selective in several cases, choosing areas where there is no yellow signal in genetic backgrounds where the authors argue there is no colocalization, when other cells appear to have significant colocalization, eg. 5g: Gln+ top cell; 7c: FLAG-hSPAR bottom cell and cells for deltaTM-hSPAR; also, vice versa in 5g, Gln-. It would be more appropriate to measure the overall colocalization throughout all cells to confirm a significant change. In this respect, the cell fractionation in Fig. 5h is also a concern. The lysosomal fraction for Gln- seems to have a higher level of tubulin present, which would explain the result through cytoplasmic contamination.

4. Much of the data analysis involves multiple comparisons, yet the data are apparently analysed by two-tailed t-test. A non-parametric multiple comparisons test should be used. I think this is unlikely to affect the final conclusions, but the correct test should be employed.

Minor comments

Line 78: The V-ATPase is an activator of mTORC1 lysosomal recruitment, not an inhibitor?

The manuscript is on the whole well-written, but would benefit from careful proof reading. I have highlighted some concerns about the language used to express specific points below as examples:

Line 37: Replace 'discovered' with 'uncovered'

Line 56: 'proven as the' with 'shown to be'

Lines 63-64: Replace 'far beyond' with 'not well'.

Lines 76-77: Replace 'cell' with 'cells'.

Line 237: 'glutamine entry transport' would be better phrased as, 'glutamine transporter'

Line 371: Substitute, 'regulations on' with 'regulators'

Line 371, 379 etc: add 'the' before 'cytoplasm', 'glutamine transporter' respectively etc

Dear Dr. Klimmeck and Reviewers,

Thank you very much for your precious time on reviewing our manuscript (MS ID: EMBOJ-2024-118191). We sincerely appreciate all valuable comments and suggestions from you to greatly improve the quality of our manuscript. The main changes of our revised manuscript (EMBOJ-2024-118191_R1) are summarized as following:

- (1) We have supplemented new results showing that hSPAR inhibits TRIM21-dependent ubiquitination of P27KIP1 by disrupting the interaction between TRIM21 and SKP2, and that P27KIP1 does not interact with TRIM21 on lysosomes.
- (2) We have re-conducted all the co-localization experiments by using LAMP1 as a lysosome marker to strengthen our conclusion about the investigated co-localization and protein-protein interactions.
- (3) We have included non-tumorous MCF10A cells as an additional control beside the originally used HEK293T cells for the glutamine related experiments.
- (4) We have supplemented new results showing that hSPAR does not affect the tested arginine or leucine transporter, and that depletion of arginine or leucine does not influence the lysosomal localization of P27KIP1.
- (5) We have supplemented new results showing that hSPAR does not impact the tested PI3K signaling related phosphorylation of P27KIP1 (T187) or alter the nuclear localization of P27KIP1.
- (6) We have made modifications to all the requested images by changing the scale bar size, and adding additional figure legends.
- (7) We have improved the methods annotation, applied statistics, and discussions as suggested by both reviewers.

Our responses to reviewers' comments are described in a point-by-point manner. All the modifications related to the figures and tables have been discussed in our response. All the text changes have been highlighted with red in the revised manuscript. We hope that our manuscript will be acceptable for publication in *the EMBO Journal*.

Best Regards,
Xiangting Wang

Point by Point Response

Comments from Reviewer #1: The manuscript entitled, "Micropeptide hSPAR, a glutamine regulator, suppresses tumor growth via TRIM21-P27KIP1-mTOR pathway" reports that hSPAR inhibits the E3 ligase TRIM21-mediated P27KIP1 degradation, resulting in the cytoplasmic accumulation of P27KIP1 in breast cancer cells. hSPAR also functions as a glutamine transport inhibitor, decreasing glutamine levels in cancer cells, which promotes the

translocation of P27KIP1 to the lysosome where it disrupts the Ragulator complex and mTORC1 activity. Overall, the authors conclude that hSPAR inhibits mTOR activity by blocking glutamine entry into the cell and simultaneously decreasing P27KIP1 ubiquitination, resulting in increased P27KIP1 lysosomal localization and disruption of the mTORC1 complex. The study provides insights into mTOR signaling, identifies an important role for P27KIP1 lysosomal localization in regulating mTORC1, and would be of interest to the readership of EMBO. The study was for the most part thorough and well designed; however, there are several aspects that should be considered and addressed prior to publication. While the authors' findings adequately demonstrate that TRIM21-hSPAR interactions decrease TRIM21-P27KIP1 interactions and, thereby, prevent P27KIP1 ubiquitination, the mechanism by which this occurs is unclear and would benefit from further investigation. These and other points that should be considered prior to publication are listed below.

Response: We sincerely thank Reviewer#1 for the positive comments on our manuscript. The constructive suggestions are great helpful to improve the impact of our manuscript.

Major comments:

1. TRIM21 functions as part of an E3 ubiquitin ligase complex that is like SCF(Skp2) and consists of TRIM21, Skp2, Skp1, and Cul1. In this complex, Skp2 captures the Thr187-phosphorylated P27. Skp1 then couples P27 to the complex putting it into proximity of the catalytic core (Cul1 and TRIM21). Thus, one possible explanation for the decrease in TRIM21-dependent ubiquitination of P27KIP1 is that hSPAR sequesters TRIM21 and prevents the assembly of the SCF(Skp2)-like complex. One way to examine this is to perform co-IPs in vector control, Δ ATG1+2, and Flag-hSPAR expressing cells to probe for TRIM21-Skp2 interactions.

Response: This is a great suggestion. Following this suggestion, we have conducted Co-IP assays. Our results showed a decreased TRIM21-SKP2 and TRIM21-P27KIP1 interaction in Flag-hSPAR expressing cells (revised Figure EV2F, also shown below, upper panel). In addition, our ubiquitination experiments results showed TRIM21 was the mediator for the Flag-hSPAR-reduced ubiquitination of P27KIP1 (original and revised Figure 4B-C; also shown below, middle and bottom panels). Collectively, our data suggest that hSPAR inhibits TRIM21-dependent ubiquitination of P27KIP1 by affecting the interaction between TRIM21 and SKP2.

Revised Figure EV2F: Changes of interaction between TRIM21 and P27KIP1, TRIM21 and SKP2 in Flag-hSPAR overexpressing MDA-MB-231 cells.

Revised Figure 4B (original Figure 4B): Changes of the ubiquitination level of P27KIP1 detected by Co-IP and immunoblotting from MDA-MB-231 cells transfected with Vector Ctrl, Δ ATG1+2 or Flag-hSPAR, together with HA-Ub, followed by treatment with DMSO or the proteasome inhibitor MG132 (5 μ M).

Revised Figure 4C (original Figure 4C): Changes of the ubiquitination level of P27KIP1 after co-transfection with GFP-TRIM21 in the presence of Flag-hSPAR and HA-Ub detected by Co-IP and immunoblotting from MDA-MB-231 cells.

hSPAR also appears to place TRIM21 in proximity to P27KIP1 at the lysosome (Fig. 5B), raising the question of whether hSPAR is simply preventing direct TRIM21-P27KIP1 interactions or if it is also blocking TRIM21's catalytic activity in some way. An initial way to probe this is to utilize proximity ligation assays (PLA) to examine the TRIM21-P27KIP1 and TRIM21-hSPAR interactions in-situ, which could give more precise insight to the co-localization and interactions of these proteins. Additionally, a Ubiquitin Chain Formation Assay (PMID: 33619271) using hSPAR-bound TRIM21 as the E3 ligase could shed light on the impact of hSPAR on TRIM21 catalytic activity.

Response: This point is initiated by a hypothesis from the Reviewer#1. Based on our data (hSPAR induced lysosomal localization of both TRIM21 and P27KIP1), it may be possible that P27KIP1 keeps interaction with TRIM21 on lysosomes. To address this question/hypothesis, we have provided additional results in the revised Figure EV3F (also shown below). Our results show that P27KIP1 does not interact with TRIM21, but binds to LAMTOR1 in the hSPAR expressing MDA-MB-231 cells. Therefore, our results support the Reviewer#1's another hypothesis—“hSPAR is simply preventing direct TRIM21-P27KIP1 interactions” (quote). For this case, we think the Ubiquitin Chain Formation Assay is not necessary for the current scope of our manuscript. We have also discussed these results with the Editor. With a positive response, the Editor agreed with us and overruled the PLA

experiments. We hope that Reviewer#1 would also consider our explanation as reasonable and acceptable.

Revised Figure EV3F: TRIM21 dissociates P27KIP1 on lysosomes. Changes of interaction between P27KIP1 and TRIM21, P27KIP1 and LAMTOR1 were detected by Co-IP and immunoblotting in the indicated fractions extracts from MDA-MB-231 cells after transfection with the indicated constructs. Left, immunoblotting of inputs. Right, immunoblotting using antibody against P27KIP1, TRIM21 and LAMTOR1 following IP of P27KIP1.

2. The authors conclude that hSPAR inhibits glutamine uptake, successfully demonstrating that this is due to hSPAR-mediated downregulation of the glutamine transport protein, SLC38A2. However, the metabolomics study indicated that hSPAR expression is correlated with the dysregulation of other metabolites in addition to glutamine. It would be useful to include a table that summarizes the analyzed metabolites and their trends as well as a discussion point regarding what the trends implicate.

Response: As suggested, we have supplemented Table. EV1. The related "Discussion" is also provided below.

In addition to glutamine, our metabolomics data indicate that overexpression of hSPAR results in alterations of various amino acids in MDA-MB-231 cells (Table. EV1), suggesting that hSPAR plays a crucial role in amino acid metabolism processing. The underlying mechanisms await further exploration.

3. How is glutamine availability causing the translocation of P27KIP1 to the lysosome? Given the importance of this finding, ideally this would be further investigated or at the least discussed in more detail with some ideas about the possible mechanism involved.

Response: We appreciate this insightful scientific question. As suggested, we have added discussions in the revised "Discussion" section (also provided below).

It has been reported that phosphorylation of threonine residues at position 157 and 198 of P27KIP1 can promote its localization in the cytoplasm (Liang *et al*, 2002; Shin *et al*, 2002; Viglietto *et al*, 2002). While phosphorylation of serine residue at

position 10 of P27KIP1 can promote its localization in the nucleus (Ishida *et al*, 2015; Rodier *et al*, 2001). Our data show that a large amount of P27KIP1 is translocated to the lysosomes in glutamine-deficient MDA-MB-231 cells. It would be of great interest to explore whether certain posttranslational modifications of P27KIP1 are involved in P27KIP1's lysosomal accumulation in the future investigations.

4. The activity (i.e., phosphorylation) of P27KIP1 should be examined. Given phosphorylation of P27KIP1 by AKT impairs P27 nuclear import and opposes P27-mediated G1 arrest (PMID: 12244302), it would also be helpful to examine nuclear localization and accumulation of P27KIP1 relative to cytoplasmic. This will be particularly helpful in Figure 5.

Response: As suggested, we have conducted additional experiments and supplemented these results in the revised Figure EV2D, EV2E, and Figure 5A. In the revised Figure EV2D and EV2E, we find that the phosphorylation of P27KIP1 at T187 (phosphorylation that regulated by AKT) is not affected by hSPAR (also shown below, left and middle panels). In the revised Figure 5A, our results show that hSPAR does not alter the nuclear localization of P27KIP1 (also shown below, right panel).

Left and middle, Revised Figure EV2D and EV2E: Immunoblotting against p-P27KIP1(T187), Flag and GAPDH in extracts from MDA-MB-231 cells transfected with Vector Ctrl, Δ ATG1+2, or Flag-hSPAR. **Right, Revised Figure 5A (original Figure 5A):** Immunoblotting of nuclear, cytoplasmic (membrane components removed), and membrane fractions prepared from MDA-MB-231 cells transfected with the indicated constructs against P27KIP1, TRIM21, FIBRILLARIN (nuclear marker), β -Tubulin (cytoplasmic marker), ATP1V1A (membrane marker), and Flag.

5. Previous reports of P27KIP1 lysosomal localization did not specify glutamine as the only amino acid involved in this process. Further, arginine and leucine are both also linked to mTORC1 activity. Thus, if arginine and leucine were also dysregulated in response to hSPAR expression, it would be interesting to further probe the role of hSPAR in regulating their uptake by first probing the effect of hSPAR on their entry transport proteins.

Response: This paragraph includes 2 questions. Firstly, as suggested, we have conducted immunoblotting experiments to detect the impact of hSPAR on the expression levels of

arginine transporter (SLC7A1) and leucine transporter (SLC7A5). Our results show that hSPAR does not affect the expression levels of SLC7A1 or SLC7A5 (revised Figure EV3C and EV3D, also shown below).

Revised Figure EV3C and EV3D: Immunoblotting against SLC7A1, SLC7A5, Flag and GAPDH in extracts from MDA-MB-231 cells transfected with Vector Ctrl, ΔATG1+2, or Flag-hSPAR.

Secondly, to determine whether arginine and leucine would affect lysosomal-localization of P27KIP1, we also performed subcellular fractionation assays with or without arginine/leucine supplementation in MDA-MB-231 cells. Our results show that the depletion of arginine or leucine does not affect the lysosomal localization of P27KIP1 (revised Figure EV3E, also shown below).

Revised Figure EV3E: Immunoblotting of whole cell extracts (Left panel), cytoplasmic (lysosome components removed) and lysosomal extracts (Right panel) prepared from MDA-MB-231 cells cultured with or without arginine, leucine or glutamine against P27KIP1, GAPDH, LAMP2 (lysosomal marker) and β-Tubulin (cytoplasmic marker).

6. Conclusions throughout the manuscript from imaging studies are supported by representative images of live cell images/immunofluorescence and single line-scans, which would benefit from additional analysis. Further quantification of signal co-localization such as Pearson's correlation coefficient or Mander's overlap coefficient would be helpful to support claims of localization. Additional techniques such as in situ proximity ligation assay (PLA) could be used to visualize co-localization and protein-protein interactions and lysosomal localization and would significantly strengthen the manuscript.

Response: This point is related to Reviewer#1's minor point2. Under minor point2, Reviewer#1 suggested us to use LAMP1/2 as the lysosomal marker. As suggested, we have used LAMP1 in our revised manuscript. The related figures are revised Figure 2E, 5G, 7B and

7C. Following the suggestions herein, we used Pearson's correlation coefficient for the revised Figure 2E, 5G, 7B and 7C, as well as Figure 2D which is based on live cells. The Pearson's correlation coefficient analyses are consistent with our original results using Lyso-Tracker as the lysosome marker (original Figure 2E, 5G, 7B, and 7C, and now as revised Figure EV2A, EV3B, EV5C and EV5D). The revised Figure 7B is shown below as a representative.

With these newly provided results, we have strengthened our conclusion about the investigated co-localization and protein-protein interactions. We have also discussed these results with the Editor, and have obtained the Editor's approval to waive the PLA experiments. We hope that Reviewer#1 would also consider our explanation as reasonable and acceptable.

Revised Figure 7B: Co-immunofluorescence staining of P27KIP1 (green) and LAMP1 (red) in MDA-MB-231 cells transfected with the indicated constructs. Nuclei were stained with Hoechst (blue). The graphs display the fluorescence intensity (arbitrary units) of P27KIP1 and LAMP1 over the distance from adjacent image (depicted by the arrows). The graphs display the values of Pearson's correlation Rr of P27KIP1 and LAMP1 with the indicated constructs. Scale bar, 25 μm .

7. To better compare differences cancer cells to normal, healthy cells, MCF10A mammary cells should be used as normal cell controls for MDA-MB-231 cells (for example in Fig. 5e).

Response: Original Figure 5e is the data from metabolomics. Following Reviewer1's suggestion, we have performed a glutamine uptake experiment using MCF10A cells as a control. Our results indicate that these cells do not exhibit any significant changes in cellular glutamine levels, reinforcing our previous observations (revised Figure EV3A, also shown below). Given these new results, we have discussed with the Editor and obtained his approval to replace metabolomics experiments with the above biochemical experiments because that the main focus of the current manuscript is glutamine-mediated events.

Revised Figure EV3A: Levels of glutamine in MCF10A cells transfected with Vector Ctrl, ΔATG1+2 or Flag-hSPAR.

8. This raises the question, why are the effects of hSPAR not observed in non-cancerous cells; further discussion of this is warranted?

Response: As suggested, we have added discussion to the revised manuscript. The related "Discussion" is also provided below.

Previously, Matsumoto *et al.* have reported that SPAR interacts with V-ATPase and facilitates V-ATPase's inhibitory role on mTORC1 assembly in HEK293T cells upon amino acid re-stimulation (Matsumoto *et al.*, 2017). In our work, hSPAR-inhibited effect on mTOR activity occurs in breast cancer cells under complete culturing media and mice with regular diet. Under these conditions, hSPAR shows no interaction with V-ATPase or other reported lysosomal proteins. Instead, hSPAR fulfills its mTOR inhibitory effect through acting as a cancer specific metabolism regulator and P27KIP1 stabilizer as we discussed above. Therefore, we have identified a new regulatory mechanism of hSPAR in breast cancer cells, distinct from HEK293T and MCF10A cells. The underlying mechanisms for such differential in different types of cells await further exploration.

9. Where does the ability of SPAR to bind the v-ATPase and hamper the assembly of mTORC1 fit into the story?

Response: In the present manuscript, by performing Co-IP, mass spectrum analysis, and immunoblotting validations, our results show that hSPAR does not bind to V-ATPase in MDA-MB-231 cells (original and revised Figure 3). V-ATPase is not involved in the tested roles of hSPAR in our current study.

Minor comments:

1. It would be useful to include a discussion of why TRIM21 was selected out of the list of hSPAR-interacting proteins.

Response: In this manuscript, by performing Co-IP and mass spectrum analysis, our results show that hSPAR binds to E3 ubiquitin ligase TRIM21 in MDA-MB-231 cells. Moreover, the predicted hSPAR structure by Dali also suggest that hSPAR might interact with ubiquitin

protein ligase (shown below). Given these results, we have selected TRIM21 as the focus of our manuscript. Considering the possible application to our future investigations, the Dali analysis and our explanation herein are not included in the revised manuscript.

Figure Legend: A Gene Ontology annotation assay of proteins similar to hSPAR structure by Dali database (www.embl-ebi.ac.uk/dali).

2. LysoTracker probes are fluorescent acidotropic probes for labeling and tracking acidic organelles in live cells. It is unusual to have LysoTracker used in live cells followed by immuno-fluorescence staining. Why wasn't LAMP1/2 immunostaining performed here?

Response: As suggested, we have re-conducted all co-localization experiments (original Figure 2E, 5G, 7B and 7C) by using LAMP1 as an additional lysosome marker following the experimental methods outlined in the reference (e.g. Matsumoto *et al.*, 2017). All these new results are consistent with our original results using Lyso-Tracker as the lysosome marker (revised Figure 2E, 5G, 7B and 7C). Our original results using Lyso-Tracker (original Figure 2E, 5G, 7B, and 7C) are now as revised Figure EV2A, EV3B, EV5C and EV5D. The Figure 7C is shown below as a representative.

Revised Figure 7C: Co-immunofluorescence staining of mTOR (green) and LAMP1 (red) in MDA-MB-231 cells transfected with the indicated constructs. Nuclei were stained with Hoechst (blue). The graphs display the fluorescence intensity (arbitrary units) of mTOR and LAMP1 over the distance from adjacent

image (depicted by the arrows). The graphs display the values of Pearson's correlation R_r of mTOR and LAMP1 with the indicated constructs. Scale bar, 25 μm .

3. Please indicate the phosphorylation sites for phospho-proteins evaluated via Western blotting on all figures.

Response: As suggested, we have indicated the phosphorylation sites for phospho-proteins evaluated *via* Western blotting. These figures include revised Figure 2F, 2H, 2J, 3C, 4D, 4F, 6A, 6D, 6F, 7D, 7E, 7F and 7O (originally also as Figure 2F, 2H, 2J, 3C, 4D, 4F, 6A, 6D, 6F, 7D, 7E, 7F and 7O). The representative revised Figure 2F is shown below.

Revised Figure 2F (original Figure 2F): Immunoblotting against p-mTOR, mTOR, p-S6K, S6K, p-S6, S6, Flag and GAPDH in extracts from MDA-MB-231 cells transfected with Vector Ctrl, $\Delta\text{ATG1+2}$, or Flag-hSPAR.

4. Quantification of Western blots should be normalized to Ctrl conditions so that changes in relative expression levels between treatments can be more easily visualized by the reader.

Response: As suggested, we have quantified all the Western blots by normalizing to Ctrl. These figures include revised Figure 1C, 1D, 2G, 2I, 3D, 4E, 4G, 5C, 6B, 6E, 6G, EV1E, EV2E, EV3D, EV4B, EV4F, EV4J, EV5E-5G and EV5L (originally also as Figure 1C, 1D, 2G, 2I, 3D, 4E, 4G, 5C, 6B, 6E, 6G and Extended Data Figure 5B, 5F, 5J, 7A-C and 7H). The representative revised Figure 2G is shown below.

Revised Figure 2G (original Figure 2G): Quantified relative levels of p-mTOR/mTOR, p-S6K/S6K and p-S6/S6 (n=3 independent biological samples).

Data are presented as the mean \pm SEM and analyzed using One-way ANOVA with Dunnett' multiple comparisons test.

5. Please include amino acid numbers on construct diagrams to improve the clarity of the construct design.

Response: As suggested, we have included amino acid numbers on the construct diagrams in the revised Figure EV1A, EV5A and EV5B (originally as Extended Data Figure 1A, 6A and 6B). These data are also shown below.

Revised Figure EV1A (original Extended Data Figure 1A): The fully conserved residues are indicated by red ‘*’, the specially similar residues are indicated by blue ‘:’, and the weakly similar residues are indicated by green ‘.’. N-terminal domain is highlighted with blue box, TM domain is highlighted with red box, C-terminal domain is highlighted with yellow box.

Left, Revised Figure EV5A (original Extended Data Figure 6A): hSPAR is composed of an N-terminal domain, a transmembrane (TM) domain and a C-terminal domain as predicted by Alphafold2. **Right, Revised Figure EV5B (original Extended Data Figure 6B):** Diagram of Flag-tagged full-length hSPAR and three hSPAR domain deletion variants.

6. Please increase the size of all scale bars. Currently, they are too small to read and even see in some images.

Response: As suggested, the size of all scale bars have been increased.

7. No citation was included for the previously reported treatment for the removal of soluble P27KIP1 or mTOR from cells.

Response: As suggested, we have cited the related reference (Nowosad *et al.*, 2020) in the “Methods” section.

8. There were several typos throughout the manuscript, including on line 164, 222, 228, 240, 290, and 359.

Response: We sincerely appreciate Reviewer#1 for the careful guidance and apologize for the typos in our original manuscript. We have carefully checked the entire manuscript and made appropriate modifications-in the revised manuscript.

Comments from Reviewer #2: This paper focuses on the negative regulation of mTORC1 by the micropeptide, hSPAR. It is topical, because there is growing realisation that micropeptides, derived from what were previously defined as 'non-coding RNAs', have important functions. A role for hSPAR downregulation in mTORC1 signalling and muscle regeneration has previously been shown (Matsumoto et al., 2017 Nature). It was postulated to involve an inhibitory interaction with the V-ATPase, which recruits mTORC1 to the lysosomal membrane on acute addition of amino acids in HEK293 cells, but the mechanism mediating its long-term effect on mTORC1 during muscle regeneration was not characterised. To some extent, the current study builds on these findings, providing mechanistic insights into the role for hSPAR in cancer mediated via its C-terminus.

The authors identify two additional mechanisms by which hSPAR inhibits mTORC1. They show that the C-terminus of hSPAR (hSPAR-C) promotes the non-canonical role of P27KIP1 as an inhibitor of the mTORC1 axis by stabilising cytoplasmic P27KIP1, and also leads to glutamine deprivation-induced lysosomal localisation of P27KIP1. Furthermore, they show that hSPAR-C inhibits the amino acid transporter SLC38A2, therefore decreasing cellular glutamine levels, and blocking mTORC1 through this combinatorial mechanism more strongly than suppressing the two mechanisms separately. This logical analysis, involving overexpression and knockdown experiments, sheds light on the mechanisms underlying their findings at the beginning of the manuscript that hSPAR acts as a tumour suppressor in breast cancer cells, which might not be considered entirely surprising given the previous publication on its mTORC1-mediated role in muscle cell regeneration. The schematic in Figure 7p nicely illustrates the final model and could be the basis for a helpful graphical abstract.

Response: We sincerely thank Reviewer#2 for the positive comments on our manuscript, and the constructive suggestions that are great helpful to improve the impact of our manuscript.

I have some concerns that I think need to be addressed to reflect the current state of the field, appropriately present the data, and to support the conclusions:

1. I am not an expert in this specific field, but it appears that the authors have omitted a number of publications on the lncRNA associated with hSPAR, LINC00961. They cite two references, Matsumoto et al., 2017, and Spencer et al., 2020, which identifies differing roles for hSPAR and LINC00961 in endothelial cells.

There are, however, also several other papers where roles have been identified for LINC00961, but the possible contribution of hSPAR has not been fully studied and some of these are relevant to cancer, so I am surprised that none of these have been cited eg.

Jiang B, Liu J, Zhang YH, Shen D, Liu S, Lin F, Su J, Lin QF, Yan S, Li Y, Mao WD, Liu ZL. Long noncoding RNA LINC00961 inhibits cell invasion and metastasis in human non-small cell lung cancer. *Biomed Pharmacother.* 2018 Jan;97:1311-1318. doi:

10.1016/j.biopha.2017.11.062.

Huang Z, Lei W, Tan J, Hu HB. Long noncoding RNA LINC00961 inhibits cell proliferation and induces cell apoptosis in human non-small cell lung cancer. *J Cell Biochem.* 2018 Nov;119(11):9072-9080. doi: 10.1002/jcb.27166.

Lu XW, Xu N, Zheng YG, Li QX, Shi JS. Increased expression of long noncoding RNA LINC00961 suppresses glioma metastasis and correlates with favorable prognosis. *Eur Rev Med Pharmacol Sci.* 2018 Aug;22(15):4917-4924. doi: 10.26355/eurev_201808_15630.

Chen D, Zhu M, Su H, Chen J, Xu X, Cao C. LINC00961 restrains cancer progression via modulating epithelial-mesenchymal transition in renal cell carcinoma. *J Cell Physiol.* 2019 May;234(5):7257-7265. doi: 10.1002/jcp.27483.

Yin J, Liu Q, Chen C, Liu W. Small regulatory polypeptide of amino acid response negatively relates to poor prognosis and controls hepatocellular carcinoma progression via regulating microRNA-5581-3p/human cardiolipin synthase 1. *J Cell Physiol.* 2019 Aug;234(10):17589-17599. doi: 10.1002/jcp.28383.

Zhang L, Shao L, Hu Y. Long noncoding RNA LINC00961 inhibited cell proliferation and invasion through regulating the Wnt/ β -catenin signaling pathway in tongue squamous cell carcinoma. *J Cell Biochem.* 2019 Aug;120(8):12429-12435. doi: 10.1002/jcb.28509.

Pan LN, Sun YR. LINC00961 suppresses cell proliferation and induces cell apoptosis in oral squamous cell carcinoma. *Eur Rev Med Pharmacol Sci.* 2019 Apr;23(8):3358-3365. doi: 10.26355/eurev_201904_17699.

Mu X, Mou KH, Ge R, Han D, Zhou Y, Wang LJ. Linc00961 inhibits the proliferation and invasion of skin melanoma by targeting the miR-367/PTEN axis. *Int J Oncol.* 2019 Sep;55(3):708-720. doi: 10.3892/ijo.2019.4848.

Wu H, Dai Y, Zhang D, Zhang X, He Z, Xie X, Cai C. LINC00961 inhibits the migration and invasion of colon cancer cells by sponging miR-223-3p and targeting SOX11. *Cancer Med.* 2020 Apr;9(7):2514-2523. doi: 10.1002/cam4.2850. Epub 2020 Feb 11. PMID: 32045135; PMCID: PMC7131851.

Response: The related references have now been cited in the revised "Introduction" section with discussion. The related "Introduction" is also provided below.

Recently, Matsumoto *et al.* found that *LINC00961* also has the capacity to encode functional micropeptide known as small regulatory polypeptide of amino acid response (SPAR)(Matsumoto *et al.*, 2017; Spencer *et al.*, 2020). *LINC00961* is located on human chromosome 9. Previous studies have shown that *LINC00961* functions as an lncRNA to regulate cell proliferation, invasion, and apoptosis in different cancer cells(Chen *et al.*, 2019; Huang *et al.*, 2018; Jiang *et al.*, 2018; Lu *et al.*, 2018; Mehrpour Layeghi *et al.*, 2020; Mu *et al.*, 2022; Pan & Sun, 2019; Wu *et al.*, 2020a; Yin *et al.*, 2019; Zhang *et al.*, 2019).

Most relevant to the manuscript, Mu *et al.*, 2022 and other groups looking at the normal function of LINC00961 have postulated an effect on PTEN/PI3K, which modulates mTORC1 activity. In this case, the authors not only should cite and discuss the reference, but it would be straightforward to check whether the genetic manipulations they undertake affect PI3K signalling in addition to mTORC1. For example, are there differences in PI3K regulation when LINC00961 is knocked down but not when hSPAR is overexpressed?

Response: We sincerely thank Reviewer#2 for this scientific guidance. To address this concern about PI3K signaling, we have detected the impact of hSPAR on the AKT phosphorylation level, as that AKT is a downstream target of PI3K signal pathway, and that the level of AKT phosphorylation represents the activity of PI3K signal. Our result show that hSPAR overexpression does not affect AKT phosphorylation or the activity of PI3K signal (shown below). Considering these negative results suggest an indirect link with the scope of our current manuscript, we did not include this result in our revised manuscript.

Figure legend: Immunoblotting against p-AKT, AKT, Flag and GAPDH in extracts from MDA-MB-231 cells transfected with Vector Ctrl, Δ ATG1+2, or Flag-hSPAR.

One additional clinical study was directly related to breast cancer and so should also be discussed.

Mehrpour Layeghi S, Arabpour M, Esmaceli R, Naghizadeh MM, Tavakkoly Bazzaz J, Shakoori A. Evaluation of the potential role of long non-coding RNA LINC00961 in luminal breast cancer: a case-control and systems biology study. *Cancer Cell Int.* 2020 Oct 2;20:478. doi: 10.1186/s12935-020-01569-1.

Response: As suggested, this study has been cited and discussed in the revised manuscript together with other *LINC00961*-related references.

2. Based on their data, the authors conclude that hSPAR-C only affects glutamine levels and cell growth specifically in cancer cells. This conclusion seems to be based on their work in breast cancer cell lines compared to HEK-293T cells, which the authors term 'non-cancerous', but, as a cell line, they are far from normal. The conclusion made by the authors from the data in Fig. S5 is that hSPAR has no effect on mTORC1 in non-cancerous cells, but the Matsumoto et al., 2017 paper suggests that it does regulate steady-state mTORC1 activity in regenerating muscle over a period of days. Unless other data are provided, I think the previously published data need to be discussed and this conclusion toned down.

Response: In the revised manuscript, we have incorporated MCF10A as an additional control cell line for our investigations on the glutamine uptake. Our results indicate that hSPAR does not change cellular glutamine levels in MCF10A cells, reinforcing our previous observations in HEK293T cells (revised Figure EV3A, also shown below). In addition, as suggested, we have toned down our related conclusions across the entire revised manuscript, and we have removed the description of HEK293T as "non-cancerous cells".

Revised Figure EV3A: Levels of glutamine in MCF10A cells transfected with Vector Ctrl, ΔATG1+2 or Flag-hSPAR.

3. The co-localisation data analysis in Figs. 5g and 7c appears extremely selective in several cases, choosing areas where there is no yellow signal in genetic backgrounds where the authors argue there is no colocalization, when other cells appear to have significant colocalization, eg. 5g: Gln+ top cell; 7c: FLAG-hSPAR bottom cell and cells for deltaTM-hSPAR; also, vice versa in 5g, Gln-. It would be more appropriate to measure the overall colocalization throughout all cells to confirm a significant change. In this respect, the cell fractionation in Fig. 5h is also a concern. The lysosomal fraction for Gln- seems to have a higher level of tubulin present, which would explain the result through cytoplasmic contamination.

Response: As suggested, we have measured the overall colocalization of the related proteins and lysosome markers throughout all cells for the original Figure 5G and 7C (now as the revised Figure 5G and 7C, also shown below). Please note that, in the revised Figure 5G and 7C, we have used LAMP1 as an additional lysosome marker as suggested by Reviwer#1, (minor point2) in order to further strengthen the credibility of experimental data on protein-protein interactions. All of these new results (revised Figure 2E, 5G, 7B and 7C) suggested by both Reviewers are consistent with our previous results using Lyso-Tracker as the lysosome marker (original Figure 2E, 5G, 7B and 7C, and now as revised Figure EV2A, EV3B, EV5C and EV5D).

Revised Figure 5G: Co-immunofluorescence staining of P27KIP1 (green) and the lysosomal marker LAMP1 (red) in MDA-MB-231 cells cultured with or without glutamine. Cells were permeabilized with digitonin to remove the soluble P27KIP1. Nuclei were stained with Hoechst (blue). The graphs display the fluorescence intensity (arbitrary units) of P27KIP1 and LAMP1 over the distance from adjacent

image (depicted by the arrows). The graphs display the values of Pearson's correlation R_r of P27KIP1 and LAMP1 with or without glutamine. Scale bar, 25 μm .

Revised Figure 7C: Co-immunofluorescence staining of mTOR (green) and LAMP1 (red) in MDA-MB-231 cells transfected with the indicated constructs. Nuclei were stained with Hoechst (blue). The graphs display the fluorescence intensity (arbitrary units) of mTOR and LAMP1 over the distance from adjacent image (depicted by the arrows). The graphs display the values of Pearson's correlation R_r of mTOR and LAMP1 with the indicated constructs. Scale bar, 25 μm .

As suggested, for the original Figure 5H, we have re-conducted cytoplasm-lysosome separation experiment as the revised Figure 5H. Our results show that the cellular components and lysosomal components are well separated (the lysosomal components do not contaminate by cytoplasmic components).

Revised Figure 5H: Immunoblotting of whole cell extracts (Left panel), cytoplasmic (lysosome components removed) and lysosomal extracts (Right panel) prepared from MDA-MB-231 cells cultured with or without glutamine against P27KIP1, GAPDH, LAMP2 (lysosomal marker) and β -Tubulin (cytoplasmic marker).

4. Much of the data analysis involves multiple comparisons, yet the data are apparently analysed by two-tailed t-test. A non-parametric multiple comparisons test should be used. I think this is unlikely to affect the final conclusions, but the correct test should be employed.

Response: As suggested, we have used non-parametric multiple comparisons test in the revised Figure 1C, 1D, 1G, 1K, 2G, 2L, 3D, 3F, 4E, 4G, 4I, 5F, 5J, 6G, EV1E, EV1G, EV2E, EV3A, EV3D, EV4B, EV4D, EV4F, EV4H, EV4J, EV4L, EV5E, EV5F and EV5H (originally also as Figure 1C, 1D, 1G, 1K, 2G, 2L, 3D, 3F, 4E, 4G, 4I, 5F, 5J, 6G and Extended Data Figure 5B, 5D, 5F, 5H, 5J, 5L, 7A, 7B and 7D). As expected by the Reviewer#2, these new analyses show similar conclusion with our previous analyses.

Minor comments

Line 78: The V-ATPase is an activator of mTORC1 lysosomal recruitment, not an inhibitor?

Response: The role of V-ATPase on mTORC1 is complicated. Upon amino acids starvation, V-ATPase tightly binds with Ragulator and Rags to inhibit mTORC1 lysosomal localization and activation. With amino acids stimulation, V-ATPase complex is disassembled, thus promoting mTORC1 activation. In the original manuscript, we cited V-ATPase's role under the condition of amino acids starvation, but did not express it clearly. In the revised manuscript, we have modified the related sentence (also shown below).

In amino acids re-stimulated HEK293T cells, lysosomal-localized SPAR tightly bound with V-ATPase and stabilized V-ATPase's interaction with Ragulator complex to hamper the proper mTORC1 assembly (Matsumoto *et al.*, 2017).

The manuscript is on the whole well-written, but would benefit from careful proof reading. I have highlighted some concerns about the language used to express specific points below as examples:

Line 37: Replace 'discovered' with 'uncovered'

Line 56: 'proven as the' with 'shown to be'

Lines 63-64: Replace 'far beyond' with 'not well'.

Lines 76-77: Replace 'cell' with 'cells'.

Line 237: 'glutamine entry transport' would be better phrased as, 'glutamine transporter'

Line 371: Substitute, 'regulations on' with 'regulators'

Line 371, 379 etc: add 'the' before 'cytoplasm', 'glutamine transporter' respectively etc

Response: We sincerely appreciate Reviewer#2 for the careful guidance. We have carefully checked the entire manuscript and made appropriate modifications throughout the revised manuscript.

Dear Dr Wang,

Thank you for submitting your revised manuscript (EMBOJ-2024-118191R) to The EMBO Journal, as well for your patience with our response. Your amended study was sent back to the two referees for their scientific re-evaluation, and we have received detailed comments from all of them, which I enclose below. As you will see, the experts state that the work has been substantially improved by the revisions and they are now in favour of publication, pending minor revision.

Thus, we are pleased to inform you that your manuscript has been accepted in principle for publication in The EMBO Journal.

Please consider the remaining points by referee #2 carefully and adjust data presentation and statistics where appropriate.

We also now need you to take care of a number of issues related to formatting and data presentation as detailed below, which should be addressed at re-submission.

Please contact me at any time if you have additional questions related to below points.

As you might have seen on our web page, every paper at the EMBO Journal now includes a 'Synopsis', displayed on the html and freely accessible to all readers. The synopsis includes a 'model' figure as well as 2-5 one-short-sentence bullet points that summarize the article. I would appreciate if you could provide this figure and the bullet points.

Thank you for giving us the chance to consider your manuscript for The EMBO Journal. I look forward to your final revision.

Again, please contact me at any time if you need any help or have further questions.

Kind regards,

Daniel Klimmeck

>> Please add maximally five keywords to your study.

>> Remove the figures from the manuscript .doc file.

>> Author Contributions: Please remove the author contributions information from the manuscript text. Note that CRediT has replaced the traditional author contributions section as of now because it offers a systematic machine-readable author contributions format that allows for more effective research assessment. and use the free text boxes beneath each contributing author's name to add specific details on the author's contribution.

More information is available in our guide to authors.
<https://www.embopress.org/page/journal/14602075/authorguide>

>> Add a Reagents and Tools table to the Methods section, listing key reagents, experimental models, software and relevant equipment.

>> Rename the current 'Data analysis' to 'Statistical Analysis'.

>> Callouts: add figure callouts in the manuscript text for panels Fig EV4 A-L.

>> As to our journal policies we kindly ask you to check & clarify lack of data intensity within Fig.2E .

>> Consider additional changes and comments from our production team as indicated below:

DATA CHECK: FAIL

- DAS:

1. Please note that the accession ID for the Metabolomics Workbench (accession codes) is not provided in the data availability statement.

>>> now added

2. Please note that the specific URL for Metabolomics Workbench is not provided in the data availability statement.

>>> now added

- Figure legends:

1. Please define the annotated p values ****/***/**/* as well as provide the statistical test and exact p-values for the same in the legend of figures 1A, B as appropriate.

2. Please note that the exact p values are not provided in the legends of figures 1C, D, K, M; 2G, I, L; 3D, F; 4E, G, I; 5J; 6G, 7J, L; EV4 F, J, L; EV5 E, F, G, L.

3. Please indicate the statistical test used for data analysis in the legend of figure 5E."

4. Please note that the box plots need to be defined in terms of minima, maxima, centre, bounds of box and whiskers, and percentile in the legends of figures 1A, B.

5. Please note that information related to n is missing in the legends of figures 7J, K, L.

6. Although 'n' is provided, please describe the nature of entity for 'n' in the legends of figures 1A, B.

7. Please note that the error bars are not defined in the legends of figures EV2 B, G; EV3 A."

8. Please note that the scale bar needs to be defined for figures 1E, 2A; 7M, N; EV1 B, F; EV5 J, L.

9. Please note that the white arrow heads are not defined in the legend of figure EV1 F. This needs to be rectified.

Referee #1:

We have carefully considered the revised manuscript. The authors have made a sincere attempt to address the points that we raised in the original review. Thus, we feel that the manuscript is now acceptable for publication.

Referee #2:

The authors have generally addressed my comments and concerns positively and the changes made in response to both referees' comments have improved the quality and robustness of the findings, so that the conclusions are broadly supported by the data. As previously stated, I think the manuscript does shed new light on our understanding of mTORC1 regulation in breast cancer cells, primarily focusing on the analysis of MDA-MB-231 cells, but also using other cell lines. It should be of general interest to the researchers.

Regarding my point 1, I do think it is important for the authors to present the new data on AKT activation as Supplementary Data, because this directly shows that a previously reported mechanism (Mu et al., 2022) that might modulate mTORC1 signalling is not a major player under the conditions they are using.

For my point 4, the authors have now undertaken a parametric multiple comparisons test, ANOVA, rather than the suggested non-parametric one, eg Kruskal-Wallis. I am sure it will not change the ultimate outcome for the key results, but I think the non-parametric test would be the most appropriate.

Referees' comments:**Referee #1:**

We have carefully considered the revised manuscript. The authors have made a sincere attempt to address the points that we raised in the original review. Thus, we feel that the manuscript is now acceptable for publication.

We thank Referee #1 for the positive evaluation of our work.

Referee #2:

The authors have generally addressed my comments and concerns positively and the changes made in response to both referees' comments have improved the quality and robustness of the findings, so that the conclusions are broadly supported by the data. As previously stated, I think the manuscript does shed new light on our understanding of mTORC1 regulation in breast cancer cells, primarily focusing on the analysis of MDA-MB-231 cells, but also using other cell lines. It should be of general interest to the researchers.

Regarding my point 1, I do think it is important for the authors to present the new data on AKT activation as Supplementary Data, because this directly shows that a previously reported mechanism (Mu et al., 2022) that might modulate mTORC1 signalling is not a major player under the conditions they are using.

For my point 4, the authors have now undertaken a parametric multiple comparisons test, ANOVA, rather than the suggested non-parametric one, eg Kruskal-Wallis. I am sure it will not change the ultimate outcome for the key results, but I think the non-parametric test would be the most appropriate.

We thank Referee #2 for the positive evaluation of our work. While Referee #2 is generally satisfied with our responses, two minor points still require clarification, which we address below.

Regarding Referee #2's two points, we have followed point 1 to "present the new data on AKT activation" as the Appendix Figure S1 with corresponding descriptions in the main text (also provided below), methods, and source data sections.

To investigate the potential function of TRIM21 in the observed hSPAR-mediated suppression of mTOR activation and cell proliferation, we investigated downstream targets of mTOR signaling in MDA-MB-231 cells in the presence of both Flag-hSPAR and GFP-TRIM21. We found that co-expression of GFP-TRIM21 rescued the Flag-hSPAR-mediated changes in mTOR signaling (Fig. 3C, D), cell proliferation (Fig. 3E, F), and cell cycle progression (Fig. EV2B). PI3K-AKT is a previously reported upstream regulatory pathway of mTOR signaling. Our results showed that overexpressing hSPAR did not affect serine 473 phosphorylation of AKT, an indicator of the PI3K activation (Appendix Fig. S1). These results demonstrate that the inhibitory roles of hSPAR in mTOR signaling and cell proliferation require TRIM21.

Revised Appendix Fig. S1: Overexpressing hSPAR does not affect serine 473 phosphorylation of AKT. Immunoblotting against p-AKT, AKT, Flag and GAPDH in extracts from MDA-MB-231 cells transfected with Vector Ctrl, Δ ATG1+2, or Flag-hSPAR (n=3 independent biological samples).

For Referee #2's point 2 (to change the current ANOVA analysis to non-parametric analysis), we think our current analysis is more appropriate. The underlying reason is: although non-parametric analysis may have some advantages than one-way ANOVA analysis when the number of group samples is within a moderate range (12 to 100), however, when the sample number is either larger than 100 or less than 12, non-parametric analysis is not suitable. Under the latter two conditions, using non-parametric analysis may generate misleading results. While, the application of one-way ANOVA analysis has no restriction regarding the number of group samples. Additionally, we have checked recently published articles in *The EMBO Journal*. For those experiments with their sample numbers less than 12, one-way ANOVA was used (e.g. doi: 10.1038/s44318-024-00309-9, doi: 10.1038/s44318-024-00285-0, doi: 10.1038/s44318-024-00292-1, doi: 10.1038/s44318-024-00246-7). In our manuscript, the group sample numbers of all wet-lab experiments are less than 12. Collectively, we think it is appropriate to use one-way ANOVA analysis. Therefore, in the submitted revised manuscript, this part remains unaltered.

All the text changes have been highlighted with red in the revised manuscript.

Dear Dr Wang,

Thank you for submitting the revised version of your manuscript. I have now evaluated your amended manuscript and concluded that the remaining minor concerns have been sufficiently addressed.

I am thus pleased to inform you that your manuscript has been accepted for publication in the EMBO Journal.

Related I would like to hereby ask your consent on keeping the referee figures included in this file.

On a different note, I would like to alert you that EMBO Press offers a format for a video-synopsis of work published with us, which essentially is a short, author-generated film explaining the core findings in hand drawings, and, as we believe, can be very useful to increase visibility of the work. Please see the following link for representative examples and their integration into the article web page:

<https://www.embopress.org/doi/full/10.15252/emj.2019103932>

Best regards,

Daniel Klimmeck

Daniel Klimmeck, PhD
Senior Editor
The EMBO Journal
EMBO
Postfach 1022-40
Meyerhofstrasse 1
D-69117 Heidelberg
contact@embojournal.org
Submit at: <http://emboj.msubmit.net>